# Memory-Distilled Selection for Noise-Robust Anomaly Detection

**Sirojbek Safarov** [* 1]   **Jaewoo Park** [* † 1]   **Yoon Gyo Jung** [* 2]   **Kuan-Chuan Peng** [3]   **Wonchul Kim** [1]   **Seongdeok Bang** [1]   **Octavia Camps** [2]

## Abstract

Anomaly detection (AD) under data contamination is critical for deploying unsupervised defect detection in industrial environments, where curating perfectly clean training sets is impractical. However, existing methods are sensitive to contamination, suffering significant performance degradation as the noise ratio increases. In this paper, we propose Memory-Distilled Selection (MeDS), a training algorithm based on data selection. MeDS constructs an ensemble of partial memories via random subsampling, where the resulting sparsity acts as a low-pass filter that captures nominal patterns across a wide range of noise ratios, enabling coarse-level identification of contaminated samples. The aggregated distances to the bootstrapped memories are then distilled into a reconstruction score network, which is subsequently fine-tuned on clean data filtered using scores from the distilled model, enabling fine-grained localization of anomalies. MeDS is robust across a wide range of noise ratios without requiring noise-ratio-specific hyperparameter tuning, achieving 99.16% image-level AUROC on MVTecAD at a 40% noise ratio, and attaining state-of-the-art performance on both VisA and Real-IAD under noisy settings. We thoroughly verify the efficacy of MeDS on industrial AD benchmarks under noisy data scenarios, accompanied by in-depth empirical analyses.

## 1. Introduction

In high-throughput manufacturing, vision-based *anomaly detection* (AD) systems enable early defect detection and protect product quality. A common approach trains on images assumed to be normal and flags any deviation at inspection time—a strategy that has seen substantial progress (Zavrtanik et al., 2021; Roth et al., 2022; Defard et al., 2021; Zhang et al., 2023; Wang et al., 2024; Li et al., 2024). Yet the assumption of a contamination-free training set is often unrealistic: assembling and verifying large-scale industrial data is costly, and even rigorous manual screening inevitably overlooks a non-negligible fraction of anomalous samples. When contaminated data enter the training set, the performance of AD models degrades significantly.

Existing approaches to this contaminated-data setting either (1) impose strong assumptions on the statistical nature of the training data (Chen et al., 2022; Jung et al., 2025), (2) assume knowledge of the anomaly ratio in the training set (Jiang et al., 2022), or (3) assume knowledge of the range of anomaly scores for anomalous samples (Im et al., 2025). Consequently, optimal hyperparameters differ across datasets and contamination levels, requiring data-specific tuning and resulting in critical performance drops when the noise ratio rises. To the best of our knowledge, no existing method achieves robust image-level and pixel-level performance across diverse noise ratios.

To address this challenge, we propose Memory-Distilled Selection (MeDS), a training framework designed to maintain high performance across diverse noise ratios for both image-level detection and pixel-level localization. First, we introduce a *bootstrapped ensemble of memory banks* that aggregates statistics from multiple sparse, random subsamples of pretrained patch-level features extracted from the training data. As we show theoretically, the sparse nature of the memory ensemble acts as a low-pass filter under an appropriate subsampling ratio, separating normal patch features from anomalous ones in the training set. However, since the memory relies solely on features pretrained on external data, the discriminative gap between normal and anomalous features is bounded by what these frozen representations can capture. To overcome this, we *distill* the memory ensemble scores into a student network, which learns to amplify the separation through gradient-based optimization on the target domain while exploiting the early-learning regularization property of neural networks. Yet distillation alone presents a trade-off: early stopping preserves image-level robustness but yields insufficient training for pixel-level precision, while extended training risks overfitting to noise. To

---

[*]Equal contribution [†]Project lead. [1]AIVEX Inc. [2]Northeastern University [3]Mitsubishi Electric Research Laboratories (MERL). Correspondence to: Jaewoo Park <park.jaewoo@aivexvision.ai>.

*Proceedings of the 43rd International Conference on Machine Learning*, Seoul, South Korea. PMLR 306, 2026. Copyright 2026 by the author(s).

resolve this, we perform iterative data selection, fine-tuning the student network on self-filtered clean data. This enables extended training without overfitting, allowing MeDS to progress from coarse-grained noise filtering to fine-grained AD capable of precise pixel-level segmentation.

The design of progressively transitioning from low-pass filtering to fine-tuning with data selection makes MeDS robust across a wide range of contamination levels without requiring noise-ratio-specific hyperparameter tuning. On the MVTecAD benchmark, MeDS achieves 99.16% image-level AUROC at a 40% noise ratio, demonstrating strong robustness. Furthermore, it establishes state-of-the-art performance on the VisA and Real-IAD datasets under noisy settings. Beyond automated detection, we demonstrate the utility of MeDS for active label correction: by sorting training samples based on MeDS selection scores, we show that an annotator can efficiently identify and remove contaminated samples with minimal effort.

**Contributions.**

- We introduce a contamination-resilient AD training algorithm that exploits the sparsity of bootstrapped memory ensembles as a low-pass filter to isolate nominal patterns, subsequently refining pixel-level detection via distillation and iterative self-selection.

- We conduct extensive empirical analyses across datasets and noise levels, including ablation studies that dissect each component's role, detailed studies of hyperparameters, and a demonstration of the utility of MeDS for active label correction.

- We provide theoretical and empirical insights into how small-ratio memory subsampling yields a high-recall starting point under heavy contamination.

## 2. Related Work

**Anomaly detection with noisy data**   Methods for AD with noisy data assume that the training set may contain unlabeled anomalies. IGD (Chen et al., 2022) addresses label noise by fitting a single Gaussian descriptor and achieves strong image-level results on datasets such as CIFAR-10 and MNIST. However, its unimodal prior limits the fine-grained localization required in industrial AD. SoftPatch (Jiang et al., 2022) extends PatchCore (Roth et al., 2022) by incorporating a memory bank and pruning anomalous patch features using a Local Outlier Factor (LoF) score. InReaCh (McIntosh & Albu, 2023) retains only those patches that form symmetric 1-nearest-neighbor matches recurring across most training images, thereby ensuring the feature space remains tightly clustered. Because both methods are learning-free, they depend on pretrained embeddings and hand-crafted rules, making them sensitive to hyperparameter choices and

limiting their noise tolerance. FUN-AD (Im et al., 2025) trains a discriminator using binary pseudo-labels derived by thresholding normalized nearest-neighbor distances. However, its cross-entropy objective requires both normal and anomaly samples, and min-max normalization inevitably mislabels some normal samples as anomalies when contamination is low or absent. This implicit assumption of non-zero contamination causes degraded performance on clean data—the opposite failure mode of conventional one-class methods. SRR (Yoon et al., 2021) employs progressive sample selection but, as an image-embedding method, cannot localize anomalies at the pixel level. None of these methods provide the complete set of capabilities required for fine-grained defect detection under severe noise ratios. In contrast, MeDS minimizes reliance on hand-crafted rules by exploiting the noise resilience of subsampled memory ensembles and the early-learning robustness of neural networks, thereby enabling fine-grained localization of defects even when training data are largely contaminated.

**Relation to learning with noisy labels**   Our work draws on insights from the broader literature on learning with noisy labels. Reed et al. (2014) introduce a bootstrapping loss that combines the network's own soft predictions with noisy labels, demonstrating that noise averaging can protect the supervision signal. Our subsampled memory ensemble serves an analogous role: by averaging nearest-neighbor distances, it yields a noise-tolerant anomaly score. Rolnick et al. (2017) show that modern CNNs can generalize even when true labels are overwhelmed by random noise, while Liu et al. (2020) formalize the early-learning phenomenon and design a regularizer to exploit it. These findings motivate our use of a learnable reconstructor that leverages its simplicity bias through knowledge distillation. Finally, MentorNet (Jiang et al., 2018) demonstrates that a dynamically updated curriculum—one that progressively filters suspected noisy samples—markedly improves robustness, supporting the iterative self-selection strategy in the fine-tuning stage of MeDS. The high-level idea of progressive selection using a self-curated clean subset is well established in the classification literature. However, to our knowledge, a systematic formulation of progressive selection based on memory distillation—one that is tailored to industrial AD—has not been previously explored.

## 3. Preliminaries

### 3.1. Problem Setting: AD with Noisy Data

In conventional AD, models are trained on a dataset assumed to consist exclusively of normal samples. In contrast, the *noisy AD* setting relaxes this assumption: the training set $\mathcal{D}$ contains both normal and anomalous images, with their labels unknown. Formally, let $\mathcal{D} = \mathcal{D}_{\text{normal}} \cup \mathcal{D}_{\text{anomaly}}$ denote

the training set, where $\mathcal{D}_{\text{anomaly}}$ contains unlabeled anomalous samples. The *noise ratio* is defined as $|\mathcal{D}_{\text{anomaly}}|/|\mathcal{D}|$. At test time, the model must assign high anomaly scores to anomalous samples and low scores to normal samples.

### 3.2. Memory-Based Anomaly Detection

A memory-based AD model stores representative patch-level features from the training data in a memory bank $\mathcal{M}$. Given an input image $x \in \mathcal{X}$, the model computes an anomaly score map by measuring the distance from each patch feature to its nearest neighbor in $\mathcal{M}$. Formally, the *memory anomaly score* $s_{\mathcal{M}} : \mathcal{X} \to \mathbb{R}^{H \times W}$ is defined as:

$$s_{\mathcal{M}}(x)_{hw} = \min_{z \in \mathcal{M}} D\big(z, g(x)_{hw}\big), \tag{1}$$

where $g$ is a frozen pretrained encoder that produces a feature map $g(x) \in \mathbb{R}^{H \times W \times C}$, $g(x)_{hw} \in \mathbb{R}^C$ denotes the patch feature at spatial position $(h, w)$, and $D$ is the Euclidean distance metric in the latent space. $\mathcal{M}$ is typically constructed via greedy coreset sampling (Roth et al., 2022) from the set of all patch features extracted from the training data:

$$g(\mathcal{D}) := \big\{g(x)_{hw} \mid x \in \mathcal{D}, \ (h, w) \in [H] \times [W]\big\}, \tag{2}$$

where $[H] = \{1, 2, \ldots, H\}$ and $[W] = \{1, 2, \ldots, W\}$.

### 3.3. Teacher-Student Anomaly Detection

In the teacher-student AD framework, the anomaly score is defined as the discrepancy between a frozen pretrained teacher encoder $g$ and a trainable student network $f_\theta$. We refer to this score as the *reconstruction score*, denoted $s_\theta : \mathcal{X} \to \mathbb{R}^{H \times W}$:

$$s_\theta(x)_{hw} = D\big(g(x)_{hw}, f_\theta(x)_{hw}\big), \tag{3}$$

where $g(x)$ and $f_\theta(x)$ are the feature maps from the teacher and student, respectively. The student is trained to minimize the average reconstruction score over the training set:

$$\min_\theta \ \frac{1}{|\mathcal{D}|} \sum_{x \in \mathcal{D}} s_\theta(x). \tag{4}$$

In Reverse Distillation (Deng & Li, 2022), the student is implemented as a bottleneck decoder that receives the teacher's intermediate activations as input. Restricting the student's representational capacity ensures that the reconstruction score remains low only for in-distribution inputs, while out-of-distribution (anomalous) inputs incur high scores.

## 4. Proposed Method: MeDS

MeDS consists of three training phases which are executed in order: (1) bootstrapped memory construction via an ensemble of subsampled partial memories, whose sparsity

acts as a low-pass filter and thus removes noisy samples at a coarse level; (2) memory score distillation to the reconstruction score, which refines the anomaly scores and provides a well-initialized model; and (3) fine-tuning on self-selected training data, enabling the network to learn fine-grained reconstruction of normal patterns without overfitting to anomalous samples.

### 4.1. Memory Construction

**Bootstrapped memory** As the initial component of MeDS, we construct a memory-based model using an *ensemble of $B$ subsampled partial memories*, where the final anomaly score is the average of the individual scores from each subsampled memory:

$$s_{\mathbb{M}}(x) = \frac{1}{B} \sum_{b=1}^{B} s_{\mathcal{M}_b}(x). \tag{5}$$

Each partial memory $\mathcal{M}_b$ is designed to be a sparse representative of the original full memory bank, and hence is constructed by extracting a random subset of features from the full memory bank of the training set $g(\mathcal{D})$. Formally, with subsampling ratio $\rho \in (0, 1)$, we have $|\mathcal{M}_b| = \rho|g(\mathcal{D})|$. Unless specified otherwise, the sampling ratio is set to $\rho = 0.1$.

Theorem 1 below characterizes why sparse subsampling induces separation between normal and anomalous features:

**Theorem 1.** *Under a regularity condition, for any anomaly and normal patch features $q_{anom}$ and $q_{norm}$, the expected gap*

$$\Delta(m) := \mathbb{E}[D(q_{anom}, \mathcal{M})] - \mathbb{E}[D(q_{norm}, \mathcal{M})] \tag{6}$$

*satisfies $\Delta(m) > 0$, and it is decomposed into a second-order Taylor approximation $\Delta_0$ with the remainder $\epsilon_0(m)$:*

$$\Delta(m) = \Delta_0(m) + \epsilon_0(m), \tag{7}$$

*where $\Delta_0(m)$ is an integral with a weight function $\omega(m, r)$*

$$\Delta_0(m) = \int_0^\infty \delta(r) \cdot \omega(m, r) \, \mathrm{d}r \tag{8}$$

*with a non-negative function $\delta$. Notably, $\omega(m, r)$ is unimodal with respect to $m$. Here, the expectation $\mathbb{E}$ is over the memory $\mathcal{M}$ that is randomly subsampled from the extracted feature set $g(\mathcal{D})$ with the constraint $|\mathcal{M}| = m$.*

We give the proof in Appendix A. Theorem 1 characterizes the expected gap between the distance of a normal feature to the memory and that of an anomalous feature to the same memory. The positivity of $\Delta(m)$ implies that, on average, anomalous features lie farther from the memory than normal features. The key insight lies in the unimodality (*i.e.*, bell-shaped curve) of the weight function $\omega(m, r)$ with respect to $m$. Since $\Delta_0(m)$ approximates $\Delta(m)$ via Taylor

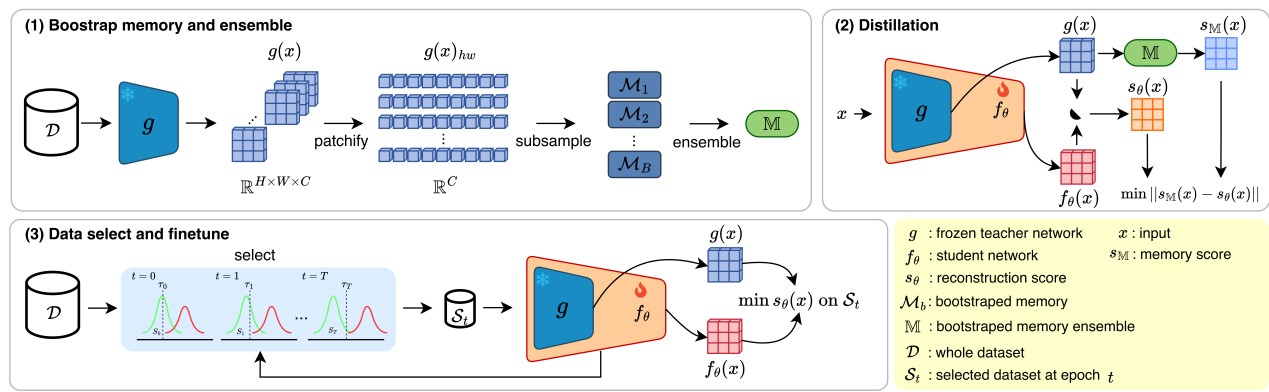

*Figure 1.* Overview of the three-stage MeDS pipeline. **First stage: memory construction.** From each training image $x \in \mathcal{D}$, a frozen teacher network $g$ extracts patch features $g(x)_{hw} \in \mathbb{R}^C$ for all spatial positions $(h, w)$. The pooled feature collection $g(\mathcal{D}) \subseteq \mathbb{R}^C$ of size $|g(\mathcal{D})| = |\mathcal{D}| \cdot H \cdot W$ is randomly subsampled $B$ times at ratio $\rho$ to form partial memory banks $\{\mathcal{M}_b\}_{b=1}^{B}$. Each memory bank $\mathcal{M}_b$ produces a per-patch nearest-neighbor distance score $s_{\mathcal{M}_b}(x) \in \mathbb{R}^{H \times W}$, and averaging these scores across all $B$ memory banks yields the ensemble score $s_{\mathbb{M}}(x) \in \mathbb{R}^{H \times W}$. The sparse subsampling acts as a low-pass filter that downweights anomalous patches. **Second stage: memory score distillation.** For further enhancement, the memory score $s_{\mathbb{M}}$ is distilled into a network-based reconstruction score $s_\theta$ via $\min_\theta \|s_{\mathbb{M}}(x) - s_\theta(x)\|_2$, where $s_\theta(x) \in \mathbb{R}^{H \times W}$ is the distance between teacher features $g(x)$ and the features $f_\theta(x) \in \mathbb{R}^{H \times W \times C}$ extracted from a student network $f_\theta$. **Third stage: fine-tuning with progressive selection.** Finally, the reconstruction score $s_\theta$ is fine-tuned on a clean subset $\mathcal{S}_t \subseteq \mathcal{D}$ obtained by a progressive selection criterion.

expansion, and $\Delta_0(m)$ is a weighted integral with the uni-modal weight $\omega(m, r)$, the gap $\Delta(m)$ tends to be larger at intermediate values of $m$. In other words, subsampling with an appropriately small memory size acts as a low-pass filter, amplifying the separation between normal and anomalous features. Ensembling in Eq. (5) reduces variance in estimating this expected gap, yielding more stable anomaly scores. This theoretical insight is empirically validated in Fig. 3, where the separation between normal and anomalous patches is maximized at a moderately small subsampling ratio.

### 4.2. Memory Score Distillation

While the bootstrapped memory ensemble provides robust coarse-level filtering, it exhibits an inherent limitation. The memory scores rely solely on features pretrained on external data, and hence, the discriminative gap between normal and anomalous features is bounded by what these frozen representations can capture; when the pretrained features do not sufficiently distinguish domain-specific anomaly patterns, the resulting separation remains fundamentally limited.

To address these limitations, we train a student network via score-space distillation. Unlike standard teacher-student distillation in AD that operates in feature space, *we distill knowledge in the score space*: the reconstruction score network is trained to minimize

$$\min_\theta \frac{1}{|\mathcal{D}|} \sum_{i=1}^{|\mathcal{D}|} \|s_{\mathbb{M}}(x_i) - s_\theta(x_i)\|_2, \qquad (9)$$

where $\|\cdot\|_2$ is the $L_2$ norm, and $s_\theta := D(g, f_\theta)$ is defined in

Eq. (3). In practice, the average is computed over the batch.

Since the reconstructor $s_\theta$ is trained from scratch (*i.e.*, initialized with random weights), it benefits from the early learning phenomenon of neural networks (Liu et al., 2020). Neural networks exhibit a simplicity bias, fitting dominant patterns before memorizing rare ones. In our setting, normal patches constitute the majority of the training data, so the student learns to reconstruct normal patterns earlier and more accurately than anomalous ones. This property allows the student to effectively denoise the memory scores: the reconstruction score $s_\theta(x)$ is lowered on normal patches earlier and more substantially than on anomalous patches, thereby improving separation. Moreover, the distillation process yields a well-initialized network that enables a smooth transition to the subsequent fine-tuning phase.

### 4.3. Fine-tuning with Progressive Selection

The distilled model benefits from early learning regularization, which improves robustness to noise during early training. However, this involves a fundamental trade-off: early stopping preserves image-level robustness but yields insufficient training for precise pixel-level localization; late stopping enables fine-grained learning but degrades performance due to noise memorization. To resolve this trade-off, we fine-tune the student network on self-selected clean data, allowing extended training without overfitting to anomalous samples.

We fine-tune the reconstruction score network $s_\theta$ on a pro-

gressively selected subset $\mathcal{S}_t$:

$$\min_\theta \frac{1}{|\mathcal{S}_t|} \sum_{x \in \mathcal{S}_t} s_\theta(x), \qquad (10)$$

where the selected data $\mathcal{S}_t$ at iteration $t$ is updated at the beginning of every epoch based on the selection criterion $\eta_t(x)$ and the threshold $\tau_t$. We define the selection criterion as a combination of the current reconstruction score and the initial distilled score:

$$\eta_t(x) = (1-\alpha_t)\max_{h,w} s_{\theta_0}(x)_{hw} + \alpha_t\max_{h,w} s_\theta(x)_{hw}, \ (11)$$

where $\max_{h,w} s_\theta(x)_{hw}$ denotes the maximum anomaly score at the patch level. Here, $\theta_0$ is the frozen parameters of the initial distilled model, fixed during fine-tuning. The selected data is given as $\mathcal{S}_t = \{x : \eta_t(x) < \tau_t\}$ based on the threshold

$$\tau_t = \text{Median}(\eta_t(x)) + k_t\text{MAD}(\eta_t(x)), \qquad (12)$$

where median and median absolute deviation (MAD) are computed over the whole noisy training data $\mathcal{D}$. The interpolation coefficient $\alpha_t = \min(1, 2t/T)$ and critical value $k_t = k(t/T)$ monotonically increase from 0 to 1 and from 0 to $k$, where $T$ is the total number of iterations. Unless specified otherwise, we set $k = 1$.

The median-based threshold assumes that normal samples constitute the majority of the training set, ensuring that at least half of the selected samples are normal. The MAD provides a robust measure of score dispersion that is less sensitive to outliers than the standard deviation. Initializing $k_t = 0$ yields a conservative threshold that selects only samples with scores below the median, and progressively increasing $k_t$ relaxes the threshold to safely incorporate extra normal samples as the model improves. This forms a positive feedback loop: an improved model gives more accurate selection scores, yielding a higher-quality training subset, which in turn further refines the model, enabling precise pixel-level localization without overfitting to anomalous samples.

### 4.4. Full Training Algorithm and Inference

MeDS trains the parametric reconstruction model $s_\theta$ through the three phases described above: constructing the bootstrapped memory ensemble $s_\mathbb{M}$, distilling its scores to initialize $s_\theta$, and fine-tuning on progressively selected data. The complete algorithm flow is described in Algorithm 1. At inference, the anomaly score map for a test sample $x$ is computed as $s_\theta(x)$ and resized to match the original image dimensions.

## 5. Experiments

### 5.1. Setup

**Datasets.** We conduct experiments on 3 widely used industrial AD datasets: MVTecAD (Bergmann et al., 2019), VisA (Zou et al., 2022), and Real-IAD (Wang et al., 2024). The noise ratios for MVTecAD are $10\%, 20\%, 40\%$, and $2\%, 5\%, 10\%$ for VisA. Setting the random seed to 0 in all cases for fairness, we uniformly sample anomalous images from the test set, stratified by class and defect type, and include them in the training set until the desired noise ratio is reached. The included anomaly samples also appear in the test set. The noise ratio of VisA is relatively lower than that of MVTecAD because the number of normal samples in the training set is significantly larger compared to the total number of anomalous samples. For Real-IAD, we follow its noisy configuration benchmarking protocol.

**Hyperparameter configuration.** For the bootstrapped memory, we subsample $B=100$ memories with subsampling ratio $\rho=0.1$. We give other configuration details in Appendix B.

**Evaluation Metrics.** We evaluate performance at both image and pixel levels. For image-level detection, we report the Area Under the Receiver Operating Characteristic curve (I-AUROC) and Average Precision (I-AP), which measure the model's ability to distinguish anomalous images from normal ones. For pixel-level localization, we report pixel-wise Average Precision (P-AP) and the Area Under the Per-Region Overlap curve (P-AUPRO) (Bergmann et al., 2019), which evaluate segmentation quality while weighting anomaly regions equally regardless of their size. All metrics are reported as percentages.

### 5.2. Result Comparison

**MVTecAD and VisA.** We evaluate MeDS on MVTecAD and VisA under the multi-class setting, where a single model is trained across all product categories. As shown in Tab. 1 and Tab. 2, existing noise-robust methods exhibit inherent trade-offs across contamination levels. SoftPatch assumes that anomalous patches form sparse clusters while normal features remain dense; this assumption holds at low noise ratios but breaks down as contamination increases. Conversely, FUN-AD implicitly assumes a non-negligible presence of anomalies, causing it to discard informative normal samples on clean data. In contrast, MeDS maintains stable performance across all noise ratios without such assumptions, consistently outperforming baselines in both image-level and pixel-level metrics. When applied as a framework to baseline methods (HVQ (Lu et al., 2023), Dinomaly (Guo et al., 2025), INP-Former (Luo et al., 2025)), MeDS incurs only marginal degradation on clean data while substantially improving robustness under contamination.

*Table 1.* Results on MVTecAD where underline highlights the best noisy AD baseline performance and **bold emphasizes the better performance between baseline and MeDS.**

| Metric | I-AUROC (↑) | | | | I-AP (↑) | | | | P-AUPRO (↑) | | | | P-AP (↑) | | | |
|---|---|---|---|---|---|---|---|---|---|---|---|---|---|---|---|---|
| Noise Ratio | 0 | 10 | 20 | 40 | 0 | 10 | 20 | 40 | 0 | 10 | 20 | 40 | 0 | 10 | 20 | 40 |
| SoftPatch | 98.80 | 98.10 | 96.89 | 93.58 | 99.60 | 99.30 | 98.87 | 97.66 | 92.80 | 86.40 | 77.75 | 69.33 | 66.30 | 57.70 | 49.38 | 35.25 |
| InReach | 92.00 | 87.41 | 78.80 | 73.57 | 97.06 | 95.14 | 91.78 | 89.34 | 86.14 | 81.81 | 75.28 | 72.06 | 52.87 | 49.65 | 45.04 | 39.67 |
| FUN-AD | 81.89 | 95.49 | 96.06 | 97.70 | 89.18 | 97.73 | 98.04 | 98.90 | 61.49 | 78.39 | 78.21 | 73.91 | 42.94 | 58.52 | 58.86 | 61.38 |
| HVQ | 96.71 | 91.08 | 91.49 | 92.14 | **98.83** | 96.53 | 96.69 | 96.69 | **91.31** | 87.76 | 88.41 | 88.69 | **47.71** | 42.47 | 42.67 | 41.88 |
| HVQ + MeDS (ours) | 95.89 | **94.95** | **94.76** | **94.26** | 98.53 | **98.09** | **98.01** | **97.80** | 91.18 | **90.40** | **90.19** | **90.13** | 47.42 | **46.13** | **44.65** | **44.82** |
| Dinomaly | 99.64 | 95.19 | 92.16 | 87.38 | **99.80** | 97.34 | 95.50 | 93.28 | 94.62 | 91.04 | 90.21 | 89.26 | 68.19 | 58.22 | 54.60 | 53.00 |
| Dinomaly + MeDS (ours) | 99.37 | **99.49** | **99.31** | **99.16** | 99.72 | **99.76** | **99.71** | **99.63** | **94.74** | **94.69** | **94.59** | **94.54** | **68.35** | **68.96** | **68.63** | **68.05** |
| INP-Former | 99.66 | 95.13 | 91.21 | 85.85 | **99.88** | 97.34 | 94.31 | 91.14 | 94.88 | 91.06 | 89.64 | 88.85 | **70.55** | 59.88 | 54.39 | 51.26 |
| INP-Former + MeDS (ours) | 99.45 | **99.39** | **99.41** | **99.17** | 99.78 | **99.79** | **99.78** | **99.68** | **95.13** | **95.25** | **95.22** | **95.21** | 67.15 | **67.79** | **67.51** | **67.39** |

*Table 2.* Results on VisA where underline highlights the best noisy AD baseline performance and **bold emphasizes the better performance between baseline and MeDS.**

| Metric | I-AUROC (↑) | | | | I-AP (↑) | | | | P-AUPRO (↑) | | | | P-AP (↑) | | | |
|---|---|---|---|---|---|---|---|---|---|---|---|---|---|---|---|---|
| Noise Ratio | 0 | 2 | 5 | 10 | 0 | 2 | 5 | 10 | 0 | 2 | 5 | 10 | 0 | 2 | 5 | 10 |
| SoftPatch | 93.85 | 93.37 | 92.47 | 89.91 | 94.88 | 94.74 | 94.12 | 92.86 | 88.39 | 86.77 | 82.88 | 75.09 | 47.50 | 45.56 | 44.73 | 37.03 |
| InReach | 83.99 | 79.34 | 73.40 | 64.15 | 86.82 | 84.09 | 80.64 | 74.06 | 78.80 | 73.78 | 65.45 | 51.30 | 31.37 | 29.54 | 27.76 | 24.15 |
| FUN-AD | 82.57 | 90.69 | 92.27 | 94.79 | 82.71 | 90.31 | 92.40 | 95.50 | 51.22 | 60.93 | 64.59 | 66.48 | 29.36 | 37.04 | 45.41 | 46.55 |
| HVQ | 88.88 | 87.92 | 87.12 | 86.10 | **91.02** | 90.02 | 89.34 | 88.33 | **84.33** | 83.83 | 84.51 | 84.00 | **34.17** | **31.81** | 31.58 | **33.24** |
| HVQ + MeDS (ours) | 88.36 | **88.26** | **87.32** | **87.19** | 90.25 | **90.31** | **89.63** | **89.74** | 83.19 | 83.29 | 83.27 | 83.40 | 30.26 | 31.00 | 31.38 | 31.99 |
| Dinomaly | 98.47 | 97.35 | 96.06 | 93.56 | **98.63** | 97.67 | 96.65 | 94.06 | 94.38 | 93.93 | 93.63 | 92.59 | **52.80** | 49.81 | 46.94 | 46.70 |
| Dinomaly + MeDS (ours) | 97.53 | 97.27 | **97.51** | **97.43** | 96.99 | 96.87 | **97.12** | **97.01** | **94.39** | **94.06** | **94.10** | **94.39** | 51.38 | **50.93** | **51.27** | **51.46** |
| INP-Former | 98.15 | 96.78 | 95.30 | 94.45 | **98.37** | 96.96 | 95.58 | 94.52 | **94.70** | 93.96 | 93.24 | 93.20 | 47.60 | 42.58 | 39.89 | 42.21 |
| INP-Former + MeDS (ours) | 98.02 | **97.03** | **97.04** | **96.54** | 96.66 | 96.78 | **96.53** | **96.30** | 94.19 | **94.26** | **94.24** | 93.96 | **49.58** | **43.12** | **43.51** | **42.70** |

**Real-IAD.** We evaluate on Real-IAD following the official noisy-data protocol under the single-class setting, where a separate model is trained for each product category. This configuration is necessary because certain baselines, such as SoftPatch, cannot scale to the large number of categories in Real-IAD under a multi-class scenario. As shown in Tab. 3, SoftPatch achieves competitive image-level performance but underperforms in pixel-level metrics, highlighting its limitation in fine-grained localization. MeDS achieves the best performance across all noise scenarios in both the image and pixel levels.

### 5.3. Ablation

We conduct an ablation study to evaluate the contribution of each component with Dinomaly as our baseline (row 1); results are reported in Tab. 4. The bootstrapped memory ensemble (row 2) improves image-level performance under noisy conditions but shows limited pixel-level localization capability. Memory distillation (row 3) addresses this limitation by transferring the coarse-level robustness of the memory ensemble while refining fine-grained localization through gradient-based learning, yielding consistent gains across all metrics. Fine-tuning with progressive selection further improves both image-level and pixel-level performance, with particularly pronounced gains in P-AP, demon-

strating its effectiveness in enabling precise segmentation without overfitting to anomalous samples. We additionally analyze the fine-tuning phase by isolating the effects of initialization and the selection criterion (rows 4-6): using the distilled model for both initialization and selection outperforms alternatives using the memory ensemble without distillation or random initialization, confirming that both components are essential.

### 5.4. Analysis

We conduct detailed analyses of the hyperparameters of MeDS by experimenting with the Dinomaly baseline on MVTecAD.

**Bootstrapped memory ensemble.** We use the ensemble size $B = 100$ and subsampling ratio $\rho = 0.1$ by default. The results in Fig. 2 indicate that the bootstrapped memory (left) prefers a lower sampling ratio for both image-level and pixel-level performance, but its test set performance is overall poor. In contrast, the final model (*i.e.*, MeDS) maximizes performance at a moderately low subsampling ratio. This aligns with the trend shown in Fig. 3, which shows that the normal and anomalous samples in the training set are maximally separated at a moderate subsampling ratio. Fig. 4, on the other hand, shows that model performance

*Table 3.* Results on Real-IAD where **bold highlights best performance**

| | Real-IAD | | | | | | | | | | | | | | | |
|---|---|---|---|---|---|---|---|---|---|---|---|---|---|---|---|---|
| Metric | **I-AUROC (↑)** | | | | **I-AP (↑)** | | | | **P-AUPRO (↑)** | | | | **P-AP (↑)** | | | |
| Noise Ratio | 0 | 10 | 20 | 40 | 0 | 10 | 20 | 40 | 0 | 10 | 20 | 40 | 0 | 10 | 20 | 40 |
| PatchCore | 91.30 | 90.40 | 89.50 | 88.10 | - | - | - | - | 92.60 | 93.20 | 93.00 | 92.40 | - | - | - | - |
| RD | 88.10 | 88.10 | 87.30 | 84.50 | - | - | - | - | 95.10 | 95.10 | 94.90 | 94.70 | - | - | - | - |
| UniAD | 85.40 | 84.20 | 82.80 | 80.10 | - | - | - | - | 87.60 | 87.70 | 87.30 | 86.60 | - | - | - | - |
| SoftPatch | 91.68 | 91.03 | 90.33 | 88.43 | 86.21 | 85.92 | 84.61 | 82.52 | 91.30 | 91.72 | 91.68 | 90.96 | 35.78 | 33.39 | 31.33 | 28.16 |
| Dinomaly | **92.39** | 91.38 | 89.97 | 87.78 | 83.14 | 84.69 | 82.22 | 78.67 | **95.97** | 96.16 | 95.88 | 95.47 | 40.04 | **40.58** | 38.67 | 35.41 |
| Dinomaly + MeDS (ours) | 92.34 | **92.36** | 92.08 | 90.99 | 84.76 | 84.89 | 85.11 | 83.98 | 95.53 | 95.60 | 95.54 | 95.67 | 40.41 | 40.43 | 41.15 | 41.22 |

*Table 4.* Ablation on MVTecAD dataset and Dinomaly model where **bold highlights best performance**. 'Memory' indicates usage of boostrapped memory ensemble, 'init' shows how the final model is initialized, and 'criteria' denotes the model $s_{\theta_0}$ used for data selection.

| Ablation Setting | | | | | | **I-AUROC (↑)** | | | | **I-AP (↑)** | | | | **P-AUPRO (↑)** | | | | **P-AP (↑)** | | | |
|---|---|---|---|---|---|---|---|---|---|---|---|---|---|---|---|---|---|---|---|---|---|
| | | | | | | Noise Ratio | | | | | | | | | | | | | | | |
| Memory | Distill | Fine-tune | Init | Criteria | Description | 0 | 10 | 20 | 40 | 0 | 10 | 20 | 40 | 0 | 10 | 20 | 40 | 0 | 10 | 20 | 40 |
| ✗ | ✗ | ✗ | ✗ | ✗ | Baseline | **99.64** | 96.38 | 94.16 | 90.45 | **99.80** | 97.92 | 96.46 | 94.84 | 94.62 | 91.61 | 90.81 | 90.04 | 68.19 | 59.50 | 56.19 | 54.75 |
| ✓ | ✗ | ✗ | ✗ | ✗ | Memory ensemble | 98.57 | 97.47 | 94.74 | 92.22 | 99.41 | 98.75 | 97.02 | 96.00 | 91.99 | 91.81 | 91.86 | 90.50 | 62.13 | 62.22 | 56.05 | 53.04 |
| ✓ | ✓ | ✗ | ✗ | ✗ | Distilled model | 99.08 | 99.26 | 99.09 | 98.76 | 99.45 | 99.69 | 99.56 | 99.42 | 94.39 | 92.78 | 93.21 | 91.02 | 66.15 | 62.98 | 63.54 | 56.88 |
| ✓ | ✗ | ✓ | Random | Memory | Fine-tuned with memory | 98.93 | 99.17 | 99.10 | 98.94 | 99.50 | 99.63 | 99.60 | 99.56 | **94.80** | **94.75** | 94.56 | **94.58** | 67.84 | 68.48 | 67.80 | 67.71 |
| ✓ | ✓ | ✓ | Random | Distilled | Random init fine-tune | 98.70 | 99.16 | 99.23 | 98.87 | 99.30 | 99.62 | 99.63 | 99.51 | 94.59 | 94.67 | **94.71** | 94.54 | 67.06 | 67.92 | 68.00 | 67.91 |
| ✓ | ✓ | ✓ | Distilled | Distilled | MeDS (ours) | 99.37 | **99.49** | **99.31** | **99.16** | 99.72 | **99.76** | **99.71** | **99.63** | 94.74 | 94.69 | 94.59 | 94.54 | **68.35** | **68.96** | **68.63** | **68.05** |

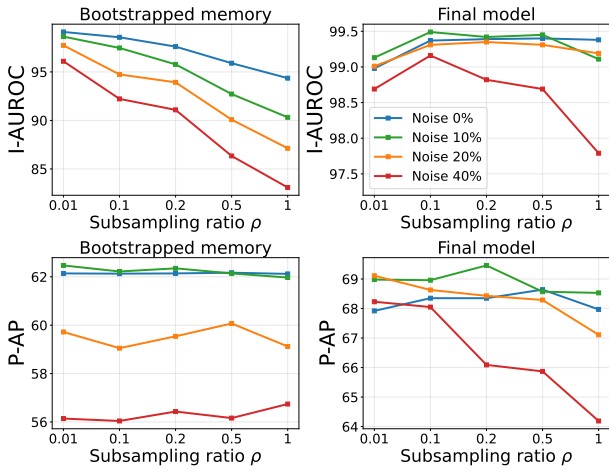

*Figure 2.* Effect of varying the subsampling ratio on the bootstrapped memory ensemble (left) and the resulting effect on the final model (right).

increases with the ensemble size.

**Train iterations of memory distillation.** We examine how the number of total iterations $T$ for memory distillation affects performance, validating the trade-off. As shown in Fig. 5, image-level performance (I-AUROC) of the distilled model improves as $T$ increases but plateaus after approximately $T = 500$, while pixel-level performance (P-AP) initially increases but degrades substantially beyond this point. This confirms our claim that extended training enables fine-grained learning but eventually leads to overfitting to anomalous samples, whereas early stopping preserves image-level robustness at the cost of pixel-level precision. Crucially, without a labeled validation set, identifying the optimal stopping point is infeasible in practice. Our progres-

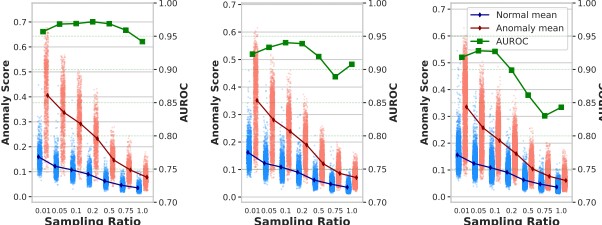

*Figure 3.* Patch-level anomaly score histogram (blue and red scatter plots) with the mean (blue and red lines) and AUROC (green line) of the memory ensemble on the training set for each subsampling ratio, at noise ratios 10%/20%/40%, respectively.

sive data selection strategy resolves this dilemma: by fine-tuning on self-filtered clean data, the final model achieves both high I-AUROC and P-AP that remain stable across a wide range of distillation iterations, effectively decoupling the final performance from the precise choice of distillation duration.

**Critical value.** We ablate the critical value $k$ used in the criterion for data selection. As shown in Fig. 6, the performance gap between different critical values is not significant, and the results indicate overall robustness to this hyperparameter.

## 5.5. Active Label Correction

We evaluate the utility of MeDS for active label correction, a human-in-the-loop setting where an annotator reviews training samples ranked by their anomaly scores to identify and remove contaminated data. By inspecting samples in descending order of the selection criterion, an effective ranking allows the annotator to encounter all contaminated samples

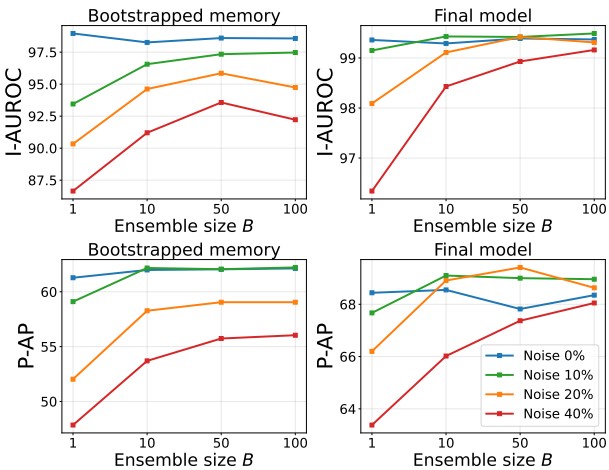

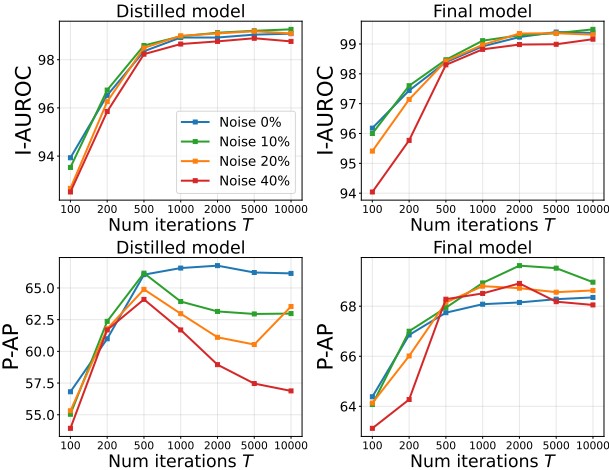

*Figure 4.* Effect of ensemble size on memory ensemble performance (left) and the resulting effect on the final model (right).

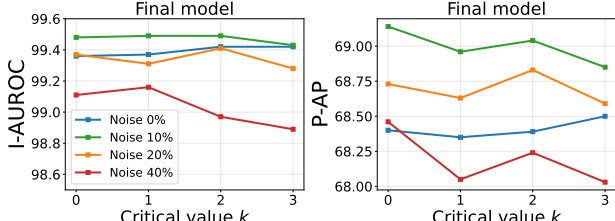

*Figure 5.* Analysis of total iterations of memory distillation. The distilled model is trained for the number of iterations indicated on the x-axis. The final model is fine-tuned with data selection with $10,000$ total iterations.

early, minimizing inspection effort. We compare selection scores from vanilla Dinomaly against those from MeDS on MVTecAD under 10%, 20%, and 40% noise ratios, using two metrics: (1) AUPRC (Area Under the Precision-Recall Curve), which measures the overall ranking quality where higher values indicate contaminated samples are consistently ranked above clean ones; and (2) inspection depth, defined as the minimum fraction of the training set that must be reviewed to identify all contaminated samples, where lower values correspond to reduced annotation cost.

As shown in Fig. 7, applying MeDS significantly improves both metrics across all noise ratios. The consistent gains in AUPRC demonstrate that MeDS produces more reliable rankings, while the substantial reductions in inspection depth translate directly to annotation savings—enabling

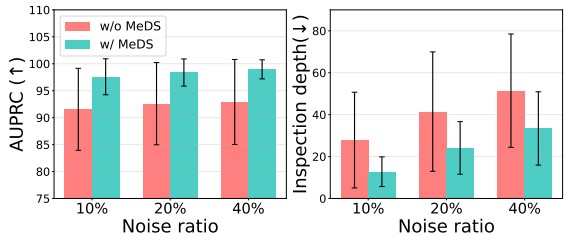

*Figure 6.* Analysis of critical value used for data selection on the final model.

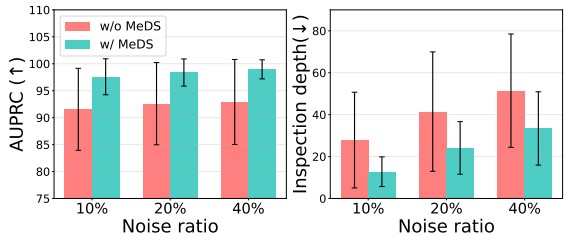

*Figure 7.* Active label correction results, where bars represent class averages and lines indicate standard deviations.

practitioners to identify all contaminated samples while reviewing far fewer images. These results highlight that MeDS serves not only as a robust automated detector but also as a practical tool for efficient dataset curation in industrial settings.

## 6. Limitations

While MeDS demonstrates strong performance across a wide range of noise ratios, several limitations warrant discussion. First, MeDS occasionally exhibits marginally lower performance compared to the baseline methods on fully clean datasets. This is likely attributable to the median-based selection mechanism, which may inadvertently exclude a small fraction of informative normal samples even in the absence of contamination. Second, MeDS incurs additional computational overhead relative to standard baseline training. MeDS requires dedicated stages for memory construction and distillation, and the progressive selection mechanism necessitates a full inference pass over the training data. While this additional cost is justified under contaminated scenarios—where baseline methods fail catastrophically—it may be unnecessary when the training data is known to be clean.

## 7. Conclusion

We presented Memory-Distilled Selection (MeDS), a training framework that enables robust anomaly detection under data contamination without requiring knowledge of the noise ratio or noise-ratio-specific hyperparameter tuning. The key insight underlying MeDS is that sparse subsam-

pling of memory banks acts as a low-pass filter that amplifies the separation between normal and anomalous features—a phenomenon we formalize theoretically and validate empirically. By distilling these ensemble-derived scores into a reconstruction score network, MeDS exploits the early-learning bias of neural networks to further sharpen the normal–anomaly boundary while providing a well-initialized starting point for subsequent fine-tuning. The progressive self-selection mechanism then enables extended training on self-filtered clean data, achieving precise pixel-level localization without overfitting to noisy samples. On MVTecAD, VisA, and Real-IAD, we show that MeDS yields consistent improvements across a wide range of noise ratios. Beyond automated detection, we demonstrated that MeDS serves as a practical tool for active label correction, enabling efficient identification and removal of contaminated samples with minimal human effort.

## Acknowledgements

Jaewoo Park and Seongdeok Bang were supported by the Technology Innovation Program (RS-2025-25455526, Development of an AI-Based Autonomous Control System for Multi-Process Integration in Self-Assembly of Multi-Variant Automotive Lighting Modules) funded By the Ministry of Trade, Industry and Resources(MOTIR, Korea). Kuan-Chuan Peng was exclusively supported by Mitsubishi Electric Research Laboratories.

## Impact Statement

This paper presents work whose goal is to advance the field of machine learning. There are many potential societal consequences of our work, none of which we feel must be specifically highlighted here.

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

# A. Theory

**Definition** (Spatial Proportion). Let $g(\mathcal{D}) \subseteq \mathcal{Z}$ be a fixed training feature set of size $N$. For any query $q$ and radius $r$, let $\pi(q, r)$ denote the cumulative proportion of the training set falling within distance $r$ of $q$:

$$\pi(q, r) = \frac{1}{N} \sum_{z \in g(\mathcal{D})} \mathbb{1}(\|z - q\| \leq r)$$

where $\mathbb{1}$ is the indicator function.

**Definition** (Density Advantage). Let $\pi_{norm}(r) = \pi(q_{norm}, r)$ and $\pi_{anom}(r) = \pi(q_{anom}, r)$. We define the Signal $\delta(r)$ as the difference in spatial accumulation between the normal and anomalous queries:

$$\delta(r) = \pi_{norm}(r) - \pi_{anom}(r)$$

**Definition** (Expected Gap). Let $D(q, \mathcal{M})$ be the random variable representing the distance from a query $q$ to its nearest neighbor in a memory bank $\mathcal{M}$ of size $m$, sampled independently with replacement from $g(\mathcal{D})$. The expected distance is defined as the integral of the survival function:

$$\mathbb{E}[D(q, \mathcal{M})] = \int_0^\infty \mathbb{P}(D(q, \mathcal{M}) > r) \, \mathrm{d}r = \int_0^\infty (1 - \pi(q, r))^m \, \mathrm{d}r$$

Let $\Delta(m)$ be the difference in expected nearest neighbor distance between the anomaly and normal queries:

$$\Delta(m) = \mathbb{E}[D(q_{anom}, \mathcal{M})] - \mathbb{E}[D(q_{norm}, \mathcal{M})]$$

**Assumption** (Strict Separability). We assume the normal query dominates the anomaly query in neighbor density. That is, $\delta(r) \geq 0$ for all $r$. Moreover, there exist $0 \leq r_0 < r_1$ such that, for every $r \in [r_0, r_1]$, $\delta(r) > 0$ and $\pi_{norm}(r) < 1$.

**Theorem 1** (Gap Decomposition). *Under the Strict Separability assumption, the expected gap satisfies $\Delta(m) > 0$ and can be decomposed as*

$$\Delta(m) = \Delta_0(m) + \epsilon_0(m)$$

*where*

$$\Delta_0(m) = \int_0^\infty \delta(r) \cdot \omega(m, r) \mathrm{d}r$$

*with $\omega(m, r) := m(1 - \pi_{norm}(r))^{m-1}$, and $\epsilon_0(m) = \int R(r; m) \mathrm{d}r$ with*

$$R(r; m) = o(\delta(r)) \quad \text{as } \delta(r) \to 0$$

*Moreover, for each fixed $r$ with $\pi_{norm}(r) \in (0, 1)$, $\omega(m, r)$ is unimodal in $m$, attaining its unique maximum at $m^* = -1/\ln(1 - \pi_{norm}(r))$.*

To prove the main theorem, we first establish the below:

**Lemma 1** (Probability Ordering). *Under Assumption 1, the probability that the nearest neighbor distance exceeds $r$ satisfies the following inequalities:*

- *For all $r \geq 0$:*
$$\mathbb{P}(D(q_{norm}, \mathcal{M}) > r) \leq \mathbb{P}(D(q_{anom}, \mathcal{M}) > r)$$

- *For every $r \in [r_0, r_1]$:*
$$\mathbb{P}(D(q_{norm}, \mathcal{M}) > r) < \mathbb{P}(D(q_{anom}, \mathcal{M}) > r)$$

*Consequently, $\mathbb{E}[D(q_{norm}, \mathcal{M})] < \mathbb{E}[D(q_{anom}, \mathcal{M})]$.*

*Proof.* **Step 1: Probability formulation.** The event $D(q, \mathcal{M}) > r$ implies that all $m$ independent samples drawn from $g(\mathcal{D})$ fall outside the ball $B(q, r)$. Let $C(q, r) = N \cdot \pi(q, r)$ be the count of neighbors within radius $r$. The probability

that a single random sample falls outside this radius is $1 - \frac{C(q,r)}{N} = 1 - \pi(q,r)$. Since the $m$ memory bank samples are independent (sampled with replacement), the joint probability is:

$$\mathbb{P}(D(q, \mathcal{M}) > r) = (1 - \pi(q, r))^m$$

**Step 2: Monotonicity analysis.** Define the function $g(u) = (1 - u)^m$ for $u \in [0, 1]$. We analyze its monotonicity by differentiating with respect to the proportion $u$:

$$g'(u) = -m(1 - u)^{m-1}$$

For any $u < 1$ and $m \geq 1$, the derivative is strictly negative ($g'(u) < 0$). Thus, $g(u)$ is a strictly monotonically decreasing function of $u$. A higher proportion of neighbors $\pi(q, r)$ strictly reduces the probability of the distance exceeding $r$.

**Step 3: Comparison.** We apply the Strict Separability Assumption to compare the probabilities:

- **Case 1 (all $r \geq 0$):** By assumption, $\delta(r) \geq 0 \implies \pi_{norm}(r) \geq \pi_{anom}(r)$. Since $g(u)$ is decreasing, $g(\pi_{norm}) \leq g(\pi_{anom})$. Thus, $\mathbb{P}_{norm} \leq \mathbb{P}_{anom}$.

- **Case 2 ($r \in [r_0, r_1]$):** By assumption, $\delta(r) > 0$ and $\pi_{norm}(r) < 1$, so $\pi_{norm}(r) > \pi_{anom}(r)$ with both arguments in the region where $g$ is strictly decreasing. Hence $g(\pi_{norm}) < g(\pi_{anom})$, and $\mathbb{P}_{norm} < \mathbb{P}_{anom}$.

The expected distance is the integral of these tail probabilities over $r \in [0, \infty)$. Since the difference $(\mathbb{P}_{anom} - \mathbb{P}_{norm})$ is non-negative everywhere and strictly positive on the interval $[r_0, r_1]$, the integral for the normal query is strictly smaller. $\square$

**Lemma 2** (Local Linearization). *Let $\psi(r; m)$ denote the gap density at radius $r$:*

$$\psi(r; m) = (1 - \pi_{anom}(r))^m - (1 - \pi_{norm}(r))^m$$

*For any fixed $m$ and $r$, this quantity expands as:*

$$\psi(r; m) = \delta(r) \cdot \omega(m, r) + R(r; m)$$

*where the leading weight is $\omega(m, r) = m(1 - \pi_{norm}(r))^{m-1}$. The remainder term $R(r; m)$ is given by the Lagrange form:*

$$R(r; m) = \frac{m(m - 1)}{2}(1 - \xi)^{m-2}(\delta(r))^2$$

*for some $\xi \in (\pi_{anom}(r), \pi_{norm}(r))$. Consequently, $R(r; m) \geq 0$ (for $m \geq 1$) and $R(r; m) = o(\delta(r))$ as $\delta(r) \to 0$.*

*Proof.* Consider the function $f(u) = (1 - u)^m$ defined on $u \in [0, 1]$. We seek to approximate the difference $f(\pi_{anom}) - f(\pi_{norm})$. Recalling that $\pi_{anom}(r) = \pi_{norm}(r) - \delta(r)$, we apply Taylor's theorem with the Lagrange remainder to expand $f(u)$ around the point $u_0 = \pi_{norm}(r)$ with perturbation $h = -\delta(r)$. There exists some $\xi$ strictly between $\pi_{anom}(r)$ and $\pi_{norm}(r)$ such that:

$$f(u_0 + h) = f(u_0) + f'(u_0)h + \frac{f''(\xi)}{2!}h^2$$

Computing the derivatives with respect to $u$:

$$f'(u) = -m(1 - u)^{m-1}$$
$$f''(u) = m(m - 1)(1 - u)^{m-2}$$

Substituting $h = -\delta(r)$ and $u_0 = \pi_{norm}(r)$:

$$(1 - \pi_{anom})^m = (1 - \pi_{norm})^m + \left[-m(1 - \pi_{norm})^{m-1}\right](-\delta(r)) + \frac{1}{2}\left[m(m - 1)(1 - \xi)^{m-2}\right](-\delta(r))^2$$

Rearranging terms to isolate the gap density $\psi(r; m)$:

$$(1 - \pi_{anom})^m - (1 - \pi_{norm})^m = \delta(r)\underbrace{\left[m(1 - \pi_{norm})^{m-1}\right]}_{\omega(m,r)} + \underbrace{\frac{m(m - 1)}{2}(1 - \xi)^{m-2}(\delta(r))^2}_{R(r;m)}$$

Since $m \geq 1$ and $(1 - \xi) \geq 0$, the remainder $R(r; m)$ is non-negative. Since $R(r; m) = O((\delta(r))^2)$, we have $R(r; m) = o(\delta(r))$ as $\delta(r) \to 0$. $\square$

**Lemma 3** (Unimodality of $\omega$). *For any fixed $r$ with $\pi_{norm}(r) \in (0,1)$, the weight function $\omega(m,r) = m(1-\pi_{norm}(r))^{m-1}$ is unimodal in $m > 0$, attaining its unique maximum at $m^* = -1/\ln(1-\pi_{norm}(r))$.*

*Proof.* Let $p = 1 - \pi_{norm}(r) \in (0,1)$. Treating $m$ as a continuous variable, we differentiate $\omega(m) = mp^{m-1}$:

$$\frac{\partial \omega}{\partial m} = p^{m-1}(1 + m \ln p)$$

Since $p^{m-1} > 0$ and $\ln p < 0$, the derivative vanishes at $m^* = -1/\ln p > 0$, is positive for $m < m^*$, and negative for $m > m^*$. Thus $\omega$ is strictly increasing then strictly decreasing, i.e., unimodal. $\square$

*Proof of Theorem.* The expected gap $\Delta(m)$ is defined as the integral of the difference in survival probabilities over all radii:

$$\Delta(m) = \int_0^\infty \left[(1 - \pi_{anom}(r))^m - (1 - \pi_{norm}(r))^m\right] \mathrm{d}r$$

Using the notation from Lemma 2, the integrand is exactly $\psi(r; m)$. We apply the Local Linearization from Lemma 2 to the integrand:

$$\Delta(m) = \int_0^\infty \left(\delta(r) \cdot \omega(m,r) + R(r;m)\right) \mathrm{d}r$$

By the linearity of the integral, we can decompose this into two distinct components:

$$\Delta(m) = \underbrace{\int_0^\infty \delta(r) \cdot \omega(m,r) \, \mathrm{d}r}_{\Delta_0(m)} + \underbrace{\int_0^\infty R(r;m) \, \mathrm{d}r}_{\epsilon_0(m)}$$

This establishes the decomposition $\Delta(m) = \Delta_0(m) + \epsilon_0(m)$.

To prove $\Delta(m) > 0$, we rely on the Strict Separability Assumption. We are given that $\delta(r) \geq 0$ for all $r$, and that $\delta(r) > 0$ with $\pi_{norm}(r) < 1$ for every $r \in [r_0, r_1]$.

1. Since $\delta(r) \geq 0$ and the weight function $\omega(m,r) \geq 0$, the integrand of $\Delta_0(m)$ is non-negative everywhere. Moreover, on $[r_0, r_1]$, we have $\delta(r) > 0$ and $\omega(m,r) > 0$, so the integrand is strictly positive on an interval of positive length. Thus, $\Delta_0(m) > 0$.

2. By Lemma 2, the remainder $R(r;m) \geq 0$ due to convexity. Thus, $\epsilon_0(m) \geq 0$.

Consequently, $\Delta(m) = \Delta_0(m) + \epsilon_0(m) > 0$. The unimodality of $\omega(m,r)$ follows from Lemma 3. $\square$

# B. Full Training Specification

## B.1. Full Training Algorithm

Algorithm 1 provides the algorithmic flow of MeDS training.

## B.2. Implementation Details and Additional Training Configurations

**Memory Construction** Memory scores are computed once per sample and cached for reuse during distillation to reduce computational overhead. Subsampling is performed at the image level rather than at the patch feature level to preserve structural bias within each image. We assume that class labels (*i.e.*, product type) are available throughout training. This assumption is reasonable in practical industrial settings, where each product is typically associated with a barcode or identifier that encodes product type information. When constructing memory scores, images from different classes are processed independently to prevent cross-class interference in the feature space.

**Memory Distillation and Fine-tuning** For simplicity and to ensure fair comparison, we use identical hyperparameter configurations for both memory distillation and fine-tuning with data selection, including learning rate, batch size, and optimizer settings. The two stages differ only in their objective functions: memory distillation minimizes the score-based distillation loss (Eq. (9)), whereas fine-tuning optimizes reconstruction on the progressively selected subset. This unified configuration simplifies the training pipeline and reduces the need for stage-specific hyperparameter tuning.

---

**Algorithm 1** MeDS Training Algorithm

---

**Input:** Training set $\mathcal{D}$, pretrained encoder $g$, reconstruction score network $s_\theta$, ensemble size $B$, subsampling ratio $\rho$, total iterations $T$, final critical value $k$

*// Phase 1: Bootstrapped Memory Construction*

Extract patch features $g(\mathcal{D}) \leftarrow \{g(x)_{hw} \mid x \in \mathcal{D}, (h, w) \in [H] \times [W]\}$

**for** $b = 1$ **to** $B$ **do**

    $\mathcal{M}_b \leftarrow$ uniformly sample $\rho|g(\mathcal{D})|$ features from $g(\mathcal{D})$

**end for**

Define $s_{\mathbb{M}}(x) \leftarrow \frac{1}{B} \sum_{b=1}^{B} s_{\mathcal{M}_b}(x)$

*// Phase 2: Memory Distillation*

Initialize $\theta$ of $s_\theta$ with random weights

**for** $t = 1$ **to** $T$ **do**

    Sample mini-batch $\mathcal{B}$ from $\mathcal{D}$

    $\mathcal{L} \leftarrow \frac{1}{|\mathcal{B}|} \sum_{x \in \mathcal{B}} \|s_{\mathbb{M}}(x) - s_\theta(x)\|_2$

    Update $\theta$ by gradient descent on $\mathcal{L}$

**end for**

Store distilled parameters $\theta_0 \leftarrow \theta$

*// Phase 3: Fine-tuning with Progressive Selection*

Initialize selected set $\mathcal{S}_0 \leftarrow \mathcal{D}$

**for** $t = 1$ **to** $T$ **do**

    **if** $t$ corresponds to an epoch boundary **then**

        $\alpha_t \leftarrow \min(2t/T, 1); \quad k_t \leftarrow k \cdot t/T$

        **for** $x \in \mathcal{D}$ **do**

            $\eta_t(x) \leftarrow (1 - \alpha_t) \max_{h,w} s_{\theta_0}(x)_{hw} + \alpha_t \max_{h,w} s_\theta(x)_{hw}$

        **end for**

        $\tau_t \leftarrow \text{Median}(\{\eta_t(x)\}_{x \in \mathcal{D}}) + k_t \cdot \text{MAD}(\{\eta_t(x)\}_{x \in \mathcal{D}})$

        $\mathcal{S}_t \leftarrow \{x \in \mathcal{D} : \eta_t(x) < \tau_t\}$

    **end if**

    Sample mini-batch $\mathcal{B}$ from $\mathcal{S}_t$

    $\mathcal{L} \leftarrow \frac{1}{|\mathcal{B}|} \sum_{x \in \mathcal{B}} s_\theta(x)$

    Update $\theta$ by gradient descent on $\mathcal{L}$

**end for**

**Output:** Trained reconstruction score network $s_\theta$

---

**Data Selection** Data selection is performed at the beginning of each epoch (*i.e.*, after a complete pass over the current selected subset). Rather than using the raw maximum patch score, which can be sensitive to outliers, we compute a robust maximum by averaging the top $n\%$ of patch scores for each image. Specifically, let $\{s_i\}_{i=1}^{HW}$ denote the patch-level scores of image $x$ sorted in descending order. The robust maximum is defined as:

$$\max_{h,w} s_\theta(x)_{hw} = \frac{1}{\lceil n \cdot HW/100 \rceil} \sum_{i=1}^{\lceil n \cdot HW/100 \rceil} s_i \tag{13}$$

where $\lceil \cdot \rceil$ is the ceil operation. We set $n = 1$ throughout our experiments; an analysis of this design choice is provided in Sec. C.2. To improve efficiency during fine-tuning, the initial selection scores $\hat{s}_{\theta_0}(x)$ from the distilled model are precomputed and cached, thereby avoiding redundant forward passes.

The interpolation coefficient follows the schedule $\alpha_t = \min(2t/T, 1)$, ensuring that the selection criterion transitions smoothly from reliance on the distilled model to full reliance on the fine-tuned model by the midpoint of training. The threshold critical value increases linearly as $k_t = k \cdot (t/T)$, where $k$ is a pre-specified hyperparameter that controls the final selection stringency. Leveraging the class information associated with each image, data selection is performed independently per class; specifically, the median and MAD used for threshold computation are calculated within each class separately. This class-wise approach addresses the issue of varying anomaly score ranges across different product categories, ensuring consistent selection behavior regardless of class-specific score distributions.

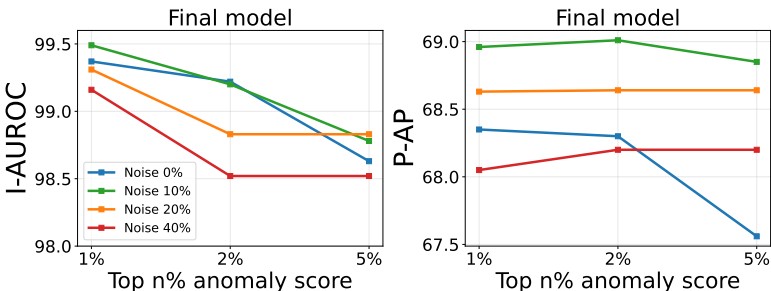

*Figure 8.* Analysis of the robust maximum for the selection score formulation.

**Adaptation to Baseline Methods** When applying MeDS to the baseline methods HVQ, Dinomaly, and INPFormer, we largely follow the original training configurations with the following modifications. For HVQ, we extend training to 200 epochs to ensure convergence under the data selection regime. For Dinomaly, we use a moderate discarding rate of 0.5 for the hard-mining loss to balance between noise robustness and learning efficiency. For INPFormer, we train for 100 epochs. All other hyperparameters remain consistent with the original implementations.

# C. Additional Experimental Results

*Table 5.* Ablation on MVTecAD dataset and INP-Former model where **bold highlights best performance**. Memory indicates usage of boostrapped memory ensemble, init shows how the final model is initialized, and criteria denotes the model $s_{\theta_0}$ used for data selection.

| | | | | | | I-AUROC (↑) | | | | I-AP (↑) | | | | P-AUPRO (↑) | | | | P-AP (↑) | | | |
|---|---|---|---|---|---|---|---|---|---|---|---|---|---|---|---|---|---|---|---|---|---|
| | | Ablation Setting | | | | | | | | | | | | Noise Ratio | | | | | | | |
| Memory | Distill | Fine-tune | Init | Criteria | Description | 0 | 10 | 20 | 40 | 0 | 10 | 20 | 40 | 0 | 10 | 20 | 40 | 0 | 10 | 20 | 40 |
| ✗ | ✗ | ✗ | ✗ | ✗ | Baseline | **99.66** | 95.13 | 91.21 | 85.85 | **99.88** | 97.34 | 94.10 | 91.14 | 94.88 | 91.06 | 89.64 | 88.85 | **70.55** | 59.88 | 54.39 | 51.26 |
| ✓ | ✗ | ✗ | ✗ | ✗ | Memory only | 98.57 | 97.47 | 94.74 | 92.22 | 99.41 | 98.75 | 97.02 | 96.00 | 91.99 | 91.81 | 91.86 | 90.5 | 62.13 | 62.22 | 56.05 | 53.04 |
| ✓ | ✓ | ✗ | ✗ | ✗ | Distilled model | 99.41 | 99.21 | 98.74 | 97.76 | 99.76 | 99.55 | 99.25 | 98.96 | 94.79 | 94.14 | 93.77 | 93.15 | 66.07 | 64.74 | 62.53 | 59.92 |
| ✓ | ✗ | ✓ | Random | Memory | Fine-tuned with memory | 97.25 | 97.87 | 98.04 | 98.13 | 98.46 | 98.63 | 98.81 | 98.94 | 94.71 | 94.55 | 94.49 | 94.34 | 64.60 | 65.15 | 64.90 | 65.64 |
| ✓ | ✓ | ✓ | Random | Distilled | Random init fine-tune | 98.38 | 98.14 | 98.95 | 98.71 | 99.13 | 98.84 | 99.57 | 99.36 | 94.60 | 94.68 | 94.73 | 94.83 | 65.64 | 65.77 | 67.10 | 66.74 |
| ✓ | ✓ | ✓ | Distilled | Distilled | MeDS (ours) | 99.45 | **99.39** | **99.41** | **99.17** | 99.78 | **99.79** | **99.78** | **99.68** | **95.13** | **95.25** | **95.22** | **95.21** | 67.15 | **67.79** | **67.51** | **67.39** |

## C.1. Additional Ablation

The ablation with INP-Former as baseline is given in Tab. 5.

## C.2. Analysis of the Selection Score

Instead of using the widely used maximum pixel anomaly score for the image level anomaly score, we minimize the risk of outlying scores by using the average of the top $n\%$ of scores instead. We evaluate three values and report the results in Fig. 8. Using 1% yields the best performance across all metrics.

## C.3. Full Performance Results

In the main paper, we reported class-averaged performance due to limited space. We report full class-wise performance results for anomaly detection experiments in Tab. 6 –29, and those corresponding to active label correction in Tab. 30.

*Table 6.* Per-class performance on MVTec-AD dataset for multi-class anomaly detection with AUROC/AP/F1-max metrics. Noise Ratio 0

| | SoftPatch | InReach | FUN-AD | HVQ | HVQ + MeDS (ours) | Dinomaly | Dinomaly + MeDS (ours) | INP-Former | INP-Former + MeDS (ours) |
|---|---|---|---|---|---|---|---|---|---|
| Carpet | 99.0/99.7/97.7 | 99.2/99.8/98.3 | **100./100./100.** | 99.1/99.7/98.3 | 99.9/100.0/99.4 | 99.8/99.9/98.9 | 99.9/100.0/99.4 | 99.9/100.0/99.4 | 99.9/100.0/99.4 |
| Grid | 98.9/99.6/98.2 | 91.7/97.0/90.3 | 82.7/90.2/90.3 | 94.7/98.1/93.2 | 90.3/97.2/89.1 | **99.9/100.0/99.1** | 99.3/99.8/99.1 | **99.9/100.0/99.1** | 99.6/99.9/99.1 |
| Leather | **100./100./100.** | **100./100./100.** | 95.3/97.8/98.4 | 99.3/99.8/95.9 | **100./100./100.** | **100./100./100.** | **100./100./100.** | **100./100./100.** | **100./100./100.** |
| Tile | 98.9/99.6/98.2 | 99.2/99.8/98.8 | 98.8/99.5/97.7 | 92.2/97.1/90.1 | 99.0/99.6/96.5 | **100./100./100.** | **100./100./100.** | **100./100./100.** | **100./100./100.** |
| Wood | 99.0/99.7/97.5 | 95.3/98.5/95.0 | 98.2/99.4/97.5 | 97.0/99.1/95.9 | 97.9/99.2/95.9 | 99.7/99.9/**99.2** | 99.5/99.8/98.4 | **99.8/99.9/99.2** | 99.7/99.9/98.4 |
| Bottle | **100./100./100.** | **100./100./100.** | 86.2/92.6/91.3 | **100./100./100.** | 99.5/99.9/99.2 | **100./100./100.** | **100./100./100.** | **100./100./100.** | **100./100./100.** |
| Cable | 99.5/99.7/96.8 | 95.1/97.2/91.6 | 79.4/83.3/81.4 | 97.2/98.3/93.3 | 95.8/97.0/89.7 | **100./100./100.** | 100.0/100.0/99.5 | **100./100./100.** | **100./100./100.** |
| Capsule | 96.4/99.2/97.3 | 52.5/85.0/90.5 | 62.1/87.3/92.4 | 92.7/98.1/95.9 | 88.3/97.2/94.2 | 98.1/99.6/98.2 | 98.1/99.6/98.2 | **98.8/99.7/98.6** | 98.6/99.7/98.2 |
| Hazelnut | **100./100./100.** | 99.1/99.5/96.3 | 98.6/99.3/97.8 | 100.0/100.0/99.3 | 99.9/99.9/99.3 | **100./100./100.** | **100./100./100.** | **100./100./100.** | **100./100./100.** |
| Metal Nut | 99.8/99.9/98.9 | 96.6/99.2/95.8 | 86.6/96.1/93.2 | 99.9/100.0/99.5 | 99.7/99.9/98.9 | **100./100./100.** | **100./100./100.** | **100./100./100.** | **100./100./100.** |
| Pill | 95.2/99.1/95.0 | 87.7/97.4/92.6 | 85.0/96.4/94.2 | 96.0/99.3/95.4 | 94.4/99.0/95.0 | 99.1/99.8/**98.6** | 98.5/99.7/98.2 | **99.2/99.9**/98.2 | 98.3/99.7/97.5 |
| Screw | 95.8/98.0/**97.1** | 75.8/90.3/86.5 | 70.6/81.2/86.5 | 91.9/96.8/92.4 | 86.3/93.8/89.9 | **98.7/99.6**/96.7 | 96.3/98.8/95.0 | 97.6/99.2/95.4 | 96.9/99.3/95.3 |
| Toothbrush | **100./100./100.** | 98.9/99.6/98.3 | 51.4/70.1/85.7 | 93.9/97.3/95.2 | 93.9/97.3/95.2 | **100./100./100.** | **100./100./100.** | **100./100./100.** | **100./100./100.** |
| Transistor | **100./100./100.** | 94.9/94.5/87.4 | 70.8/60.1/66.7 | 99.9/99.8/97.6 | 99.6/99.4/97.6 | 99.1/98.2/98.8 | 99.0/98.2/96.3 | 99.6/99.4/96.4 | 98.9/98.3/95.2 |
| Zipper | 99.1/99.8/98.3 | 94.0/98.1/95.0 | 62.6/83.8/88.5 | 96.8/99.0/97.5 | 93.9/98.7/95.5 | **100./100./100.** | **100./100./100.** | 100.0/100.0/99.6 | 99.8/100.0/99.2 |
| **Mean** | 98.8/99.6/98.3 | 92.0/97.1/94.4 | 81.9/89.1/90.8 | 96.7/98.8/96.1 | 95.9/98.5/95.7 | 99.6/99.9/**99.3** | 99.4/99.7/98.9 | **99.7/99.9**/99.1 | 99.4/99.8/98.8 |

*Table 7.* Per-class performance on MVTec-AD dataset for multi-class anomaly localization with AUROC/AP/F1-max/AUPRO metrics. Noise Ratio 0

| | SoftPatch | InReach | FUN-AD | HVQ | HVQ + MeDS | Dinomaly | Dinomaly + MeDS | INP-Former | INP-Former + MeDS |
|---|---|---|---|---|---|---|---|---|---|
| Carpet | 98.9/72.1/68.4/94.6 | 99.2/**76.4**/68.5/94.8 | 97.3/57.7/53.6/85.3 | 98.5/55.5/57.7/95.0 | 98.7/56.8/57.7/94.9 | 99.3/67.0/70.6/97.5 | 99.3/69.3/71.3/97.9 | **99.4**/71.8/**71.5**/97.9 | 99.3/70.1/69.8/**98.0** |
| Grid | 98.6/48.0/50.7/93.3 | 97.7/26.8/31.7/89.8 | 94.3/28.9/35.2/83.1 | 97.3/25.1/31.6/91.2 | 95.8/23.1/30.5/86.9 | 99.4/54.9/58.0/97.3 | 99.4/55.4/57.6/97.4 | **99.4**/**57.3**/**59.3**/97.4 | 99.4/49.7/54.2/**97.7** |
| Leather | 99.3/57.0/55.5/96.9 | 98.8/35.1/39.4/92.0 | 98.7/33.6/37.4/97.3 | 98.8/35.9/36.6/97.7 | 98.8/33.9/36.7/97.7 | 99.3/49.4/51.9/96.8 | 99.3/48.6/52.1/97.2 | 99.4/54.6/55.9/97.9 | 99.4/53.0/54.9/**98.6** |
| Tile | 95.9/63.3/69.5/81.3 | 96.5/73.1/70.6/85.0 | 85.2/43.0/43.0/12.2 | 88.7/36.1/48.0/77.1 | 91.6/40.9/51.5/85.8 | 98.2/**80.3**/76.5/90.7 | **98.3**/80.0/**77.2**/90.5 | 97.6/75.4/73.4/87.8 | 97.6/69.0/74.0/88.6 |
| Wood | 95.1/57.8/56.8/88.1 | 93.4/46.2/48.3/80.5 | 89.1/32.4/33.0/53.7 | 92.0/35.5/40.9/86.7 | 92.0/36.3/41.9/86.3 | **97.7**/72.8/68.7/93.5 | 97.5/72.8/68.2/93.8 | 97.7/**74.5**/**69.1**/93.7 | 97.4/70.8/67.3/**94.0** |
| Bottle | 98.7/83.0/79.1/94.7 | 98.0/76.3/73.8/91.0 | 90.8/29.7/37.1/46.0 | 98.3/72.8/71.4/94.6 | 98.2/72.4/70.0/94.5 | **99.1**/87.6/83.2/96.6 | 99.1/**88.2**/83.2/**96.8** | 98.9/85.7/80.7/96.7 | 98.9/85.7/80.7/96.7 |
| Cable | 98.4/73.8/70.3/93.0 | 97.6/60.5/61.8/85.0 | 90.8/29.7/37.1/46.0 | 97.8/51.8/57.9/87.4 | 97.3/47.9/52.9/85.5 | 97.1/63.1/65.5/93.3 | 98.1/66.9/68.2/93.4 | **98.8**/76.8/**75.8**/94.6 | **98.8**/75.7/74.4/**95.0** |
| Capsule | 98.7/52.1/55.9/94.5 | 94.4/20.8/27.6/64.4 | 93.7/9.3/20.8/15.4 | **98.8**/45.3/49.8/91.2 | 98.7/44.8/49.3/90.8 | 98.7/**61.3**/**60.5**/97.5 | 98.7/60.6/59.4/**97.7** | 98.7/61.3/58.7/97.6 | 98.7/59.2/55.9/97.0 |
| Hazelnut | 97.7/68.5/67.8/95.1 | 97.5/49.4/53.3/88.2 | 97.7/65.0/56.9/79.7 | 97.7/65.0/56.9/79.7 | 97.8/62.5/57.2/92.4 | 99.4/83.0/76.3/96.9 | **99.5**/81.0/**77.1**/96.9 | 99.4/77.2/75.0/96.8 | 99.4/77.2/75.0/96.8 |
| Metal Nut | **98.4**/90.5/**86.6**/94.0 | 98.2/88.5/84.5/89.6 | 97.5/**90.7**/82.2/68.9 | 96.7/68.7/76.2/90.8 | 96.4/67.4/74.7/90.6 | 96.7/77.9/85.9/94.4 | 96.8/78.3/86.5/94.8 | 97.3/80.5/86.3/95.0 | 97.0/77.3/85.1/94.8 |
| Pill | 97.2/**79.0**/**74.0**/94.0 | 95.9/51.0/57.6/89.6 | 94.9/6.6/14.7/37.1 | 96.3/50.9/58.0/95.1 | 95.7/46.4/54.5/94.7 | **99.7**/61.3/60.5/98.5 | 99.6/60.5/60.4/98.3 | 97.7/74.4/70.3/97.5 | 97.4/69.6/67.3/97.2 |
| Screw | 99.2/45.6/47.1/95.8 | 98.1/19.3/30.6/90.7 | 94.9/6.6/14.7/37.1 | 98.7/27.8/36.2/93.8 | 97.9/15.4/23.0/90.8 | **99.7**/61.3/**60.5/98.5** | 99.6/60.5/60.4/98.3 | 99.6/**62.0**/59.9/98.0 | 99.6/59.0/59.7/96.9 |
| Toothbrush | 98.8/**63.9**/64.2/89.8 | 98.8/55.9/62.6/84.6 | 84.1/7.2/14.3/40.0 | 98.5/38.0/49.6/88.5 | 98.6/41.0/51.1/90.7 | 98.9/52.1/62.7/95.3 | 98.9/52.4/63.4/95.3 | **99.1**/57.3/**66.0**/95.7 | 99.0/51.8/62.3/95.7 |
| Transistor | 96.3/70.5/65.4/92.2 | 96.9/72.6/**69.0**/85.6 | 95.3/53.4/64.5/26.0 | 98.2/71.6/68.9/**95.1** | **98.3**/**72.8**/68.3/94.8 | 92.9/58.2/56.6/76.4 | 93.5/59.5/57.4/76.6 | 95.1/66.3/63.1/79.7 | 95.4/65.7/61.8/83.6 |
| Zipper | 98.8/69.8/69.5/94.9 | 93.8/41.4/48.0/81.3 | 89.7/29.9/32.4/76.5 | 97.6/40.2/50.1/92.5 | 96.5/33.7/43.1/91.3 | **99.3**/**79.6**/**75.5**/**97.1** | 99.2/78.7/74.2/97.1 | 99.0/76.5/72.6/96.6 | 98.8/73.3/69.9/96.5 |
| **Mean** | 98.1/66.3/65.4/92.8 | 97.0/52.9/55.2/86.1 | 93.7/44.3/46.1/59.1 | 97.0/47.7/53.1/91.3 | 96.9/46.4/51.2/91.2 | 98.3/68.2/68.2/94.6 | 98.3/68.4/68.3/94.7 | **98.5**/**70.5**/**69.5**/94.9 | 98.4/67.1/67.5/**95.1** |

*Table 8.* Per-class performance on MVTec-AD dataset for multi-class anomaly detection with AUROC/AP/F1-max metrics. Noise Ratio 10

| | SoftPatch | InReach | FUN-AD | HVQ | HVQ + MeDS (ours) | Dinomaly | Dinomaly + MeDS (ours) | INP-Former | INP-Former + MeDS (ours) |
|---|---|---|---|---|---|---|---|---|---|
| Carpet | 99.0/99.7/98.9 | 97.1/99.2/97.1 | **100./100./100.** | 99.6/99.9/98.9 | 99.8/100.0/98.9 | 98.6/99.6/97.8 | 99.9/100.0/99.4 | 99.3/99.8/98.9 | 99.9/100.0/99.4 |
| Grid | 97.2/99.1/97.3 | 94.2/97.9/92.9 | 99.6/99.8/98.3 | 68.9/88.2/84.4 | 87.1/95.6/89.1 | **99.9/100.0/99.1** | 99.7/99.9/99.1 | 99.8/99.9/99.1 | 99.7/99.9/99.1 |
| Leather | **100./100./100.** | 99.1/99.8/99.5 | **100./100./100.** | **100./100./100.** | **100./100./100.** | **100./100./100.** | **100./100./100.** | **100./100./100.** | **100./100./100.** |
| Tile | 98.0/99.4/98.2 | 97.7/99.3/96.4 | 99.4/99.8/98.8 | 98.1/99.3/95.8 | 99.2/99.7/97.1 | **100./100./100.** | **100./100./100.** | 98.9/99.6/97.1 | **100./100./100.** |
| Wood | 99.0/99.7/96.7 | 89.2/96.3/91.2 | 99.5/99.8/**99.2** | 97.5/99.2/95.9 | 97.3/99.1/95.7 | 91.0/96.8/92.7 | **99.7/99.9/99.2** | 82.1/93.8/90.1 | **99.7/99.9**/98.4 |
| Bottle | **100./100./100.** | 98.7/99.7/99.2 | 99.6/99.9/99.2 | 99.6/99.9/99.2 | 99.6/99.9/99.2 | 98.4/99.6/97.6 | **100./100./100.** | 98.5/99.5/97.6 | **100./100./100.** |
| Cable | 99.2/99.5/96.8 | 84.8/92.0/82.6 | 95.8/98.0/94.4 | 89.4/93.1/86.0 | 94.9/96.7/90.3 | 81.8/96.0/89.5 | **100./100./100.** | 94.8/97.3/91.0 | 100.0/100.0/99.5 |
| Capsule | 95.7/99.0/96.4 | 49.4/83.9/90.5 | 92.3/98.4/92.6 | 83.1/95.2/93.5 | 82.2/95.2/93.5 | 94.3/98.7/96.3 | 97.9/99.6/97.3 | 95.5/98.9/96.4 | **98.3/99.8/98.2** |
| Hazelnut | 95.5/98.3/96.4 | 97.1/98.5/94.3 | 99.9/100.0/99.3 | 99.6/99.8/97.9 | 99.8/99.9/98.6 | 82.6/90.9/82.1 | **100./100./100.** | 87.0/92.2/86.9 | **100./100./100.** |
| Metal Nut | 100.0/100.0/99.5 | 90.6/98.0/93.3 | 99.9/100.0/99.5 | 95.5/99.9/98.9 | 98.8/100.0/99.5 | 97.7/99.5/95.8 | **100./100./100.** | 95.5/99.0/94.7 | **100./100./100.** |
| Pill | 95.4/99.2/95.1 | 81.2/96.2/91.6 | 95.8/99.2/96.8 | 92.7/98.6/94.8 | 94.6/99.0/95.4 | 97.0/99.5/96.2 | 98.5/99.7/**98.2** | 97.8/99.6/96.9 | **98.8/99.9**/98.0 |
| Screw | 95.6/97.5/96.3 | 65.4/87.3/85.3 | 86.7/94.7/90.0 | 76.0/90.1/86.2 | 84.4/93.4/89.4 | 91.7/97.4/91.2 | **97.8/99.3/96.3** | 91.8/97.3/91.1 | 95.8/**99.3**/93.4 |
| Toothbrush | 99.7/99.9/98.4 | 96.7/99.0/98.3 | 71.1/85.2/87.5 | 94.7/97.7/95.2 | 94.7/97.7/95.2 | 98.3/99.4/98.3 | **100./100./100.** | 98.1/99.3/96.7 | **100./100./100.** |
| Transistor | **99.3/99.1/97.4** | 77.4/82.2/76.9 | 93.8/91.4/84.2 | 95.6/95.1/87.5 | 97.8/97.7/92.1 | 86.3/82.8/75.9 | 99.0/98.2/96.2 | 88.2/83.9/77.5 | 98.9/98.4/96.5 |
| Zipper | 98.3/99.5/97.9 | 92.7/97.8/94.9 | 98.4/99.6/97.0 | 72.2/92.1/88.7 | 93.0/97.7/95.5 | **100./100./100.** | **100./100./100.** | 99.8/99.9/98.7 | 99.9/100.0/99.6 |
| **Mean** | 98.1/99.3/97.7 | 87.4/95.1/92.3 | 95.5/97.7/95.8 | 91.1/96.5/93.5 | 95.0/98.1/95.3 | 95.2/97.3/94.2 | **99.5**/99.8/**99.1** | 95.1/97.3/94.2 | 99.4/**99.8**/98.8 |

*Table 9.* Per-class performance on MVTec-AD dataset for multi-class anomaly localization with AUROC/AP/F1-max/AUPRO metrics. Noise Ratio 10

| | SoftPatch | InReach | FUN-AD | HVQ | HVQ + MeDS (ours) | Dinomaly | Dinomaly + MeDS (ours) | INP-Former | INP-Former + MeDS (ours) |
|---|---|---|---|---|---|---|---|---|---|
| Carpet | 97.4/66.5/66.6/91.4 | 99.0/75.7/68.9/93.7 | 99.2/**78.0**/69.7/95.8 | 98.7/57.4/57.9/95.4 | 98.7/57.0/57.6/94.7 | 99.3/71.5/70.2/97.5 | 99.3/68.3/**70.8**/97.8 | 99.3/71.9/69.1/97.2 | **99.3**/68.2/69.5/**97.9** |
| Grid | 96.7/45.9/50.9/87.4 | 97.8/26.9/32.4/90.0 | 98.9/53.0/54.5/94.9 | 89.6/12.5/19.8/72.3 | 95.7/22.1/29.5/85.1 | 99.0/42.7/46.1/96.0 | **99.4**/**56.3**/**58.9**/97.1 | 99.0/45.6/46.8/96.0 | 99.4/51.9/55.0/**97.7** |
| Leather | 98.7/51.5/54.4/96.4 | 97.3/39.5/45.1/95.7 | **99.7**/**65.5**/**64.4**/98.4 | 98.8/34.0/39.7/97.9 | 98.8/34.0/39.9/97.9 | 99.3/49.7/52.7/97.6 | 99.2/55.9/53.8/96.6 | 99.2/55.6/52.6/96.6 | 99.5/56.2/56.6/**98.6** |
| Tile | 94.9/69.4/70.6/81.0 | 95.2/71.9/71.6/79.6 | 89.9/50.8/48.8/63.3 | 92.1/46.3/54.7/81.5 | 91.7/41.1/51.9/81.8 | 97.8/78.1/75.1/87.8 | **98.2**/78.9/**76.9**/90.8 | 97.7/**80.6**/74.8/85.2 | 97.7/71.7/74.1/88.9 |
| Wood | 93.2/56.3/55.5/85.2 | 87.1/43.1/45.7/81.3 | 96.3/55.9/53.9/88.9 | 92.8/40.1/44.7/87.3 | 92.4/37.3/42.8/86.3 | 96.0/58.9/57.4/86.4 | **97.6**/**73.2**/**69.3**/93.4 | 94.8/55.6/49.3/83.3 | 97.5/70.8/67.0/**93.9** |
| Bottle | 94.5/74.3/75.0/82.0 | 97.6/73.4/72.8/88.5 | **99.2**/**92.0**/**84.2**/96.4 | 97.8/66.7/68.6/94.0 | 98.2/72.1/70.6/94.6 | 97.9/79.1/75.9/93.5 | 99.1/87.6/83.4/96.6 | 98.0/80.7/76.0/94.0 | 99.0/86.5/81.7/**96.9** |
| Cable | 90.2/51.3/57.9/85.1 | 95.5/54.9/55.7/78.2 | 92.0/43.6/52.5/56.0 | 92.8/33.0/38.7/80.9 | 97.2/46.3/52.4/86.5 | 85.7/39.8/47.2/81.1 | 98.6/**74.0**/**74.6**/94.0 | 89.2/46.7/50.4/84.9 | **98.7**/71.6/73.7/**94.8** |
| Capsule | 98.8/52.4/56.2/94.8 | 87.8/14.6/22.7/52.6 | 98.3/50.7/51.0/80.6 | 98.5/50.7/51.0/80.6 | 98.6/62.1/61.9/91.4 | 98.0/59.1/59.5/91.0 | 98.6/**60.4**/59.7/**97.5** | 98.1/58.2/59.2/92.9 | **99.5**/80.3/**76.6**/96.7 |
| Hazelnut | 75.3/34.4/42.9/71.9 | 96.3/47.4/52.1/85.3 | 99.2/81.7/75.9/83.0 | 98.5/62.1/61.9/91.4 | 98.6/62.8/62.8/91.9 | 98.0/59.1/59.5/91.0 | 99.4/**82.2**/76.4/**97.1** | 98.1/58.2/59.2/92.9 | **99.5**/80.3/**76.6**/96.7 |
| Metal Nut | 83.6/64.4/64.8/87.6 | 93.4/74.6/72.5/81.6 | **93.9**/**96.4**/90.6/87.9 | 94.2/59.4/64.6/90.3 | 96.1/66.3/73.4/90.7 | 83.6/48.8/50.6/86.5 | 97.2/80.7/87.0/94.9 | 96.4/52.7/54.2/86.6 | 97.2/79.2/84.8/**95.1** |
| Pill | 96.7/78.2/73.7/92.3 | 93.7/48.1/54.5/88.1 | **99.0**/81.9/**76.1**/84.5 | 95.2/43.5/52.7/94.1 | 95.9/47.2/55.1/94.7 | 96.6/71.5/66.3/96.6 | 97.9/74.0/70.0/97.2 | 97.0/73.4/67.7/96.6 | 97.3/70.5/68.3/**97.3** |
| Screw | 96.8/44.0/46.7/91.3 | 95.0/21.5/31.5/82.7 | 97.7/40.6/42.8/59.1 | 97.5/17.4/26.1/89.9 | 98.1/19.5/27.3/91.1 | 99.5/50.2/53.3/97.6 | **99.6**/59.4/60.0/**98.3** | 99.3/51.6/52.6/96.8 | 99.4/58.8/57.5/97.8 |
| Toothbrush | 98.8/**63.4**/64.4/88.1 | 98.4/54.2/60.5/83.3 | 90.1/9.4/18.1/39.2 | 98.6/41.8/50.9/88.4 | 98.6/58.9/61.8/95.5 | 98.9/51.7/62.7/95.3 | 98.8/61.7/**64.5**/95.2 | 98.8/61.7/64.5/95.2 | **99.0**/51.8/62.6/**98.6** |
| Transistor | 79.2/52.9/55.1/71.3 | 91.1/58.9/57.9/70.8 | 95.6/55.2/62.8/36.0 | 93.4/56.1/57.0/87.4 | **97.1**/**66.8**/**64.8**/93.1 | 85.1/38.3/39.6/66.7 | 93.4/60.2/58.2/75.7 | 88.1/44.6/44.8/69.0 | 95.4/66.0/63.5/82.9 |
| Zipper | 95.0/59.9/64.0/90.3 | 90.0/39.9/48.2/75.8 | 97.8/66.3/62.8/91.7 | 92.2/18.5/26.4/76.0 | 96.7/34.3/43.9/89.1 | 98.8/73.8/70.7/96.0 | **99.2**/**77.7**/**73.8**/97.0 | 98.4/69.4/67.2/94.9 | 98.9/74.3/70.8/96.7 |
| **Mean** | 92.6/57.7/59.9/86.4 | 94.3/49.6/52.8/81.8 | 96.8/61.8/60.9/77.1 | 95.4/42.5/47.6/87.8 | 96.8/46.1/51.3/90.4 | 95.6/58.2/58.6/91.0 | 98.4/**69.0**/**68.9**/94.7 | 96.1/59.9/58.9/91.1 | **98.4**/67.8/67.9/**95.2** |

*Table 10.* Per-class performance on MVTec-AD dataset for multi-class anomaly detection with AUROC/AP/F1-max metrics. Noise Ratio 20

| | SoftPatch | InReach | FUN-AD | HVQ | HVQ + MeDS (ours) | Dinomaly | Dinomaly + MeDS (ours) | INP-Former | INP-Former + MeDS (ours) |
|---|---|---|---|---|---|---|---|---|---|
| Carpet | 98.0/99.4/96.0 | 93.2/98.2/93.0 | **100.0/100.0/99.4** | 99.7/99.9/98.9 | 99.8/99.9/98.9 | 98.0/99.3/97.8 | 99.9/100.0/**99.4** | 98.1/99.3/97.8 | 99.9/100.0/**99.4** |
| Grid | 96.6/98.9/96.4 | 90.5/96.8/90.6 | 99.3/99.8/**99.1** | 78.0/92.0/85.5 | 84.5/94.7/85.5 | 99.5/99.8/**99.1** | 99.6/99.9/**99.1** | **99.7/99.9/99.1** | **99.7/99.9/99.1** |
| Leather | **100./100./100.** | 91.8/97.6/93.3 | **100./100./100.** | **100./100./100.** | **100./100./100.** | **100./100./100.** | **100./100./100.** | **100./100./100.** | **100./100./100.** |
| Tile | 97.3/99.2/97.0 | 88.7/96.2/90.0 | 98.6/99.5/97.1 | 98.1/99.3/96.4 | 98.4/99.4/97.0 | 99.6/99.8/98.8 | **100./100./100.** | 98.5/99.4/96.5 | **100./100./100.** |
| Wood | 97.8/99.3/96.7 | 58.6/85.5/86.3 | 98.7/99.6/96.7 | 97.1/99.1/95.1 | 97.5/99.2/95.9 | 84.2/93.8/89.9 | 99.7/99.9/**99.2** | 67.2/85.5/89.5 | **99.7/99.9/**98.4 |
| Bottle | 99.5/99.8/98.4 | 98.7/99.7/99.2 | **100./100./100.** | 99.2/99.7/99.2 | 99.8/100.0/99.2 | 98.2/99.5/96.8 | **100./100./100.** | 98.5/99.6/97.6 | **100./100./100.** |
| Cable | 96.1/98.1/93.9 | 80.5/90.3/82.2 | 96.6/98.1/92.5 | 88.9/93.2/87.0 | 90.5/94.9/88.9 | 81.7/91.1/81.2 | 98.9/99.6/99.5 | 86.8/93.2/84.2 | **100./100./100.** |
| Capsule | 95.2/98.9/96.4 | 39.6/82.0/90.5 | 95.7/99.1/96.0 | 82.3/95.1/92.9 | 86.8/96.1/93.8 | 91.1/97.8/94.5 | 97.9/99.5/**97.7** | 92.7/98.0/95.2 | **98.6/99.7/**97.7 |
| Hazelnut | 95.2/98.1/96.4 | 87.9/94.6/86.9 | 99.9/99.9/99.3 | 99.9/99.9/99.3 | 100.0/100.0/99.3 | 82.9/90.4/84.4 | **100./100./100.** | 81.2/87.3/84.2 | **100./100./100.** |
| Metal Nut | 95.3/99.0/96.1 | 85.5/96.7/90.0 | 99.7/99.9/99.5 | 99.2/99.8/97.8 | 99.7/99.9/98.9 | 93.0/98.4/93.0 | **100./100./100.** | 89.3/97.6/91.2 | **100./100./100.** |
| Pill | 95.4/99.2/95.1 | 72.7/94.6/91.6 | 96.8/99.3/96.9 | 91.8/98.4/94.2 | 94.4/99.0/95.4 | 95.3/99.1/95.8 | 98.1/99.6/**98.2** | 96.3/99.3/96.4 | **98.4/99.7/**97.5 |
| Screw | 93.3/97.0/93.7 | 50.8/80.4/85.3 | 91.5/97.2/90.0 | 75.5/90.2/86.9 | 84.7/93.8/89.5 | 85.7/95.2/88.4 | **96.6/98.9/**94.3 | 87.2/95.6/88.5 | 95.9/98.8/**94.4** |
| Toothbrush | 99.7/99.9/98.4 | 92.8/97.8/93.3 | 66.7/86.3/85.7 | 94.4/97.7/93.8 | 93.9/97.3/95.2 | 98.6/99.5/98.3 | **100./100./100.** | 96.1/98.5/94.9 | **100./100./100.** |
| Transistor | 97.2/97.4/94.9 | 58.6/68.6/64.4 | 98.4/97.6/93.8 | 96.2/95.4/88.0 | 98.8/98.4/95.1 | 74.5/68.6/68.3 | 99.1/98.4/**97.5** | 76.9/61.6/71.1 | **99.2/98.7/**96.4 |
| Zipper | 96.6/98.9/97.5 | 92.2/97.7/93.2 | 99.0/99.7/97.4 | 72.0/90.8/88.5 | 92.7/97.6/95.9 | 100.0/100.0/99.6 | **100./100./100.** | 99.8/99.9/98.7 | 99.9/100.0/99.6 |
| **Mean** | 96.9/98.9/96.5 | 78.8/91.8/88.6 | 96.1/98.4/96.2 | 91.5/96.7/93.6 | 94.8/98.0/95.2 | 92.2/95.5/92.4 | 99.3/99.7/**99.0** | 91.2/94.3/92.3 | **99.4/99.8/**98.8 |

*Table 11.* Per-class performance on MVTec-AD dataset for multi-class anomaly localization with AUROC/AP/F1-max/AUPRO metrics. Noise Ratio 20

| | SoftPatch | InReach | FUN-AD | HVQ | HVQ + MeDS (ours) | Dinomaly | Dinomaly + MeDS (ours) | INP-Former | INP-Former + MeDS (ours) |
|---|---|---|---|---|---|---|---|---|---|
| Carpet | 81.7/42.2/54.3/74.2 | 96.3/65.8/64.6/87.3 | **99.3/79.3/70.9**/95.4 | 98.7/56.2/57.8/95.7 | 98.7/56.2/57.3/95.0 | 99.2/68.3/68.6/97.4 | 99.3/69.0/70.8/97.5 | 99.1/67.0/66.7/96.8 | **99.3**/69.9/69.8/**98.0** |
| Grid | 95.3/44.2/49.3/85.4 | 99.1/**56.8**/57.4/94.6 | 99.1/**56.8**/57.4/94.6 | 95.4/21.6/28.8/85.1 | 95.4/21.5/28.9/85.1 | 99.2/45.5/53.6/96.8 | **99.4**/56.6/**58.8**/97.1 | 99.0/45.5/53.6/96.8 | 99.4/52.1/55.2/**97.7** |
| Leather | 97.5/53.8/56.1/92.2 | 96.5/32.9/38.8/87.2 | **99.6/62.2/61.8**/95.5 | 99.2/40.4/44.0/98.0 | 98.8/34.1/36.9/97.8 | 99.0/42.2/47.0/96.4 | 99.3/49.9/52.2/97.1 | 98.9/40.9/44.9/96.6 | 99.5/56.1/58.0/**98.7** |
| Tile | 82.9/49.1/58.6/63.8 | 88.5/61.6/64.6/71.6 | 95.5/70.5/70.5/79.3 | 92.7/49.0/56.5/82.7 | 91.7/41.0/52.1/81.1 | 97.0/71.1/71.2/87.1 | **98.2/77.9/77.0/90.3** | 96.3/71.7/67.6/82.6 | 97.7/70.0/74.7/89.0 |
| Wood | 90.4/52.1/52.2/76.8 | 77.0/25.2/33.7/64.0 | 94.1/46.4/50.6/80.0 | 93.2/42.0/46.4/87.7 | 92.1/36.7/42.1/86.9 | 96.6/60.7/63.1/86.7 | **97.5/72.2/67.9/93.9** | 94.8/48.8/52.0/79.3 | 97.4/71.7/67.4/93.9 |
| Bottle | 76.9/53.1/58.6/61.5 | 97.7/76.6/73.5/89.6 | 99.0/**89.8**/81.5/90.0 | 97.9/66.2/69.2/94.3 | 98.1/71.2/69.6/94.3 | 97.3/75.4/73.4/92.8 | **99.1**/87.5/**83.5**/96.9 | 97.5/76.3/73.3/93.2 | 99.1/87.1/82.3/**97.0** |
| Cable | 77.4/39.7/49.7/71.0 | 90.5/46.0/51.1/67.8 | 93.8/52.7/58.8/54.8 | 93.2/32.9/38.9/80.0 | 88.3/32.6/40.2/80.1 | 81.6/35.1/42.8/77.6 | 98.0/70.6/72.4/93.4 | 95.3/39.5/45.6/80.0 | **98.8/74.0/74.7/95.1** |
| Capsule | 98.2/48.1/53.2/93.1 | 83.1/14.1/21.7/46.5 | 98.7/54.4/55.0/80.5 | 98.7/45.3/50.1/89.2 | **98.7**/43.6/48.5/90.8 | 98.1/47.9/51.1/96.3 | 98.7/**61.1/59.9/97.7** | 99.1/48.1/51.3/96.1 | 98.7/57.9/55.7/97.0 |
| Hazelnut | 80.2/42.4/49.5/62.8 | 96.1/53.1/54.9/79.8 | 98.9/75.7/69.9/97.0 | 98.5/63.1/62.7/91.3 | 98.7/62.9/63.0/92.3 | 98.4/66.3/63.1/90.8 | **99.4/82.5/76.3/96.9** | 98.2/62.8/59.9/90.7 | **99.4**/78.3/75.3/96.9 |
| Metal Nut | 69.0/45.8/48.4/81.9 | 90.2/66.6/64.0/68.5 | **99.4/97.0/91.4**/93.4 | 92.3/54.4/55.8/80.4 | 93.2/54.4/55.8/80.4 | 78.8/41.0/43.3/83.2 | 96.5/77.6/68.4/**98.4** | 81.2/43.6/46.5/83.4 | 96.8/78.6/84.2/95.1 |
| Pill | 96.8/**78.0/74.5**/91.5 | 93.5/47.4/54.7/85.3 | **98.1**/62.5/66.7/72.9 | 94.9/40.9/49.8/93.9 | 95.9/46.5/54.9/94.8 | 96.5/63.4/63.8/95.5 | 97.8/73.6/70.1/**97.3** | 96.8/67.5/65.6/96.0 | 97.5/70.9/68.0/97.2 |
| Screw | 93.6/40.1/45.5/85.0 | 93.3/21.4/27.2/47.0 | 96.7/17.7/27.7/90.1 | 94.9/40.9/49.8/93.9 | 98.2/20.1/28.7/91.0 | 99.3/41.1/48.7/97.0 | **99.6/60.3/60.2**/97.9 | 99.0/45.3/49.7/96.6 | 99.5/57.4/56.2/97.6 |
| Toothbrush | 97.8/**63.9/64.5**/86.3 | 98.4/54.2/60.1/82.2 | 86.8/19.6/26.9/44.8 | 98.6/41.6/51.5/88.7 | 98.6/41.8/51.9/89.0 | 98.8/56.5/61.1/95.2 | 98.9/52.4/63.3/95.3 | 98.6/58.2/58.9/94.2 | **99.0**/50.5/61.6/**95.5** |
| Transistor | 67.2/43.2/49.5/59.9 | 87.2/52.1/50.1/59.9 | 95.7/56.4/**64.8**/31.4 | 91.7/53.5/55.9/86.5 | **96.2**/61.2/63.8/**93.0** | 82.7/31.5/35.3/64.1 | 93.5/60.9/59.1/75.9 | 85.5/34.8/40.1/67.7 | 95.3/**64.3**/62.2/83.0 |
| Zipper | 85.9/44.9/53.2/80.8 | 88.6/38.3/46.7/74.4 | 97.9/61.8/59.4/81.8 | 92.5/19.1/27.1/76.6 | 96.8/34.3/44.3/90.4 | 98.8/72.4/70.5/96.4 | **99.2/77.3/73.5/96.9** | 98.1/67.4/67.1/94.7 | 98.9/73.6/70.8/96.7 |
| **Mean** | 86.1/49.4/54.5/77.7 | 91.6/45.0/49.2/75.3 | 97.0/62.2/62.3/75.5 | 95.6/42.7/48.1/88.4 | 96.1/44.6/50.3/90.2 | 94.8/54.6/57.1/90.2 | 98.3/**68.6/68.7**/94.6 | 95.1/54.4/56.2/89.6 | **98.4**/67.5/67.8/**95.2** |

*Table 12.* Per-class performance on MVTec-AD dataset for multi-class anomaly detection with AUROC/AP/F1-max metrics. Noise Ratio 20

| | SoftPatch | InReach | FUN-AD | HVQ | HVQ + MeDS (ours) | Dinomaly | Dinomaly + MeDS (ours) | INP-Former | INP-Former + MeDS (ours) |
|---|---|---|---|---|---|---|---|---|---|
| Carpet | 96.9/99.1/94.9 | 81.5/94.7/86.4 | 99.8/99.9/**99.4** | 99.7/99.9/98.9 | 99.8/99.9/98.9 | 98.4/99.5/97.8 | **99.9/100.0/99.4** | 97.9/99.2/97.3 | **99.9/100.0/99.4** |
| Grid | 97.2/99.2/96.4 | 90.1/96.5/91.4 | **99.8/99.9/99.1** | 75.3/90.8/84.8 | 86.8/95.4/87.7 | 99.4/99.8/**99.1** | 99.8/99.9/**99.1** | 99.5/99.8/**99.1** | 99.6/99.9/**99.1** |
| Leather | 99.6/99.9/98.4 | 90.2/97.3/92.7 | **100./100./100.** | **100./100./100.** | **100./100./100.** | 99.9/100.0/99.5 | **100./100./100.** | **100./100./100.** | **100./100./100.** |
| Tile | 93.2/97.9/93.8 | 79.3/92.7/83.6 | 99.3/99.7/98.2 | 98.0/99.2/96.5 | 98.2/99.3/96.3 | 99.7/99.9/98.8 | **100./100./100.** | 97.0/98.8/95.3 | **100./100./100.** |
| Wood | 96.5/98.9/95.1 | 66.0/88.3/86.3 | 99.7/99.9/**99.2** | 96.8/99.0/95.1 | 97.3/99.2/95.9 | 81.4/91.5/90.2 | 99.7/99.9/**99.2** | 58.6/75.7/89.5 | **99.8/100.0/**99.2 |
| Bottle | 99.0/99.7/97.7 | 99.4/99.8/97.7 | **100./100./100.** | 99.2/99.7/99.2 | 99.8/99.9/99.2 | 97.5/99.3/95.9 | **100./100./100.** | 97.7/99.3/96.8 | **100./100./100.** |
| Cable | 84.8/92.2/82.6 | 62.4/81.6/76.0 | 94.8/97.1/89.3 | 85.1/90.8/82.4 | 86.8/93.1/86.7 | 58.4/77.2/76.0 | 96.7/98.7/97.8 | 63.4/77.5/76.8 | **99.0/99.6/98.9** |
| Capsule | 93.2/98.3/95.6 | 46.6/82.7/90.5 | 94.4/98.8/94.7 | 83.7/95.3/92.6 | 86.3/95.9/93.1 | 85.2/95.2/93.0 | **98.3/99.6/**97.7 | 88.3/96.1/94.2 | 98.3/**99.7/98.2** |
| Hazelnut | 91.0/96.2/91.8 | 85.1/93.1/83.8 | 99.4/99.7/97.9 | 99.9/100.0/99.3 | 99.9/100.0/99.3 | 75.9/86.2/79.0 | **100./100./100.** | 74.7/80.4/81.6 | **100./100./100.** |
| Metal Nut | 86.5/97.2/90.4 | 72.9/92.6/89.4 | 100.0/100.0/99.5 | 98.2/99.6/96.9 | 99.7/99.9/99.5 | 76.0/94.5/89.4 | **100./100./100.** | 68.1/92.0/89.4 | **100./100./100.** |
| Pill | 89.0/98.0/91.6 | 56.9/91.0/91.6 | 97.8/99.6/96.8 | 91.9/98.4/94.5 | 93.7/98.8/94.2 | 93.9/98.8/95.1 | **98.9/99.8/98.2** | 95.7/99.2/95.7 | 98.4/99.7/97.5 |
| Screw | 86.3/93.7/89.1 | 29.3/65.5/85.3 | 88.5/96.0/89.7 | 78.1/90.4/85.9 | 80.3/92.3/86.9 | 74.9/90.5/85.3 | **95.1/98.5/94.6** | 78.5/91.6/86.1 | 93.7/98.2/94.0 |
| Toothbrush | **100./100./100.** | 94.2/98.1/93.3 | 96.9/98.9/94.9 | 93.3/97.1/95.2 | 92.8/96.9/95.2 | 96.9/98.9/94.9 | **100./100./100.** | 92.8/96.8/93.8 | **100./100./100.** |
| Transistor | 97.2/97.4/94.9 | 59.8/69.2/61.8 | 98.6/97.8/94.0 | 95.8/94.9/86.1 | 98.8/**98.4**/93.3 | 73.3/68.0/64.4 | **99.0**/98.0/96.3 | 75.9/60.9/70.7 | **99.0**/98.3/**96.4** |
| Zipper | 92.2/97.1/94.3 | 89.9/97.0/91.8 | 96.5/99.1/95.9 | 87.5/95.6/94.8 | 93.8/98.0/96.3 | 99.8/99.9/98.7 | **100./100./100.** | 99.5/99.9/98.3 | 99.9/100.0/99.6 |
| **Mean** | 93.5/97.7/93.8 | 73.6/89.3/86.8 | 97.7/99.1/96.6 | 92.1/96.7/93.4 | 94.3/97.8/94.8 | 87.4/93.3/90.5 | 99.2/99.6/**98.8** | 85.8/91.1/91.0 | **99.2/99.7/**98.8 |

*Table 13.* Per-class performance on MVTec-AD dataset for multi-class anomaly localization with AUROC/AP/F1-max/AUPRO metrics. Noise Ratio 20

| | SoftPatch | InReach | FUN-AD | HVQ | HVQ + MeDS (ours) | Dinomaly | Dinomaly + MeDS (ours) | INP-Former | INP-Former + MeDS (ours) |
|---|---|---|---|---|---|---|---|---|---|
| Carpet | 79.6/33.0/42.5/59.8 | 96.3/66.3/64.2/85.8 | 99.2/**77.3**/69.9/95.8 | 98.7/57.1/58.3/95.7 | 98.7/56.2/57.4/94.9 | **99.4**/76.0/**71.7**/97.8 | 99.3/68.4/71.2/97.5 | 99.3/74.8/69.3/97.2 | 99.3/67.6/69.1/**97.8** |
| Grid | 95.8/45.3/50.2/86.8 | 97.5/27.8/33.9/88.7 | 99.1/54.3/57.2/90.6 | 94.3/19.2/26.9/82.7 | 95.3/21.5/28.7/85.6 | 99.2/48.6/54.2/96.9 | **99.4/55.9/59.0**/97.5 | 99.2/49.2/54.9/96.8 | 99.4/52.3/55.0/**97.7** |
| Leather | 93.1/50.3/51.2/79.3 | 95.5/36.0/42.5/87.3 | 99.2/58.2/57.0/82.6 | 99.3/44.9/47.1/98.0 | 99.1/48.7/50.9/97.7 | 99.1/48.7/50.6/96.8 | 99.3/48.4/52.5/97.5 | 99.0/45.5/49.2/96.9 | **99.5/54.4/57.2/98.7** |
| Tile | 67.4/30.9/40.7/42.7 | 81.0/48.6/50.5/55.6 | 91.7/70.9/67.6/63.2 | 93.0/48.5/56.2/83.0 | 91.4/40.5/52.1/81.1 | 96.1/69.6/68.8/84.3 | **98.2/77.7/77.2/90.3** | 95.5/67.9/66.5/80.5 | 97.7/71.0/74.8/89.2 |
| Wood | 86.9/45.5/49.0/72.3 | 79.0/27.8/36.8/62.0 | 96.2/51.6/59.8/85.3 | 93.7/42.9/47.3/88.0 | 92.2/37.5/42.7/86.9 | 96.2/62.6/62.5/85.6 | **97.6/73.0/68.6**/93.4 | 94.7/44.7/53.0/79.4 | 97.5/71.3/67.4/**94.0** |
| Bottle | 51.3/24.1/32.9/40.7 | 97.8/75.5/73.6/89.5 | 99.0/**75.7/69.6**/68.2 | 98.5/43.2/50.0/89.7 | 97.8/65.0/68.3/94.2 | 97.3/75.4/72.4/93.3 | **99.2/88.3/84.1/**96.8 | 97.3/74.9/71.7/93.5 | 99.1/85.3/81.8/**97.0** |
| Cable | 56.8/19.9/30.9/42.4 | 90.4/35.9/39.5/65.3 | 87.0/29.1/37.7/38.0 | 83.6/26.7/35.2/76.9 | 84.4/30.4/38.0/78.9 | 75.1/26.7/34.1/71.4 | 96.8/65.1/69.1/91.7 | 76.9/27.5/35.8/74.5 | **98.5/73.1/74.4/94.5** |
| Capsule | 89.2/36.9/44.8/87.7 | 77.9/7.9/14.9/38.4 | **98.7**/56.1/56.3/78.6 | 98.6/40.4/46.8/89.7 | 98.7/43.3/48.4/89.9 | 98.7/43.3/48.9/96.2 | 98.7/60.0/**59.7/97.0** | 98.7/44.5/50.3/96.3 | 98.7/56.7/55.5/96.8 |
| Hazelnut | 70.2/26.3/34.3/54.0 | 95.2/50.0/52.9/80.2 | 98.5/77.8/73.5/56.7 | 98.5/61.5/62.3/91.6 | 98.6/62.4/62.7/92.4 | 98.2/63.5/61.8/89.9 | 99.4/**82.8**/76.5/97.0 | 98.2/60.1/60.3/90.7 | **99.5**/79.0/**76.9**/96.8 |
| Metal Nut | 47.6/25.5/27.4/72.7 | 78.4/38.1/41.7/61.4 | **99.3/96.8/90.8**/77.2 | 86.3/41.9/46.6/85.3 | 95.8/65.1/71.9/91.2 | 72.6/33.0/37.0/79.9 | 96.3/76.7/84.3/94.3 | 75.9/34.2/40.0/79.5 | 97.1/79.6/84.9/**95.4** |
| Pill | 58.9/20.6/28.5/76.5 | 91.4/38.0/46.8/82.6 | **98.3**/67.5/**70.2**/64.4 | 94.6/37.1/45.1/94.2 | 95.9/46.6/55.2/94.4 | 94.7/51.8/54.5/94.5 | 97.7/**72.6**/69.8/**97.5** | 97.3/70.5/67.6/97.1 | 97.3/70.5/67.6/97.1 |
| Screw | 87.6/32.5/41.0/77.3 | 92.9/5.1/12.7/75.3 | 98.1/47.9/49.2/57.8 | 98.0/19.9/29.4/91.8 | 98.2/22.7/32.1/92.0 | 99.3/41.4/46.8/96.8 | **99.6**/59.7/60.3/**97.6** | 99.2/42.8/49.2/96.5 | 99.4/56.0/55.0/97.3 |
| Toothbrush | 96.9/61.5/62.0/83.8 | 98.1/50.0/55.5/75.5 | 98.5/43.2/50.0/89.7 | 98.5/43.3/50.0/89.7 | 98.1/49.5/52.8/95.1 | 98.9/53.2/63.3/95.5 | 98.9/53.3/63.5/95.5 | 97.8/44.5/49.5/94.3 | **99.1/53.4/63.7/95.9** |
| Transistor | 67.2/43.2/49.5/59.9 | 85.9/50.1/49.4/61.3 | 95.2/54.9/**65.1**/35.3 | 90.7/50.8/53.8/83.2 | **96.7**/64.3/63.4/**93.1** | 82.7/31.2/35.4/64.6 | 93.4/60.1/58.6/76.6 | 85.3/33.8/39.2/67.3 | 95.3/**65.5**/62.4/83.3 |
| Zipper | 75.9/33.2/42.6/63.9 | 87.6/38.0/47.8/71.9 | 96.8/51.0/47.8/55.2 | 95.5/29.2/38.8/86.4 | 97.0/35.9/46.3/90.2 | 98.7/74.4/70.8/95.8 | **99.2/78.8/74.2/97.4** | 98.2/71.7/68.5/94.2 | 98.8/73.3/70.2/96.5 |
| **Mean** | 75.0/35.3/41.8/66.7 | 89.7/39.7/44.2/72.1 | 97.0/63.6/63.3/69.2 | 94.7/41.9/47.5/88.7 | 95.9/44.8/50.4/90.1 | 93.6/53.0/54.8/89.3 | 98.2/**68.0/68.5**/94.5 | 93.9/51.3/54.2/88.8 | **98.4**/67.4/67.7/**95.2** |

*Table 14.* Per-class performance on VISA dataset for multi-class anomaly detection with AUROC/AP/F1-max metrics. Noise Ratio 0

| | SoftPatch | InReach | FUN-AD | HVQ | HVQ + MeDS (ours) | Dinomaly | Dinomaly + MeDS (ours) | INP-Former | INP-Former + MeDS (ours) |
|---|---|---|---|---|---|---|---|---|---|
| Candle | **98.7**/**98.7**/**95.0** | 94.7/94.8/88.2 | 89.6/85.2/85.2 | 93.6/93.4/86.5 | 93.7/93.2/87.7 | 98.4/98.5/93.7 | 97.9/97.8/93.7 | 98.7/**98.7**/94.5 | 98.5/97.4/92.1 |
| Capsules | 74.6/82.5/80.3 | 55.1/69.9/77.2 | 66.8/77.4/76.9 | 69.7/83.1/77.2 | 70.1/83.5/76.9 | 98.5/99.0/96.5 | 98.0/98.4/96.1 | **98.8**/**99.2**/97.1 | 98.3/98.5/**97.1** |
| Cashew | 97.0/98.6/94.2 | 70.2/82.3/83.1 | 93.5/96.7/91.2 | 92.9/96.6/90.8 | 93.7/96.9/90.7 | 97.9/99.0/95.2 | 97.3/98.7/94.2 | 98.0/**99.1**/95.0 | **98.8**/98.9/**96.1** |
| Chewinggum | 99.1/99.6/97.5 | 82.1/92.2/81.5 | 93.8/97.4/91.6 | 98.2/99.1/95.5 | 98.8/99.5/96.1 | **99.6**/**99.8**/**98.0** | 99.5/99.7/97.5 | 98.6/99.4/96.5 | 98.7/99.4/96.0 |
| Fryum | 94.9/97.7/90.9 | 89.4/95.1/86.2 | 90.6/94.5/91.2 | 88.1/94.5/85.7 | 87.2/94.1/85.0 | 98.7/99.3/96.5 | 98.6/99.3/95.5 | 99.0/99.5/96.5 | **99.1**/**99.6**/**97.5** |
| Macaroni1 | 96.1/96.0/90.3 | 81.5/75.9/75.8 | 71.3/67.6/72.7 | 85.1/85.3/80.6 | 84.6/84.6/80.0 | **98.0**/**97.6**/**94.3** | 96.9/95.8/93.5 | 96.4/96.2/91.3 | 96.8/95.3/91.7 |
| Macaroni2 | 75.9/74.8/74.2 | 65.5/61.0/66.7 | 59.5/55.2/68.0 | 78.7/76.9/74.4 | 79.7/75.8/75.5 | **95.8**/**95.6**/**90.1** | 92.2/89.3/88.7 | 93.5/93.0/87.7 | 90.6/83.4/84.2 |
| Pcb1 | 97.1/96.8/94.1 | 91.5/91.3/84.1 | 86.6/82.6/84.1 | 92.8/91.4/87.0 | 91.6/90.7/86.5 | **98.8**/**98.6**/96.5 | 97.5/95.4/96.2 | 98.7/98.6/95.2 | 96.3/95.1/89.4 |
| Pcb2 | 96.0/96.8/91.7 | 92.5/92.8/85.8 | 75.3/67.7/75.1 | 90.7/91.0/85.7 | 90.0/90.4/83.8 | **98.6**/98.1/**96.5** | 97.4/93.3/96.0 | 98.3/**98.4**/96.0 | 96.8/95.4/94.0 |
| Pcb3 | 97.5/97.5/92.6 | 94.0/93.3/88.0 | 74.2/72.7/72.4 | 84.2/84.7/78.3 | 80.3/80.2/76.9 | **98.6**/98.7/**96.1** | 97.5/97.5/92.4 | 98.5/**98.7**/95.6 | 97.9/98.0/93.4 |
| Pcb4 | 99.7/99.8/98.0 | 94.5/95.0/88.7 | 93.2/87.6/93.5 | 99.4/99.2/97.5 | 98.3/98.0/96.1 | 99.8/99.8/98.0 | 99.5/99.5/97.5 | **99.9**/**99.9**/**98.5** | 99.6/99.6/98.5 |
| Pipe Fryum | **99.6**/**99.8**/**97.6** | 96.9/98.3/95.5 | 96.5/98.0/94.5 | 93.4/97.0/89.8 | 92.4/96.2/90.3 | 99.0/99.5/96.5 | 98.1/99.0/96.0 | 99.2/99.6/97.5 | 98.8/99.4/97.5 |
| **Mean** | 93.9/94.9/91.4 | 84.0/86.8/83.4 | 82.6/81.9/83.0 | 88.9/91.0/85.8 | 88.4/90.2/85.5 | **98.5**/**98.6**/**95.7** | 97.5/97.0/94.8 | 98.1/98.4/95.1 | 97.5/96.7/94.0 |

*Table 15.* Per-class performance on VISA dataset for multi-class anomaly localization with AUROC/AP/F1-max/AUPRO metrics. Noise Ratio 0

| | SoftPatch | InReach | FUN-AD | HVQ | HVQ + MeDS (ours) | Dinomaly | Dinomaly + MeDS (ours) | INP-Former | INP-Former + MeDS (ours) |
|---|---|---|---|---|---|---|---|---|---|
| Candle | 99.0/34.0/41.6/94.1 | 98.1/13.9/22.3/92.8 | 98.8/35.4/39.0/88.1 | 98.9/16.9/27.2/94.5 | 98.9/16.7/26.3/94.5 | 99.4/41.9/47.6/95.0 | 99.4/**43.7**/**47.6**/95.6 | 99.5/37.9/45.3/96.7 | **99.5**/37.9/44.3/**96.7** |
| Capsules | 98.6/54.6/55.8/80.2 | 94.1/10.0/19.2/61.5 | 92.6/6.8/13.5/33.3 | 97.9/43.8/47.8/71.8 | 97.8/38.0/42.9/69.2 | **99.6**/**64.8**/**66.7**/97.7 | 99.5/62.0/62.8/97.5 | 99.5/63.7/65.2/**98.3** | 99.5/56.8/58.4/97.8 |
| Cashew | 98.2/62.2/61.7/92.8 | 98.3/75.4/**74.8**/56.0 | 99.3/**78.6**/70.2/73.6 | **99.3**/61.0/63.5/89.3 | 99.1/58.4/60.6/89.7 | 95.8/60.4/59.7/92.7 | 96.2/60.5/58.4/**93.5** | 94.2/57.7/60.1/92.1 | 94.6/57.7/59.5/91.1 |
| Chewinggum | 98.2/52.5/50.7/81.0 | 94.4/13.4/28.3/44.3 | 97.9/66.5/63.4/73.3 | 98.8/41.7/43.0/76.3 | 98.9/50.1/48.8/77.8 | 96.6/**51.4**/53.2/93.5 | 96.5/51.2/52.8/93.7 | 96.2/45.7/49.8/94.3 | 96.2/45.3/49.5/**94.6** |
| Fryum | 92.3/45.4/49.5/80.5 | 95.8/46.6/52.5/74.9 | 94.7/35.8/40.3/63.0 | 97.4/48.3/53.6/82.4 | **97.5**/48.3/**53.7**/82.3 | 96.6/**51.4**/53.2/93.5 | 96.5/51.2/52.8/93.7 | 96.2/45.7/49.8/94.3 | 96.2/45.3/49.5/**94.6** |
| Macaroni1 | 99.3/30.5/32.7/95.5 | 97.7/6.0/10.3/91.8 | 96.5/1.2/4.1/26.2 | 98.5/6.7/16.1/89.4 | 98.5/6.3/15.4/90.8 | 99.7/**33.0**/**40.0**/96.8 | 99.6/27.8/38.5/96.6 | **99.7**/23.5/30.9/96.7 | 99.7/21.0/27.8/**96.9** |
| Macaroni2 | 97.8/16.3/26.2/88.9 | 96.6/2.3/4.0/89.3 | 95.6/1.1/4.4/14.4 | 97.5/2.7/7.4/88.6 | 97.6/2.7/7.6/88.6 | **99.8**/**24.4**/**35.9**/**98.8** | 99.7/23.8/35.2/98.6 | 99.8/14.7/25.1/98.1 | 99.8/13.7/23.8/98.1 |
| Pcb1 | **99.6**/86.9/**80.3**/91.4 | 99.3/82.8/78.0/86.5 | 97.3/26.3/37.2/54.6 | 99.2/69.0/63.1/87.4 | 98.9/46.4/51.2/84.9 | 99.5/86.2/79.5/94.7 | 99.4/84.5/78.7/94.5 | 99.5/**87.2**/80.1/**95.2** | 98.8/63.9/63.5/91.3 |
| Pcb2 | 98.3/25.3/32.3/84.7 | 97.5/8.6/16.3/83.0 | 92.4/9.6/19.6/34.0 | 97.6/8.1/15.5/80.5 | 97.4/7.3/14.3/79.8 | 98.1/**47.4**/49.7/91.3 | 98.2/45.4/**49.8**/91.7 | 98.0/37.4/41.5/91.7 | **98.4**/33.0/41.1/**91.8** |
| Pcb3 | 98.8/**46.5**/43.1/89.5 | 98.4/11.5/22.5/87.4 | 90.9/12.3/19.8/24.9 | 97.7/13.8/22.4/76.1 | 97.5/15.8/21.8/73.4 | 98.3/39.1/45.5/94.2 | 98.3/41.9/**46.7**/94.0 | 98.5/37.2/42.5/93.7 | **98.9**/37.8/43.1/93.5 |
| Pcb4 | 97.7/49.7/49.8/88.1 | 98.0/34.0/40.6/85.6 | 95.1/13.9/20.6/49.1 | 98.0/32.8/36.1/83.2 | 96.4/10.2/18.8/75.1 | 98.6/**50.3**/**53.1**/**94.4** | 98.2/40.8/44.2/92.7 | **98.6**/49.0/52.1/93.8 | 98.3/38.2/43.2/92.9 |
| Pipe Fryum | 98.9/66.1/62.5/94.0 | **99.4**/**71.6**/**72.0**/92.3 | 99.1/64.8/58.5/80.1 | 99.4/65.2/66.9/92.5 | **99.4**/63.0/65.9/92.2 | 99.1/61.8/63.7/95.3 | 99.1/60.4/62.5/95.8 | 99.1/57.9/63.3/**97.4** | 99.0/54.6/62.1/97.4 |
| **Mean** | 98.1/47.5/48.8/88.4 | 97.3/31.4/36.8/78.8 | 95.8/29.4/32.5/51.2 | 98.4/34.2/38.5/84.3 | 98.2/30.3/35.6/83.2 | **98.6**/**52.8**/**55.3**/94.4 | 98.6/51.4/53.9/94.4 | 98.5/47.6/51.5/**94.7** | 98.5/43.6/48.4/94.2 |

*Table 16.* Per-class performance on VISA dataset for multi-class anomaly detection with AUROC/AP/F1-max metrics. Noise Ratio 2

| | SoftPatch | InReach | FUN-AD | HVQ | HVQ + MeDS (ours) | Dinomaly | Dinomaly + MeDS (ours) | INP-Former | INP-Former + MeDS (ours) |
|---|---|---|---|---|---|---|---|---|---|
| Candle | **98.6**/**98.5**/**94.1** | 89.8/91.3/83.7 | 92.4/91.2/89.3 | 93.7/93.6/87.1 | 93.4/92.9/87.3 | 97.7/97.8/91.8 | 97.7/97.7/92.7 | 97.9/98.0/92.9 | 97.9/97.8/92.8 |
| Capsules | 73.0/82.4/77.6 | 51.4/67.7/77.2 | 88.8/89.0/89.7 | 67.9/81.9/76.9 | 71.1/83.8/77.5 | 97.0/98.1/93.7 | 97.9/98.4/96.1 | 97.5/98.3/94.7 | **98.4**/**98.6**/**97.1** |
| Cashew | 96.7/98.5/94.6 | 70.4/82.4/81.7 | 96.6/98.4/93.0 | 92.7/96.5/89.8 | 91.7/96.1/89.2 | 95.8/98.0/92.5 | 97.2/98.6/94.6 | 97.3/98.7/95.6 | **97.7**/**98.8**/**96.1** |
| Chewinggum | 99.1/99.6/**97.5** | 80.1/91.3/80.7 | 95.5/97.9/93.9 | 98.4/99.2/95.5 | 98.7/99.4/96.1 | 98.0/99.1/95.4 | **99.4**/**99.7**/97.4 | 98.4/99.3/96.0 | 98.3/99.3/95.4 |
| Fryum | 94.6/97.6/90.0 | 87.3/94.3/85.0 | 95.5/97.9/93.9 | 87.5/94.2/86.2 | 87.2/94.1/85.6 | 98.0/99.1/95.4 | 96.8/99.3/95.6 | 98.8/99.4/96.5 | **99.2**/**99.6**/**96.9** |
| Macaroni1 | 96.5/**96.4**/91.7 | 71.1/69.5/67.3 | 90.8/90.6/85.8 | 83.4/83.0/79.2 | 85.3/84.7/80.5 | 96.6/95.5/91.5 | **96.8**/95.7/**92.6** | 92.9/92.0/86.9 | 95.3/94.4/90.5 |
| Macaroni2 | 71.6/72.9/70.9 | 58.4/57.2/64.4 | 68.3/63.8/72.0 | 75.1/72.8/72.7 | 77.7/76.7/73.4 | **92.2**/**91.7**/86.7 | 91.1/87.6/**86.9** | 87.1/85.9/82.9 | 88.7/84.0/85.3 |
| Pcb1 | 97.4/97.1/94.5 | 88.3/88.9/81.0 | 91.0/90.7/85.5 | 91.8/90.0/85.7 | 90.3/88.9/85.0 | **98.2**/**98.0**/96.0 | 97.6/95.8/**96.6** | 97.9/97.5/95.2 | 96.7/96.3/92.4 |
| Pcb2 | 96.3/96.8/91.6 | 89.9/90.7/82.8 | 86.1/84.2/79.8 | 88.9/89.7/82.9 | 90.0/90.5/84.5 | **98.3**/**98.1**/96.0 | 96.9/94.0/**96.5** | 97.3/97.2/94.4 | 96.2/95.5/94.4 |
| Pcb3 | 97.7/97.7/92.9 | 84.7/88.3/81.5 | 87.7/89.2/81.8 | 88.7/89.2/83.4 | 81.6/81.1/77.2 | 97.8/97.9/**96.1** | 97.0/97.4/93.2 | **98.2**/**98.2**/95.5 | 97.6/98.0/93.7 |
| Pcb4 | 99.7/99.7/97.5 | 89.8/91.5/82.5 | 93.5/89.9/93.0 | 99.2/99.0/96.6 | 99.0/98.9/97.0 | 99.5/99.5/97.5 | 99.5/99.5/97.1 | **99.8**/**99.8**/**98.0** | 99.6/99.5/98.0 |
| Pipe Fryum | **99.5**/**99.8**/**97.6** | 91.1/96.1/92.3 | 99.2/99.5/98.0 | 93.6/97.0/91.2 | 93.1/96.8/91.5 | 97.6/98.8/94.6 | 97.7/98.8/96.0 | 98.3/99.2/95.0 | 98.7/99.4/97.5 |
| **Mean** | 93.4/94.7/90.9 | 79.3/84.1/80.0 | 90.7/90.3/88.0 | 87.9/90.0/85.2 | 88.3/90.3/85.4 | **97.3**/**97.7**/94.1 | 97.3/96.9/**94.6** | 96.8/97.0/93.6 | 97.0/96.8/94.2 |

*Table 17.* Per-class performance on VISA dataset for multi-class anomaly localization with AUROC/AP/F1-max/AUPRO metrics. Noise Ratio 2

| | SoftPatch | InReach | FUN-AD | HVQ | HVQ + MeDS (ours) | Dinomaly | Dinomaly + MeDS (ours) | INP-Former | INP-Former + MeDS (ours) |
|---|---|---|---|---|---|---|---|---|---|
| Candle | 98.9/34.0/41.7/93.6 | 95.1/11.2/19.6/89.8 | 98.9/39.6/42.0/78.7 | 99.0/17.4/27.6/94.3 | 98.9/16.7/26.1/94.3 | 99.4/41.5/**46.9**/95.1 | 99.4/**41.7**/46.8/95.3 | 99.5/33.4/45.0/96.6 | **99.5**/39.2/45.9/**96.6** |
| Capsules | 96.9/52.6/54.7/76.2 | 92.8/9.2/18.6/57.4 | 98.5/19.9/32.4/56.0 | 97.8/41.2/45.7/69.0 | 97.8/40.6/43.6/70.6 | **99.6**/**61.0**/**64.5**/97.1 | 99.5/59.3/61.9/96.8 | 99.5/59.3/60.5/97.8 | 99.4/56.0/58.2/**97.8** |
| Cashew | 98.3/63.3/62.5/90.8 | 98.0/75.5/**74.9**/49.4 | **99.5**/**82.5**/73.9/78.1 | 99.2/58.0/61.3/88.6 | 99.1/54.3/59.0/89.1 | 95.2/61.1/60.4/91.6 | 96.2/60.6/58.9/**92.5** | 92.9/57.9/59.5/90.0 | 94.3/56.9/58.9/91.2 |
| Chewinggum | 97.6/44.5/47.0/80.3 | 93.8/12.6/27.2/43.4 | **99.2**/**80.2**/**73.7**/69.8 | 98.7/36.7/39.8/76.1 | 98.8/46.1/45.2/76.6 | 98.9/61.0/60.9/85.6 | 99.1/74.0/69.2/**88.4** | 98.6/38.1/48.2/87.5 | 98.8/56.0/62.0/88.1 |
| Fryum | 92.0/45.0/49.7/78.3 | 95.8/44.9/51.7/70.8 | 94.9/51.7/50.9/52.9 | 97.3/46.7/53.7/82.1 | **97.5**/47.9/**53.9**/81.5 | 96.5/50.9/52.9/93.7 | 96.6/51.0/52.7/93.4 | 96.3/46.3/50.4/94.2 | 96.1/44.6/49.1/**94.4** |
| Macaroni1 | 99.0/**30.8**/33.1/95.6 | 90.6/5.2/10.0/84.0 | 99.2/6.8/16.1/71.4 | 98.4/7.5/17.6/89.6 | 98.4/6.4/14.3/89.8 | 99.7/28.8/37.4/**96.8** | 99.6/28.4/**37.9**/96.1 | 99.6/18.3/27.1/96.2 | **99.7**/20.6/27.5/96.6 |
| Macaroni2 | 92.3/15.3/25.1/83.5 | 90.9/1.2/2.9/84.4 | 97.5/2.9/10.0/39.3 | 97.5/2.4/7.7/87.2 | 97.5/2.8/8.3/88.5 | **99.7**/23.1/**35.2**/**98.7** | 99.7/**23.3**/34.6/98.5 | 99.7/12.5/22.8/97.4 | 99.7/12.8/22.9/98.1 |
| Pcb1 | 99.4/**87.8**/**81.1**/90.1 | 97.6/82.0/77.5/76.4 | 97.5/29.9/35.6/43.4 | 99.2/60.8/57.5/86.7 | 99.0/59.6/56.7/84.3 | **99.5**/85.1/77.0/94.8 | 99.4/85.6/78.7/94.9 | 99.4/87.1/78.9/94.7 | 99.0/72.0/71.0/92.0 |
| Pcb2 | 98.2/25.3/32.1/86.3 | 94.7/7.8/15.2/81.2 | 97.3/27.1/37.1/51.8 | 97.6/8.3/15.4/80.7 | 97.5/7.7/14.9/79.7 | 97.8/37.8/42.5/90.7 | 98.1/**42.0**/**48.8**/91.0 | 97.7/29.3/34.8/90.4 | **98.7**/31.0/39.3/**93.0** |
| Pcb3 | 98.5/**46.3**/43.0/87.0 | 97.1/12.0/23.1/77.8 | 95.4/21.6/30.3/61.1 | 97.8/14.2/23.1/75.8 | 97.5/15.9/22.3/73.8 | 98.1/43.5/**47.9**/94.0 | 98.4/43.4/47.4/**94.1** | 97.9/35.6/40.3/93.0 | **98.8**/36.7/42.7/93.3 |
| Pcb4 | 89.2/35.8/40.4/84.7 | 96.4/30.2/39.2/80.2 | 97.0/24.8/29.4/47.8 | 97.9/30.5/37.0/82.9 | 96.9/13.6/22.8/78.5 | **98.3**/**44.8**/**49.4**/**93.4** | 98.0/40.2/43.0/92.4 | 98.2/41.3/46.6/92.3 | 98.3/37.2/42.1/92.8 |
| Pipe Fryum | 98.6/**65.9**/62.3/94.8 | 97.9/62.7/**64.7**/90.6 | 98.8/63.5/58.2/75.2 | 99.3/58.0/61.1/93.1 | **99.3**/60.5/64.2/92.8 | 99.0/59.2/61.1/95.7 | 99.1/61.6/62.6/95.3 | 99.0/54.7/61.7/**97.4** | 99.0/54.4/62.6/97.3 |
| **Mean** | 96.4/45.6/47.7/86.8 | 95.0/29.5/35.4/73.8 | 97.9/37.0/40.4/61.8 | 98.3/31.8/37.3/83.8 | 98.2/31.0/35.9/83.3 | 98.5/49.8/53.0/93.9 | **98.6**/**50.9**/**53.5**/94.1 | 98.2/42.6/48.0/94.0 | 98.4/43.1/48.5/**94.3** |

*Table 18.* Per-class performance on VISA dataset for multi-class anomaly detection with AUROC/AP/F1-max metrics. Noise Ratio 5

| | SoftPatch | InReach | FUN-AD | HVQ | HVQ + MeDS (ours) | Dinomaly | Dinomaly + MeDS (ours) | INP-Former | INP-Former + MeDS (ours) |
|---|---|---|---|---|---|---|---|---|---|
| Candle | **98.3/98.4/93.8** | 82.8/87.7/80.8 | 92.9/92.7/88.9 | 94.1/93.8/88.1 | 93.6/93.2/87.4 | 96.5/96.6/89.9 | 97.7/97.7/92.8 | 97.3/97.2/91.9 | 97.9/97.6/93.3 |
| Capsules | 69.3/81.0/76.9 | 50.1/66.2/76.9 | 90.5/93.2/88.5 | 67.7/82.4/76.9 | 66.3/81.5/76.9 | 95.9/97.6/93.3 | 98.2/**98.6**/97.1 | 95.1/97.0/92.4 | **98.3/98.4/98.0** |
| Cashew | 97.0/98.6/94.6 | 67.1/82.4/81.7 | 94.8/97.4/91.2 | 91.4/95.9/90.1 | 92.6/96.4/89.4 | 94.3/97.3/90.7 | 97.7/98.9/95.2 | 97.0/98.5/95.0 | **97.7/98.9/95.5** |
| Chewinggum | 98.6/99.4/97.5 | 78.5/90.7/80.5 | 96.2/98.4/94.2 | 98.7/99.4/96.6 | 98.6/99.4/**97.5** | 97.9/99.1/95.3 | **99.4/99.7**/97.5 | 96.9/98.5/93.6 | 98.3/99.2/95.6 |
| Fryum | 95.1/97.8/90.8 | 82.1/92.0/81.8 | 96.4/98.4/93.7 | 85.9/93.7/85.3 | 86.3/93.8/85.0 | 97.7/98.8/94.3 | 98.6/99.3/95.4 | 98.0/99.1/94.5 | **98.8/99.4/96.0** |
| Macaroni1 | 96.0/**96.0**/90.9 | 66.1/64.3/64.6 | 92.0/92.5/86.1 | 82.0/81.8/78.8 | 84.2/84.4/80.2 | 95.8/94.4/92.2 | **97.0**/95.3/**93.5** | 90.2/88.7/85.8 | 95.1/94.0/90.7 |
| Macaroni2 | 67.9/69.4/70.3 | 46.9/49.3/64.4 | 80.1/76.5/76.7 | 73.3/69.9/73.5 | 72.9/71.1/71.9 | 89.8/**89.0**/83.8 | **91.5**/88.9/**88.8** | 83.3/80.8/79.0 | 88.7/83.2/85.0 |
| Pcb1 | 96.6/96.7/94.0 | 88.6/87.8/82.5 | 93.8/90.9/89.2 | 89.0/87.0/83.5 | 90.9/90.0/86.7 | 97.3/**96.9**/94.2 | **97.8**/96.6/**96.5** | 97.0/96.2/94.7 | 97.0/96.0/95.0 |
| Pcb2 | 95.3/96.3/91.6 | 83.0/86.1/76.6 | 90.8/88.8/86.1 | 89.1/89.8/82.7 | 90.1/90.5/85.1 | 97.3/**97.2**/93.0 | **97.5**/94.5/**96.0** | 96.1/95.9/90.2 | 96.6/94.7/94.4 |
| Pcb3 | 97.5/97.5/92.2 | 68.6/79.2/72.0 | 87.3/87.3/79.6 | 82.5/83.2/77.6 | 80.6/80.1/77.1 | 96.4/96.6/91.8 | 96.9/97.2/93.1 | 97.1/97.2/93.0 | **97.7/97.8/93.4** |
| Pcb4 | 98.5/98.7/94.3 | 86.7/89.9/81.4 | 96.0/93.5/93.8 | 98.7/98.5/95.2 | 98.6/98.5/96.1 | 99.1/99.0/97.0 | 99.3/99.3/97.1 | 99.6/99.6/97.5 | **99.8/99.7/98.5** |
| Pipe Fryum | **99.4/99.7**/97.6 | 80.2/92.1/85.2 | 98.5/99.1/**98.0** | 93.1/96.8/91.1 | 93.1/96.8/91.7 | 94.8/97.2/92.3 | 98.4/99.2/96.5 | 96.0/98.1/92.0 | 98.7/99.4/97.5 |
| **Mean** | 92.5/94.1/90.4 | 73.4/80.6/77.4 | 92.4/92.4/88.9 | 87.1/89.3/84.9 | 87.3/89.6/85.4 | 96.1/96.6/92.3 | **97.5/97.1/94.9** | 95.3/95.6/91.6 | 97.0/96.5/94.4 |

*Table 19.* Per-class performance on VISA dataset for multi-class anomaly localization with AUROC/AP/F1-max/AUPRO metrics. Noise Ratio 5

| | SoftPatch | InReach | FUN-AD | HVQ | HVQ + MeDS (ours) | Dinomaly | Dinomaly + MeDS (ours) | INP-Former | INP-Former + MeDS (ours) |
|---|---|---|---|---|---|---|---|---|---|
| Candle | 98.8/35.3/42.2/92.2 | 90.0/11.2/21.3/84.0 | 98.7/39.1/42.8/78.0 | 99.0/17.6/27.3/94.7 | 98.9/16.7/26.4/94.0 | 99.3/37.9/45.9/94.9 | 99.4/**41.9/47.5**/95.1 | 99.5/37.8/46.1/96.4 | **99.5**/34.1/43.2/**96.7** |
| Capsules | 93.2/47.4/51.1/71.2 | 90.7/7.9/16.4/53.8 | 99.2/41.5/46.4/68.2 | 97.9/43.9/46.2/73.1 | 97.9/38.7/43.4/70.0 | 99.6/**61.0/64.4**/97.3 | 99.5/60.6/62.1/97.0 | 99.3/52.6/57.8/97.5 | 99.4/54.5/55.5/**97.7** |
| Cashew | 96.8/63.1/62.8/88.7 | 94.4/70.9/73.2/47.0 | **99.7/85.3/76.4**/76.8 | 99.1/53.6/58.9/89.3 | 99.1/54.0/59.1/89.3 | 94.2/59.7/59.8/88.4 | 96.5/61.3/59.6/**93.5** | 91.2/53.6/55.6/87.3 | 93.8/56.6/58.8/90.5 |
| Chewinggum | 96.9/38.2/42.1/80.8 | 93.0/13.3/28.6/36.9 | 98.9/**84.1/78.0**/73.5 | 98.8/42.9/45.3/77.7 | 99.0/52.4/50.1/78.0 | 96.5/51.3/53.0/93.5 | 99.1/72.7/68.6/87.4 | 98.5/46.8/52.6/86.0 | 98.8/59.7/64.2/**87.5** |
| Fryum | 87.5/44.7/50.1/75.9 | 94.7/44.4/50.6/70.5 | 95.3/40.5/42.6/70.5 | 97.3/45.5/52.7/82.8 | **97.4**/48.4/**54.0**/80.9 | 96.5/**51.3**/53.0/93.5 | 96.6/50.6/52.7/93.9 | 96.2/47.2/50.8/94.2 | 96.2/44.9/49.2/**94.3** |
| Macaroni1 | 98.1/**30.1**/33.0/93.1 | 89.0/3.2/8.4/79.1 | 99.4/9.9/20.7/63.5 | 98.4/6.1/14.7/89.5 | 98.4/7.7/16.9/89.9 | 99.6/24.4/36.1/96.6 | 99.6/28.4/**38.3**/96.4 | 99.5/13.7/26.1/95.5 | **99.7**/21.0/27.6/**96.7** |
| Macaroni2 | 85.3/13.8/23.9/74.5 | 84.8/0.5/3.5/64.2 | 98.5/5.0/13.8/49.1 | 97.1/2.4/7.9/86.9 | 97.2/2.7/7.8/87.6 | 99.7/20.7/32.7/**98.7** | 99.7/**23.1/34.4**/98.6 | 99.6/11.0/21.4/96.3 | **99.7**/12.8/22.8/98.1 |
| Pcb1 | 99.1/**87.0/81.0**/87.4 | 94.7/70.2/68.0/56.7 | 99.4/78.5/72.6/60.2 | 99.2/55.5/58.6/87.0 | 99.1/57.6/54.9/85.9 | 99.1/53.4/59.2/94.9 | 99.4/86.4/79.3/94.5 | 99.2/55.7/62.0/94.4 | 99.2/81.1/75.9/93.4 |
| Pcb2 | 95.8/24.5/31.9/79.8 | 91.7/8.7/17.5/72.4 | 96.4/20.6/33.5/58.2 | 97.6/10.6/17.3/80.5 | 97.5/8.2/15.1/78.8 | 97.8/44.2/49.2/91.1 | 98.2/**44.9/51.2**/90.7 | 97.6/32.6/40.3/90.6 | **98.6**/28.5/38.5/**92.5** |
| Pcb3 | 98.2/**48.2**/45.1/84.7 | 94.0/10.5/22.5/64.6 | 94.8/29.9/36.2/35.5 | 97.7/13.3/21.8/75.6 | 97.5/16.7/21.4/73.8 | 97.9/36.8/44.2/**94.0** | 98.5/42.7/**47.9**/93.9 | 98.0/29.3/37.2/93.0 | **98.8**/33.3/41.6/93.5 |
| Pcb4 | 86.6/38.1/42.2/73.3 | 91.0/27.0/37.9/68.2 | 97.5/33.9/37.5/34.9 | 98.1/31.2/38.1/83.8 | 96.7/12.1/21.4/77.8 | 98.2/**43.8/48.4/93.0** | 98.1/40.7/43.9/92.5 | 97.8/39.7/45.4/90.6 | **98.3**/39.4/42.5/92.7 |
| Pipe Fryum | 98.5/66.3/62.7/92.0 | 98.1/65.3/65.8/87.9 | 99.3/**76.8/68.5**/86.6 | 99.2/56.4/60.1/93.1 | **99.4**/61.4/64.6/93.2 | 99.0/62.8/61.7/95.0 | 99.1/61.5/63.7/95.8 | 99.0/58.7/62.5/**97.2** | 99.0/56.1/62.0/**97.2** |
| **Mean** | 94.6/44.7/47.4/82.8 | 92.2/27.8/34.5/65.5 | 98.1/45.4/47.4/62.9 | 98.3/31.6/37.4/84.5 | 98.2/31.4/36.2/83.3 | 98.3/46.9/51.8/93.6 | **98.6/51.3/54.1**/94.1 | 98.0/39.9/46.5/93.2 | 98.4/43.5/48.5/**94.2** |

*Table 20.* Per-class performance on VISA dataset for multi-class anomaly detection with AUROC/AP/F1-max metrics. Noise Ratio 10

| | SoftPatch | InReach | FUN-AD | HVQ | HVQ + MeDS (ours) | Dinomaly | Dinomaly + MeDS (ours) | INP-Former | INP-Former + MeDS (ours) |
|---|---|---|---|---|---|---|---|---|---|
| Candle | 97.7/**97.9**/92.2 | 66.0/75.9/67.7 | 93.5/92.9/88.9 | 93.4/93.0/87.0 | 93.2/92.7/87.6 | 94.6/94.7/87.0 | **97.9**/97.8/**93.2** | 96.6/96.6/91.6 | 97.8/97.6/92.5 |
| Capsules | 65.7/79.8/76.9 | 43.7/62.2/76.9 | 92.5/94.2/90.0 | 67.4/81.0/76.9 | 69.3/82.9/76.9 | 95.2/97.0/91.9 | **98.1/98.5**/96.5 | 95.5/97.2/92.4 | 97.7/98.2/**97.1** |
| Cashew | 95.2/97.7/93.1 | 64.2/77.5/81.4 | 94.5/97.3/91.2 | 92.4/96.1/90.5 | 92.5/96.4/89.7 | 90.4/95.3/89.7 | 97.1/**98.6**/94.4 | 95.1/97.7/92.0 | **97.2**/98.6/**96.0** |
| Chewinggum | 95.7/97.9/94.8 | 81.6/91.8/83.0 | 97.8/99.4/96.4 | 97.5/98.9/96.6 | 98.6/99.4/96.6 | 94.3/97.5/90.8 | **99.3/99.7/97.0** | 96.7/98.4/94.5 | 98.3/99.3/95.5 |
| Fryum | 93.8/97.3/90.4 | 75.9/89.7/80.0 | 97.8/99.0/95.0 | 85.9/93.5/85.2 | 86.1/93.8/84.0 | 96.9/98.5/94.1 | 98.4/99.3/95.4 | 97.1/98.6/93.0 | **99.0/99.5/96.0** |
| Macaroni1 | 94.6/94.0/89.5 | 50.0/53.9/63.0 | 95.2/95.6/87.4 | 81.2/81.2/77.9 | 84.3/85.2/80.4 | 92.1/88.5/88.0 | **97.0/95.6/93.1** | 89.8/87.3/84.8 | 94.9/93.8/90.3 |
| Macaroni2 | 63.7/66.4/66.9 | 31.9/41.0/64.4 | 81.1/83.0/75.6 | 69.6/69.1/68.6 | 69.8/72.0/67.8 | 82.5/79.8/80.7 | **92.2/87.9/88.8** | 80.6/78.7/79.0 | 86.3/81.6/82.8 |
| Pcb1 | 95.1/93.7/91.9 | 83.7/82.8/80.9 | 97.0/**95.9**/92.9 | 86.7/83.5/82.0 | 89.8/88.8/85.0 | 95.9/94.3/93.1 | **97.4**/95.2/**96.2** | 95.8/93.1/93.7 | 95.4/95.1/89.4 |
| Pcb2 | 94.0/95.5/91.6 | 75.0/79.1/71.3 | 96.0/**96.4**/91.0 | 87.4/87.6/80.2 | 88.6/89.4/84.0 | 95.3/94.6/91.1 | **96.9**/93.4/**96.0** | 95.1/93.9/90.6 | 96.4/95.1/93.3 |
| Pcb3 | 96.9/97.4/90.1 | 48.9/65.7/66.5 | 94.9/95.7/87.5 | 81.2/82.0/76.1 | 81.1/80.6/77.4 | 94.7/94.6/91.5 | 96.8/97.5/94.1 | 96.0/95.7/92.5 | **97.2/97.9/93.3** |
| Pcb4 | 97.2/97.4/91.1 | 76.1/79.2/72.3 | 97.2/96.1/93.5 | 97.9/97.4/94.7 | 98.6/98.4/96.1 | 98.3/97.8/96.5 | 99.5/99.5/97.5 | 99.4/99.3/97.5 | **99.7/99.7/98.5** |
| Pipe Fryum | **99.5/99.8/97.6** | 72.8/88.8/80.9 | 99.2/99.6/**99.0** | 92.5/96.5/90.8 | 94.2/97.4/91.5 | 92.6/96.2/90.3 | 98.4/99.1/96.5 | 95.7/97.8/92.3 | 98.5/99.2/98.0 |
| **Mean** | 90.7/92.9/88.9 | 64.2/74.0/74.0 | 94.8/95.4/90.7 | 86.1/88.3/83.8 | 87.2/89.7/84.8 | 93.6/94.1/90.4 | **97.4/97.0/94.9** | 94.5/94.5/91.2 | 96.5/96.3/93.6 |

*Table 21.* Per-class performance on VISA dataset for multi-class anomaly localization with AUROC/AP/F1-max/AUPRO metrics. Noise Ratio 10

| | SoftPatch | InReach | FUN-AD | HVQ | HVQ + MeDS (ours) | Dinomaly | Dinomaly + MeDS (ours) | INP-Former | INP-Former + MeDS (ours) |
|---|---|---|---|---|---|---|---|---|---|
| Candle | 98.5/34.7/41.5/90.9 | 84.3/7.3/17.3/77.7 | 98.5/37.5/40.9/80.9 | 99.0/17.7/27.1/94.3 | 98.9/16.9/26.3/94.5 | 99.3/38.8/44.9/94.2 | 99.4/**42.1/47.6**/95.4 | 99.5/38.0/46.5/96.3 | **99.5**/35.6/43.8/**96.7** |
| Capsules | 90.3/43.2/48.9/59.0 | 87.6/5.9/13.6/48.7 | 99.1/50.5/51.4/66.7 | 98.0/41.7/46.5/70.6 | 97.8/38.7/43.6/68.9 | 99.5/**64.5/64.6**/97.3 | 99.5/59.5/62.2/96.9 | 99.3/54.4/58.4/**97.6** | 99.4/55.2/57.1/97.5 |
| Cashew | 96.9/66.2/64.9/83.2 | 80.4/51.2/59.9/33.5 | **99.3/77.7/69.2**/80.9 | 98.7/48.5/54.5/88.8 | 99.0/52.9/58.7/87.9 | 91.5/54.1/55.1/81.5 | 96.4/60.2/58.8/**93.3** | 92.6/59.5/58.9/89.1 | 94.0/56.5/58.9/90.4 |
| Chewinggum | 79.6/17.2/31.7/72.2 | 91.9/19.1/35.4/35.5 | **99.3/82.6/75.8**/71.8 | 99.0/53.9/54.0/77.0 | 98.9/51.6/49.5/78.4 | 98.8/64.6/65.3/84.5 | 99.2/74.4/69.7/**89.4** | 98.4/47.3/49.9/86.6 | 98.8/59.7/64.2/88.4 |
| Fryum | 79.9/40.3/48.4/74.8 | 93.2/41.3/47.5/62.5 | 94.6/40.4/42.3/80.5 | 97.3/45.8/51.9/82.6 | **97.4**/48.0/**54.0**/82.2 | 96.5/50.7/51.5/93.6 | 96.5/50.8/52.4/93.9 | 96.0/44.0/49.3/93.7 | 96.2/45.1/49.3/**94.3** |
| Macaroni1 | 95.3/25.7/31.2/87.8 | 77.8/2.6/7.3/57.3 | 99.5/11.6/23.6/69.7 | 98.4/6.4/15.2/89.9 | 98.3/6.6/15.1/89.8 | 99.5/22.9/32.1/96.0 | 99.6/**29.5/38.1**/96.7 | 99.6/17.6/26.5/95.8 | **99.7**/18.5/26.0/96.5 |
| Macaroni2 | 71.7/11.2/22.5/65.5 | 72.6/0.1/1.4/36.3 | 98.5/7.0/18.1/45.3 | 97.1/2.7/8.7/86.6 | 97.2/2.7/8.8/87.9 | 99.6/17.4/28.3/98.1 | **99.8/24.1/35.4/98.8** | 99.5/10.9/20.1/96.1 | **99.7**/12.8/22.6/97.7 |
| Pcb1 | 59.5/26.0/34.2/75.2 | 92.3/65.1/65.8/40.0 | 99.2/70.3/62.2/87.2 | 99.2/53.8/58.1/86.8 | 99.1/65.5/60.7/86.2 | 99.0/51.5/56.4/94.7 | **99.4/86.7/79.7/94.8** | 99.1/59.1/61.0/94.3 | 98.7/64.9/63.1/91.0 |
| Pcb2 | 89.5/35.7/44.9/73.9 | 85.3/8.9/19.3/57.3 | 97.1/22.6/37.6/50.3 | 97.5/14.6/24.0/80.1 | 97.3/8.2/15.1/79.0 | 97.7/**48.3/54.1**/90.3 | 98.2/45.0/51.9/91.2 | 97.2/34.5/42.8/90.2 | **98.3**/30.6/39.9/**91.5** |
| Pcb3 | 92.3/**55.1/53.9**/83.4 | 92.8/11.3/21.9/47.7 | 96.7/48.0/50.6/50.1 | 97.8/29.8/31.9/76.4 | 97.4/15.7/21.8/74.3 | 97.5/45.9/50.7/94.0 | 98.5/43.8/48.5/**94.2** | 97.7/35.8/42.6/92.9 | **98.7**/38.9/42.2/93.4 |
| Pcb4 | 54.1/18.1/27.9/51.9 | 86.7/18.0/29.4/44.2 | 97.2/31.3/36.4/30.2 | 97.8/27.7/35.7/82.5 | 97.1/14.9/24.0/79.2 | 97.9/40.7/**46.1**/92.3 | 98.1/**41.6**/44.8/92.2 | 97.4/36.6/43.7/88.6 | **98.3**/38.7/43.4/**92.9** |
| Pipe Fryum | 98.6/70.7/64.2/82.5 | 95.4/58.9/62.5/74.9 | **99.5/78.5/71.5**/84.2 | 99.1/56.3/60.4/92.3 | 99.4/62.2/64.7/92.5 | 98.8/61.2/59.8/94.5 | 99.1/59.8/62.2/95.9 | 99.1/68.7/64.8/**97.2** | 99.0/55.8/62.1/**97.2** |
| **Mean** | 83.8/37.0/42.9/75.0 | 86.7/24.1/31.8/51.3 | 98.2/46.5/48.3/66.5 | 98.2/33.2/39.0/84.0 | 98.2/32.0/36.9/83.4 | 98.0/46.7/50.7/92.6 | **98.6/51.5/54.3/94.4** | 97.9/42.2/47.0/93.2 | 98.4/42.7/47.6/94.0 |

*Table 22.* Per-class performance on Real-IAD dataset for single-class anomaly detection with AUROC/AP/F1-max metrics. Noise Ratio 0

| | SoftPatch | PatchCore | RD | UniAD | Dinomaly | Dinomaly + MeDS (ours) |
|---|---|---|---|---|---|---|
| audiojack | 89.5/**83.7/77.0** | 89.0/-/- | 86.6/-/- | 87.1/-/- | **91.0**/82.4/73.0 | 90.1/78.2/66.7 |
| bottle_cap | **97.4/95.9/89.9** | 96.8/-/- | 94.9/-/- | 94.4/-/- | 95.6/92.0/82.3 | 95.8/92.1/82.5 |
| button_battery | 87.3/**81.7**/75.7 | **87.5**/-/- | 86.8/-/- | 80.7/-/- | 84.6/75.4/71.8 | 86.8/77.3/**75.8** |
| end_cap | 88.3/86.7/78.5 | 85.8/-/- | 76.1/-/- | 80.7/-/- | **90.6/87.5/80.0** | 90.0/85.5/79.6 |
| eraser | **95.0/90.6/83.0** | 94.9/-/- | 92.7/-/- | 91.6/-/- | 94.4/85.6/80.0 | 94.0/85.3/78.5 |
| fire_hood | 86.0/76.7/67.7 | 85.4/-/- | 82.1/-/- | 79.9/-/- | **90.6/82.5/73.8** | 89.9/81.5/72.3 |
| mint | 76.5/66.4/59.0 | 80.2/-/- | 74.5/-/- | 70.0/-/- | 85.4/74.0/68.1 | **86.2/74.6/69.5** |
| mounts | 90.7/72.4/**73.5** | **91.1**/-/- | 88.8/-/- | 89.5/-/- | 90.2/**76.3**/70.5 | 90.2/73.5/70.6 |
| pcb | 94.9/95.0/87.0 | 94.1/-/- | 94.0/-/- | 85.8/-/- | **95.4**/86.0/78.8 | **95.4/95.7/87.9** |
| phone_battery | 93.5/88.0/79.2 | 92.7/-/- | 91.4/-/- | 87.6/-/- | **95.3**/80.7/72.7 | 94.1/**87.5/79.0** |
| plastic_nut | **95.2/89.0**/82.6 | 94.7/-/- | 92.5/-/- | 82.7/-/- | 94.8/86.8/**83.6** | 94.1/84.8/81.5 |
| plastic_plug | 92.4/**85.8/76.9** | 92.9/-/- | **93.7**/-/- | 83.5/-/- | 88.9/68.4/74.3 | 88.2/65.6/73.7 |
| porcelain_doll | 89.6/81.6/72.5 | 91.0/-/- | 89.4/-/- | 86.8/-/- | **92.2/82.7/72.6** | 91.0/79.0/69.1 |
| regulator | 88.0/**77.0/67.9** | 86.4/-/- | **89.4**/-/- | 61.0/-/- | 83.3/67.7/58.7 | 86.8/70.1/61.5 |
| rolled_strip_base | **99.4/99.5/97.0** | 99.0/-/- | 97.0/-/- | 98.5/-/- | 99.0/99.0/96.1 | 98.8/98.8/95.9 |
| sim_card_set | 98.3/**97.6/92.0** | **98.7**/-/- | 95.5/-/- | 94.1/-/- | 97.7/86.7/81.3 | 98.0/96.8/91.1 |
| switch | 96.1/95.3/88.8 | 89.6/-/- | 91.8/-/- | 88.3/-/- | **98.2**/88.0/82.5 | **98.2/97.7/91.7** |
| tape | 98.6/97.2/91.8 | **98.8**/-/- | 98.2/-/- | 98.2/-/- | 98.4/87.0/81.2 | 98.4/96.7/89.5 |
| terminalblock | **97.0**/96.8/**90.0** | 95.2/-/- | 94.6/-/- | 89.6/-/- | 97.5/87.1/81.0 | 96.5/**95.2**/89.0 |
| toothbrush | **89.2/85.8/79.1** | 88.7/-/- | 83.2/-/- | 83.7/-/- | 88.2/85.2/77.3 | 87.6/82.6/78.1 |
| toy | **91.1/87.4**/81.7 | 90.5/-/- | 86.5/-/- | 69.0/-/- | 90.2/**87.4**/81.9 | 90.1/85.1/**82.1** |
| toy_brick | 82.5/**70.3/65.0** | **83.5**/-/- | 78.7/-/- | 77.6/-/- | 78.7/64.3/60.2 | 79.9/65.2/60.9 |
| transistor1 | 98.0/**97.7/93.1** | **98.4**/-/- | 98.2/-/- | 95.5/-/- | 97.4/87.1/82.5 | 97.5/96.9/91.9 |
| usb | 94.7/90.4/83.0 | 93.9/-/- | 93.2/-/- | 84.4/-/- | **95.5/91.0/84.8** | 95.4/90.4/84.2 |
| usb_adaptor | 87.8/**78.7**/71.5 | 84.2/-/- | 75.7/-/- | 81.3/-/- | **90.9**/78.6/**74.2** | 90.2/76.0/72.6 |
| u_block | **93.7/85.9/79.0** | 93.7/-/- | 92.6/-/- | 91.2/-/- | 93.0/83.1/74.0 | 92.0/79.8/73.0 |
| vcpill | 92.3/89.8/80.9 | 92.4/-/- | 89.9/-/- | 87.7/-/- | 94.8/83.0/75.4 | **95.1/93.0/84.5** |
| wooden_beads | 89.6/**84.3**/74.9 | 89.7/-/- | 88.9/-/- | 81.6/-/- | **90.6**/83.9/**77.6** | 90.5/83.0/76.5 |
| woodstick | 80.0/61.2/59.0 | 81.1/-/- | 84.4/-/- | 82.9/-/- | 91.6/78.0/73.3 | **91.8/78.3/73.8** |
| zipper | 97.7/**98.0**/92.5 | 97.9/-/- | 97.5/-/- | **98.3**/-/- | 97.6/96.7/**94.0** | 97.5/96.5/93.8 |
| **Mean** | 91.7/**86.2/79.7** | 91.3/-/- | 89.3/-/- | 85.4/-/- | **92.4**/83.1/77.2 | 92.3/84.8/79.2 |

*Table 23.* Per-class performance on Real-IAD dataset for single-class anomaly localization with AUROC/AP/F1-max/AUPRO metrics. Noise Ratio 0

| | SoftPatch | PatchCore | RD | UniAD | Dinomaly | Dinomaly + MeDS (ours) |
|---|---|---|---|---|---|---|
| audiojack | 98.5/41.0/46.2/90.0 | -/-/-/89.7 | -/-/-/90.9 | -/-/-/86.6 | **99.4/47.1/53.5/96.0** | 99.1/44.3/52.9/94.5 |
| bottle_cap | 99.6/32.3/36.3/97.6 | -/-/-/94.5 | -/-/-/98.0 | -/-/-/96.5 | **99.8**/41.0/42.7/**98.8** | **99.8**/33.6/37.3/98.6 |
| button_battery | 98.6/**43.8/44.1**/87.5 | -/-/-/90.0 | -/-/-/91.9 | -/-/-/76.0 | **99.0**/19.8/38.0/**92.6** | **99.0**/20.4/38.0/92.3 |
| end_cap | 97.3/17.3/27.3/91.7 | -/-/-/92.4 | -/-/-/93.7 | -/-/-/84.3 | 99.1/**29.6/33.1**/97.0 | **99.2**/28.9/32.4/**96.9** |
| eraser | 99.6/37.3/42.2/96.5 | -/-/-/98.1 | -/-/-/97.8 | -/-/-/94.8 | **99.8/44.3**/45.9/**98.8** | **99.8**/40.4/**46.5**/98.2 |
| fire_hood | 98.2/30.7/37.0/88.0 | -/-/-/90.0 | -/-/-/93.3 | -/-/-/86.4 | **99.7/45.4/49.3**/97.1 | **99.7**/45.2/49.0/**97.2** |
| mint | 97.0/15.8/25.4/79.4 | -/-/-/83.2 | -/-/-/**86.4** | -/-/-/57.8 | **98.0**/16.9/29.7/**86.4** | 97.8/16.0/29.0/86.2 |
| mounts | 98.2/33.8/37.2/84.7 | -/-/-/90.1 | -/-/-/94.4 | -/-/-/93.5 | **99.7/44.6/48.3/97.3** | 99.6/41.8/46.0/96.4 |
| pcb | 99.4/51.3/52.4/94.3 | -/-/-/94.1 | -/-/-/94.2 | -/-/-/84.1 | 98.7/51.9/55.1/97.5 | **99.8/59.6/60.1/97.9** |
| phone_battery | 99.3/43.0/45.3/95.4 | -/-/-/94.2 | -/-/-/97.7 | -/-/-/91.5 | 98.8/46.5/50.3/**98.6** | **99.8/52.3/55.7**/98.0 |
| plastic_nut | 99.4/29.3/34.4/96.4 | -/-/-/96.1 | -/-/-/96.9 | -/-/-/89.4 | **99.8**/38.0/42.5/98.3 | **99.8**/36.4/**42.9/98.3** |
| plastic_plug | **98.8/30.5/33.7**/92.5 | -/-/-/94.0 | -/-/-/**96.3** | -/-/-/87.9 | 98.5/16.4/25.8/91.7 | 98.2/15.0/22.5/90.7 |
| porcelain_doll | 98.9/26.5/33.5/92.9 | -/-/-/96.1 | -/-/-/96.9 | -/-/-/93.4 | **99.7/37.6/43.1/98.4** | 99.5/35.4/42.6/97.5 |
| regulator | 99.2/24.6/34.5/94.9 | -/-/-/96.9 | -/-/-/**98.3** | -/-/-/74.3 | **99.7/46.3/49.2**/97.7 | **99.7**/43.7/47.5/97.3 |
| rolled_strip_base | 99.7/30.2/38.9/98.6 | -/-/-/**98.9** | -/-/-/98.7 | -/-/-/98.2 | **99.8**/37.6/46.1/**98.9** | **99.8**/33.1/44.5/98.8 |
| sim_card_set | 98.6/43.8/47.1/91.0 | -/-/-/91.7 | -/-/-/94.4 | -/-/-/85.9 | 98.7/49.7/50.8/97.8 | **99.6/53.3/54.2**/96.3 |
| switch | **99.2**/61.0/59.4/94.8 | -/-/-/93.8 | -/-/-/94.7 | -/-/-/92.3 | 97.5/65.1/63.0/96.9 | 98.9/**71.6/68.1/97.6** |
| tape | 99.7/43.4/46.5/98.6 | -/-/-/98.0 | -/-/-/**99.3** | -/-/-/98.6 | 98.8/41.9/47.4/99.1 | **99.8/50.5/53.5/99.0** |
| terminalblock | 99.6/36.5/41.2/96.8 | -/-/-/97.3 | -/-/-/**98.6** | -/-/-/94.1 | 98.6/38.9/43.0/**98.4** | **99.7/46.9/49.1**/98.3 |
| toothbrush | **97.8/43.0/46.0**/90.6 | -/-/-/**92.6** | -/-/-/92.1 | -/-/-/88.2 | 96.7/32.4/39.8/89.1 | 96.6/29.6/37.9/88.7 |
| toy | **97.7**/17.0/22.9/90.4 | -/-/-/92.3 | -/-/-/93.2 | -/-/-/77.4 | 96.4/**26.2/33.8/93.5** | 96.9/24.6/33.1/93.5 |
| toy_brick | 96.4/21.6/29.8/83.9 | -/-/-/84.8 | -/-/-/**89.0** | -/-/-/82.5 | **97.8**/24.7/31.3/84.9 | 97.7/**25.1/31.5**/83.8 |
| transistor1 | 99.5/46.0/45.4/97.2 | -/-/-/**98.0** | -/-/-/97.8 | -/-/-/94.2 | 98.5/48.7/49.5/97.8 | **99.6/56.9/54.5**/97.7 |
| usb | **99.6**/42.1/44.0/97.4 | -/-/-/96.4 | -/-/-/97.4 | -/-/-/88.4 | 99.4/**48.9**/49.3/**98.4** | 99.5/47.1/**49.7/98.4** |
| usb_adaptor | 97.8/18.2/28.2/84.8 | -/-/-/92.0 | -/-/-/90.6 | -/-/-/84.1 | **99.5/25.6/31.9/96.0** | 99.3/21.5/29.4/93.0 |
| u_block | 99.4/28.4/34.0/95.1 | -/-/-/97.0 | -/-/-/**98.1** | -/-/-/94.0 | **99.7/39.7/45.8**/98.1 | **99.7**/38.1/43.7/97.7 |
| vcpill | 98.9/55.9/57.7/90.2 | -/-/-/94.6 | -/-/-/95.7 | -/-/-/93.9 | 98.5/59.0/62.4/96.4 | **99.6/68.1/68.0/96.6** |
| wooden_beads | 97.3/32.6/37.4/84.6 | -/-/-/89.2 | -/-/-/91.6 | -/-/-/85.6 | **99.3/48.1/50.4/95.1** | 99.2/45.0/48.6/94.2 |
| woodstick | 92.2/19.7/27.6/70.0 | -/-/-/71.0 | -/-/-/92.8 | -/-/-/78.4 | 99.2/**43.1**/48.6/95.4 | **99.3**/41.5/**48.2/95.4** |
| zipper | **99.2/64.3/59.5**/96.0 | -/-/-/91.0 | -/-/-/**98.4** | -/-/-/96.2 | **99.2**/46.3/50.4/**97.3** | 99.1/46.1/50.5/97.0 |
| **Mean** | 98.4/35.4/39.8.6/91.3 | -/-/-/92.6 | -/-/-/95.0 | -/-/-/87.6 | 98.9/40.0/45.0/**96.0** | **99.2/40.4/45.6**/95.5 |

*Table 24.* Per-class performance on Real-IAD dataset for single-class anomaly detection with AUROC/AP/F1-max metrics. Noise Ratio 10

| | SoftPatch | PatchCore | RD | UniAD | Dinomaly | Dinomaly + MeDS (ours) |
|---|---|---|---|---|---|---|
| audiojack | 87.9/82.5/76.8 | 87.1/-/- | 85.8/-/- | 85.2/-/- | 88.5/78.5/70.5 | **91.9/83.2/73.6** |
| bottle_cap | **97.4/96.0/90.0** | 96.7/-/- | 95.6/-/- | 93.9/-/- | 93.0/86.1/76.5 | 95.7/92.1/82.0 |
| button_battery | 83.7/**78.8**/71.4 | 83.8/-/- | 84.9/-/- | 75.2/-/- | 81.0/72.7/67.7 | **86.7**/77.1/**75.3** |
| end_cap | 87.6/85.2/76.7 | 83.5/-/- | 79.1/-/- | 79.0/-/- | 89.3/85.3/78.2 | **90.1/85.4/80.0** |
| eraser | 94.2/**89.6/83.2** | **94.5**/-/- | 90.1/-/- | 92.0/-/- | 94.2/85.2/80.2 | 94.2/85.6/79.1 |
| fire_hood | 85.6/75.6/67.5 | 84.6/-/- | 83.6/-/- | 79.3/-/- | 89.4/80.8/**73.2** | **90.4/81.9**/72.7 |
| mint | 75.5/65.8/59.4 | 79.4/-/- | 76.9/-/- | 70.3/-/- | 83.1/70.7/65.9 | **86.1/74.3/68.5** |
| mounts | 92.6/**82.6/72.1** | **93.1**/-/- | 90.3/-/- | 89.9/-/- | 91.3/80.2/70.1 | 90.4/74.4/70.4 |
| pcb | 93.9/94.5/86.8 | 93.5/-/- | 91.3/-/- | 83.5/-/- | 94.3/94.7/86.5 | **95.5/95.8/88.0** |
| phone_battery | 92.5/87.6/78.7 | 92.5/-/- | 89.2/-/- | 87.6/-/- | **94.8/89.5/81.9** | 94.4/88.0/79.5 |
| plastic_nut | **95.0/88.8/82.3** | 93.5/-/- | 89.9/-/- | 80.7/-/- | 93.8/85.4/80.9 | 94.1/85.0/81.6 |
| plastic_plug | 92.6/**86.5/78.6** | 93.0/-/- | **93.3**/-/- | 81.5/-/- | 92.6/85.7/77.2 | 88.2/64.7/74.1 |
| porcelain_doll | 89.3/81.2/71.5 | 90.7/-/- | 90.4/-/- | 86.3/-/- | **92.2/82.7/74.6** | 91.2/80.1/69.6 |
| regulator | **87.1/76.4/69.1** | 86.1/-/- | 87.2/-/- | 58.9/-/- | 81.2/66.1/57.0 | 84.1/63.0/59.0 |
| rolled_strip_base | **99.2/99.2**/95.8 | 98.6/-/- | 97.0/-/- | 98.2/-/- | 97.8/97.6/93.1 | 98.9/98.9/**96.4** |
| sim_card_set | 97.1/96.1/89.4 | 97.1/-/- | 91.0/-/- | 91.8/-/- | 96.8/95.0/90.1 | **98.1/96.8/91.6** |
| switch | 95.7/95.3/89.5 | 89.5/-/- | 83.4/-/- | 86.8/-/- | 97.9/97.5/**92.7** | **98.2/97.6**/91.4 |
| tape | **98.4/96.8/91.6** | 98.3/-/- | 98.2/-/- | 97.9/-/- | 98.1/96.2/89.6 | **98.4**/96.7/89.7 |
| terminalblock | **97.3/97.2/91.7** | 95.5/-/- | 96.4/-/- | 89.2/-/- | 95.9/94.9/86.8 | 96.3/94.9/88.7 |
| toothbrush | **90.0/86.6/79.4** | 89.0/-/- | 81.5/-/- | 83.3/-/- | 86.8/82.7/75.5 | 88.9/84.9/78.7 |
| toy | 89.3/86.2/80.3 | 87.9/-/- | 84.7/-/- | 68.0/-/- | 88.2/85.2/79.9 | **90.6/86.8/82.2** |
| toy_brick | 80.7/**68.2/62.6** | **81.6**/-/- | 72.7/-/- | 76.8/-/- | 76.2/59.7/57.5 | 78.7/64.2/59.8 |
| transistor1 | 96.2/94.5/91.3 | 97.1/-/- | 97.3/-/- | 93.9/-/- | 96.7/95.7/91.2 | **97.6/97.0/92.2** |
| usb | 94.6/90.3/83.3 | 92.6/-/- | 94.5/-/- | 80.7/-/- | 94.4/89.6/82.0 | **95.5/90.6/84.3** |
| usb_adaptor | 86.3/77.1/71.1 | 79.4/-/- | 77.5/-/- | 79.0/-/- | 88.0/72.3/70.8 | **90.4/76.6/73.0** |
| u_block | 94.4/**88.7/79.7** | **94.8**/-/- | 93.6/-/- | 90.5/-/- | 93.8/85.4/78.9 | 91.9/80.1/72.2 |
| vcpill | 91.6/89.3/80.3 | 92.0/-/- | 76.1/-/- | 86.3/-/- | 93.6/91.2/83.5 | **94.8/92.7/84.3** |
| wooden_beads | 88.0/82.4/73.0 | 87.9/-/- | 82.8/-/- | 79.6/-/- | 89.2/81.0/74.3 | **90.5/83.9/77.0** |
| woodstick | 79.7/60.9/58.0 | 80.8/-/- | 88.7/-/- | 81.9/-/- | 91.0/75.2/72.7 | **91.6/78.1/73.7** |
| zipper | 97.4/97.6/92.1 | 97.6/-/- | **98.6**/-/- | 97.6/-/- | 98.2/**98.1**/93.8 | 97.5/96.3/**94.0** |
| **Mean** | 91.0/**85.9**/79.1 | 90.4/-/- | 88.1/-/- | 84.2/-/- | 91.4/84.7/78.4 | **92.4**/84.9/**79.4** |

*Table 25.* Per-class performance on Real-IAD dataset for single-class anomaly localization with AUROC/AP/F1-max/AUPRO metrics. Noise Ratio 10

| | SoftPatch | PatchCore | RD | UniAD | Dinomaly | Dinomaly + MeDS (ours) |
|---|---|---|---|---|---|---|
| audiojack | 98.5/40.0/44.6/90.6 | -/-/-/89.6 | -/-/-/91.3 | -/-/-/85.8 | 99.3/**48.6/55.1**/95.5 | **99.3**/46.9/54.1/**95.7** |
| bottle_cap | 99.6/24.6/29.4/97.3 | -/-/-/97.8 | -/-/-/98.2 | -/-/-/96.7 | **99.8**/37.1/42.0/**98.8** | **99.8**/32.9/37.1/98.7 |
| button_battery | 98.7/**40.8/38.1**/88.1 | -/-/-/90.1 | -/-/-/91.7 | -/-/-/75.5 | 98.8/17.2/36.2/92.0 | **99.0**/19.9/**38.1**/93.5 |
| end_cap | 97.1/16.4/25.9/91.0 | -/-/-/91.8 | -/-/-/93.3 | -/-/-/83.8 | 99.0/**29.8/34.0**/96.8 | **99.2**/27.8/32.3/**96.9** |
| eraser | 99.6/36.3/39.7/96.8 | -/-/-/98.4 | -/-/-/97.7 | -/-/-/94.7 | **99.8/41.8**/41.6/**99.0** | **99.8**/38.2/**46.2**/98.1 |
| fire_hood | 98.2/28.4/35.3/87.4 | -/-/-/90.0 | -/-/-/93.5 | -/-/-/85.9 | 99.6/44.0/47.5/**97.2** | **99.7/45.2/49.3**/96.6 |
| mint | 97.1/15.9/25.7/77.1 | -/-/-/83.5 | -/-/-/**87.6** | -/-/-/58.6 | **97.9**/15.3/**29.8**/86.7 | **97.9/16.5**/29.4/**86.7** |
| mounts | 98.2/33.2/38.6/83.8 | -/-/-/90.2 | -/-/-/93.6 | -/-/-/94.3 | **99.7**/38.9/45.1/**97.7** | 99.6/**40.8/46.9**/95.1 |
| pcb | 99.3/48.8/50.4/94.0 | -/-/-/94.3 | -/-/-/94.1 | -/-/-/83.9 | 99.7/60.2/60.2/97.2 | **99.8/60.5/60.5/97.8** |
| phone_battery | 99.4/44.8/46.6/96.3 | -/-/-/95.8 | -/-/-/98.0 | -/-/-/92.8 | **99.9**/53.0/54.8/**98.8** | **99.9**/51.5/**55.6**/98.2 |
| plastic_nut | 99.4/29.5/34.6/96.4 | -/-/-/96.1 | -/-/-/97.2 | -/-/-/90.0 | **99.8**/37.6/**43.5/98.5** | **99.8**/36.3/42.8/98.2 |
| plastic_plug | 99.0/**31.2**/34.0/94.4 | -/-/-/96.0 | -/-/-/96.7 | -/-/-/88.2 | **99.6**/30.8/**36.9/97.5** | 98.1/12.9/20.8/88.5 |
| porcelain_doll | 99.0/27.5/34.3/94.0 | -/-/-/96.7 | -/-/-/97.7 | -/-/-/93.5 | **99.7**/38.0/42.3/**98.5** | 99.6/36.6/**43.4**/97.7 |
| regulator | 99.3/25.2/34.8/95.8 | -/-/-/96.9 | -/-/-/**98.0** | -/-/-/73.1 | **99.7**/46.0/50.7/97.4 | **99.7**/33.4/44.9/**97.4** |
| rolled_strip_base | 99.7/29.9/37.7/98.8 | -/-/-/98.9 | -/-/-/98.9 | -/-/-/98.3 | **99.8**/48.2/51.1/**99.0** | **99.8**/33.0/44.6/98.7 |
| sim_card_set | 99.5/50.3/49.4/96.5 | -/-/-/97.1 | -/-/-/96.2 | -/-/-/87.5 | **99.8**/51.5/53.4/**98.3** | 99.6/**54.2/54.5**/96.5 |
| switch | **99.0**/61.4/58.6/94.7 | -/-/-/93.4 | -/-/-/93.4 | -/-/-/92.2 | 97.9/66.8/65.5/96.3 | 98.9/**72.8/68.4/97.6** |
| tape | 99.7/40.8/45.1/98.5 | -/-/-/98.1 | -/-/-/**99.4** | -/-/-/98.6 | 99.8/46.7/51.4/**99.3** | **99.9/51.0/53.7**/99.1 |
| terminalblock | 99.7/37.5/40.9/97.3 | -/-/-/98.2 | -/-/-/**99.0** | -/-/-/95.0 | **99.8**/45.2/48.1/**98.9** | 99.7/**47.0/48.7**/98.2 |
| toothbrush | **98.0/42.4/45.5**/91.2 | -/-/-/**93.3** | -/-/-/91.7 | -/-/-/88.3 | 96.4/28.6/36.9/88.9 | 96.9/33.0/40.2/89.1 |
| toy | **97.4**/15.7/21.3/89.8 | -/-/-/92.2 | -/-/-/**93.5** | -/-/-/77.8 | 95.9/22.6/30.5/92.2 | 96.8/**26.1/33.5/93.4** |
| toy_brick | 96.4/21.3/29.3/82.5 | -/-/-/85.2 | -/-/-/**88.3** | -/-/-/83.1 | 97.6/21.6/28.7/84.5 | **97.7/25.3/31.5/85.0** |
| transistor1 | 99.3/40.9/39.7/96.7 | -/-/-/97.9 | -/-/-/**98.3** | -/-/-/93.8 | 99.5/54.9/54.4/98.0 | **99.6/57.4/54.9/98.0** |
| usb | **99.6**/40.1/41.2/97.3 | -/-/-/96.4 | -/-/-/97.3 | -/-/-/88.1 | 99.3/46.9/46.5/98.0 | 99.5/**47.5/49.8/98.1** |
| usb_adaptor | 98.3/16.7/26.9/87.8 | -/-/-/92.8 | -/-/-/91.8 | -/-/-/83.7 | **99.5**/18.1/27.6/**96.8** | 99.4/**22.9/31.4**/94.1 |
| u_block | 99.5/30.2/37.6/96.7 | -/-/-/97.7 | -/-/-/**98.3** | -/-/-/93.6 | **99.8**/38.6/43.8/**98.5** | 99.7/**39.2/45.0**/97.9 |
| vcpill | 98.9/53.7/58.0/90.0 | -/-/-/94.6 | -/-/-/94.9 | -/-/-/93.9 | 99.5/64.2/66.2/**96.4** | **99.6/67.9/68.1/96.4** |
| wooden_beads | 97.4/32.6/37.4/85.2 | -/-/-/89.5 | -/-/-/91.8 | -/-/-/84.9 | **99.3**/44.8/48.3/**95.3** | **99.3/47.8/49.4**/94.7 |
| woodstick | 92.0/17.7/26.6/69.2 | -/-/-/70.8 | -/-/-/93.8 | -/-/-/78.3 | 99.2/34.1/43.3/95.0 | **99.3/43.6/48.9/95.7** |
| zipper | 98.6/36.9/39.3/96.3 | -/-/-/91.3 | -/-/-/98.2 | -/-/-/96.3 | **99.1**/46.3/**50.8/97.8** | 99.1/44.7/50.0/96.7 |
| **Mean** | 98.4/34.7/38.2/91.7 | -/-/-/93.2 | -/-/-/95.1 | -/-/-/87.7 | **99.2/40.6**/45.5/**96.2** | **99.2**/40.4/**45.8**/95.6 |

*Table 26.* Per-class performance on Real-IAD dataset for single-class anomaly detection with AUROC/AP/F1-max metrics. Noise Ratio 20

| | SoftPatch | PatchCore | RD | UniAD | Dinomaly | Dinomaly + MeDS (ours) |
|---|---|---|---|---|---|---|
| audiojack | 87.4/81.3/76.0 | 85.9/-/- | 85.3/-/- | 83.6/-/- | 86.0/74.4/68.7 | **91.5/83.0/74.1** |
| bottle_cap | **97.1/95.2/89.2** | 96.6/-/- | 93.5/-/- | 93.6/-/- | 91.3/82.8/73.9 | 95.5/91.7/81.2 |
| button_battery | 82.2/**77.8**/69.9 | 82.6/-/- | 80.5/-/- | 67.2/-/- | 79.0/70.7/66.8 | **85.2**/76.5/**72.9** |
| end_cap | 85.2/82.3/75.2 | 82.0/-/- | 76.2/-/- | 77.8/-/- | 87.7/82.8/77.2 | **89.7/85.4/79.6** |
| eraser | 94.0/**89.1/83.5** | **94.4**/-/- | 90.0/-/- | 91.0/-/- | 93.3/81.9/78.3 | 94.3/85.5/78.3 |
| fire_hood | 84.6/72.0/66.9 | 83.8/-/- | 82.6/-/- | 78.9/-/- | 88.5/79.0/71.4 | **90.4/81.6/72.3** |
| mint | 76.5/66.3/60.3 | 78.3/-/- | 72.7/-/- | 64.7/-/- | 81.3/67.8/64.3 | **85.2/73.4/67.7** |
| mounts | 92.9/**83.8/73.5** | **93.0**/-/- | 90.2/-/- | 90.4/-/- | 91.4/80.9/70.9 | 90.3/76.6/70.1 |
| pcb | 93.0/93.8/86.7 | 92.5/-/- | 89.0/-/- | 81.9/-/- | 93.4/93.8/85.4 | **95.5/95.7/88.1** |
| phone_battery | 92.2/86.2/77.4 | 91.0/-/- | 88.9/-/- | 85.8/-/- | 93.7/87.5/79.8 | **94.9/89.3/81.5** |
| plastic_nut | **94.4**/87.5/80.3 | 93.3/-/- | 87.0/-/- | 77.2/-/- | 92.3/82.3/77.0 | 93.8/84.5/**79.8** |
| plastic_plug | **92.6/85.4/77.7** | 92.6/-/- | 89.7/-/- | 78.9/-/- | 91.2/82.2/75.9 | 89.2/70.3/74.4 |
| porcelain_doll | 88.8/79.8/70.9 | 90.3/-/- | 89.6/-/- | 86.2/-/- | 91.3/80.5/**73.2** | **91.7/81.6**/71.7 |
| regulator | **87.2/76.8/68.8** | 85.1/-/- | 86.8/-/- | 58.3/-/- | 80.4/64.2/55.6 | 84.5/71.0/61.6 |
| rolled_strip_base | 97.9/97.9/92.1 | 97.6/-/- | 93.7/-/- | 97.9/-/- | 96.1/95.5/90.3 | **98.9/98.8/95.8** |
| sim_card_set | 96.5/95.4/88.3 | 96.8/-/- | 93.6/-/- | 90.0/-/- | 96.0/93.8/89.0 | **98.1/96.8/91.7** |
| switch | 95.6/94.9/89.7 | 89.0/-/- | 81.8/-/- | 85.5/-/- | 97.5/96.9/91.2 | **98.0/97.5/91.6** |
| tape | 97.9/95.7/**90.5** | 97.9/-/- | 97.4/-/- | 97.8/-/- | 97.4/94.8/88.3 | **98.4/96.6**/89.7 |
| terminalblock | **97.3/97.2/92.3** | 95.2/-/- | 96.4/-/- | 89.1/-/- | 94.9/93.7/84.9 | 96.5/95.1/89.6 |
| toothbrush | 88.4/84.3/77.4 | 87.5/-/- | 82.3/-/- | 81.7/-/- | 84.2/79.4/73.4 | **89.9/87.9/79.8** |
| toy | **88.6/85.3/79.6** | 86.0/-/- | 84.0/-/- | 66.9/-/- | 84.9/80.4/77.4 | 86.8/81.6/77.2 |
| toy_brick | **79.4/65.4/62.0** | 80.1/-/- | 78.5/-/- | 75.4/-/- | 72.8/54.0/54.8 | 76.1/60.5/56.9 |
| transistor1 | 94.2/92.2/87.2 | 96.5/-/- | 95.9/-/- | 92.7/-/- | 95.4/93.9/88.9 | **97.6/97.1/92.8** |
| usb | 94.0/89.1/83.4 | 92.1/-/- | 94.3/-/- | 80.9/-/- | 93.8/88.8/80.9 | **95.6/90.3/84.7** |
| usb_adaptor | 83.5/71.7/66.7 | 76.8/-/- | 70.7/-/- | 76.7/-/- | 84.3/66.2/66.5 | **89.6/76.2/72.4** |
| u_block | **94.4/87.8/79.6** | 94.7/-/- | 93.5/-/- | 89.9/-/- | 92.5/81.1/75.0 | 91.5/78.7/71.8 |
| vcpill | 90.6/87.8/78.9 | 90.9/-/- | 91.3/-/- | 86.7/-/- | 92.8/90.0/82.1 | **94.7/92.6/84.6** |
| wooden_beads | 87.7/81.6/73.7 | 87.4/-/- | 87.4/-/- | 79.0/-/- | 87.9/77.8/72.4 | **90.3/83.6/77.1** |
| woodstick | 78.6/57.5/56.9 | 79.2/-/- | 83.8/-/- | 80.7/-/- | 89.6/71.9/71.1 | **91.4/77.7/73.0** |
| zipper | 97.0/97.1/91.8 | 97.3/-/- | 92.3/-/- | 96.9/-/- | **97.9/97.6**/93.5 | 97.5/96.3/**94.0** |
| **Mean** | 90.3/84.6/78.2 | 89.6/-/- | 87.3/-/- | 82.8/-/- | 90.0/82.2/76.6 | **92.1/85.1/79.3** |

*Table 27.* Per-class performance on Real-IAD dataset for single-class anomaly localization with AUROC/AP/F1-max/AUPRO metrics. Noise Ratio 20

| | SoftPatch | PatchCore | RD | UniAD | Dinomaly | Dinomaly + MeDS (ours) |
|---|---|---|---|---|---|---|
| audiojack | 98.4/35.8/42.3/90.5 | -/-/-/88.8 | -/-/-/90.9 | -/-/-/85.1 | 99.2/46.7/53.3/95.3 | **99.4/47.0/53.6/96.4** |
| bottle_cap | 99.6/21.0/24.4/97.8 | -/-/-/98.0 | -/-/-/98.4 | -/-/-/96.7 | **99.8/36.0/41.2/98.7** | **99.8**/33.6/35.7/98.5 |
| button_battery | 98.7/**43.3/44.9**/89.1 | -/-/-/89.5 | -/-/-/90.8 | -/-/-/74.8 | 98.7/16.5/35.0/91.7 | **99.0**/20.2/38.1/**92.6** |
| end_cap | 95.2/14.8/24.7/89.3 | -/-/-/91.5 | -/-/-/93.6 | -/-/-/83.1 | 99.0/28.6/32.5/96.8 | **99.1/29.1/32.7/97.2** |
| eraser | 99.6/30.9/34.1/97.2 | -/-/-/98.4 | -/-/-/97.7 | -/-/-/94.7 | **99.8**/37.4/37.5/98.8 | **99.8/44.7/47.0/98.9** |
| fire_hood | 98.1/20.7/26.8/87.1 | -/-/-/89.8 | -/-/-/93.1 | -/-/-/85.9 | 99.6/41.0/45.0/97.1 | **99.8/44.8/48.7/97.4** |
| mint | 97.3/16.1/26.1/80.4 | -/-/-/82.9 | -/-/-/86.0 | -/-/-/57.2 | 97.9/15.4/**30.1**/85.0 | **98.0/16.3**/29.1/**87.4** |
| mounts | 98.2/29.9/34.4/86.4 | -/-/-/88.2 | -/-/-/93.6 | -/-/-/94.2 | 99.7/37.1/43.9/**97.7** | **99.8/42.1/47.5**/96.9 |
| pcb | 99.3/44.0/45.2/94.1 | -/-/-/94.3 | -/-/-/94.4 | -/-/-/83.8 | 99.6/57.2/57.3/97.0 | **99.8/60.0/60.7/97.9** |
| phone_battery | 99.5/43.9/46.0/96.2 | -/-/-/95.9 | -/-/-/98.0 | -/-/-/92.6 | 99.8/51.6/54.1/98.7 | **99.9/54.1/56.0/98.8** |
| plastic_nut | 99.4/29.0/33.8/96.6 | -/-/-/96.2 | -/-/-/96.8 | -/-/-/89.0 | **99.8**/36.6/**43.3**/98.2 | **99.8/39.5**/43.3/**98.4** |
| plastic_plug | 99.1/**28.9/33.5**/94.1 | -/-/-/96.5 | -/-/-/96.8 | -/-/-/87.4 | **99.6**/29.5/35.7/**97.4** | 98.5/17.2/27.2/91.4 |
| porcelain_doll | 99.0/24.8/32.2/94.3 | -/-/-/96.6 | -/-/-/97.7 | -/-/-/93.4 | **99.7**/36.6/41.0/**98.5** | 99.6/**36.6/42.7**/98.1 |
| regulator | 99.3/26.3/35.0/95.8 | -/-/-/97.2 | -/-/-/**98.1** | -/-/-/73.5 | 99.7/43.5/49.4/97.3 | **99.8/47.1/51.2**/98.0 |
| rolled_strip_base | 99.5/22.2/29.3/98.4 | -/-/-/98.8 | -/-/-/**99.0** | -/-/-/98.3 | **99.8/45.8/49.8/99.0** | **99.8**/33.0/44.3/98.6 |
| sim_card_set | 99.5/47.1/47.9/96.3 | -/-/-/97.2 | -/-/-/96.6 | -/-/-/87.3 | **99.8**/52.5/55.0/**98.4** | 99.7/**56.7**/55.4/96.8 |
| switch | 98.4/55.4/50.6/93.2 | -/-/-/93.3 | -/-/-/93.2 | -/-/-/91.8 | 97.9/68.0/65.9/96.2 | **98.9/73.1/68.3/97.5** |
| tape | 99.6/38.0/43.7/98.4 | -/-/-/98.0 | -/-/-/**99.3** | -/-/-/98.6 | **99.8**/44.8/50.5/**99.2** | **99.8/49.1/53.1**/99.1 |
| terminalblock | 99.7/37.7/40.5/97.7 | -/-/-/98.3 | -/-/-/**99.1** | -/-/-/95.4 | **99.8/46.1/48.8/98.9** | 99.7/**46.1/48.8**/98.6 |
| toothbrush | **97.8/38.1/42.7**/91.0 | -/-/-/**93.2** | -/-/-/91.1 | -/-/-/88.2 | 95.9/24.6/32.9/87.7 | 96.9/34.8/41.9/87.9 |
| toy | **97.1**/15.0/20.5/90.3 | -/-/-/91.5 | -/-/-/**93.1** | -/-/-/78.0 | 95.8/16.9/**23.9**/92.4 | 97.5/**18.3**/22.6/82.9 |
| toy_brick | 96.3/18.7/26.4/82.5 | -/-/-/85.0 | -/-/-/**88.5** | -/-/-/81.8 | 97.4/17.4/24.9/82.8 | **97.5/23.4/29.9/82.9** |
| transistor1 | 99.1/37.6/38.1/95.7 | -/-/-/97.8 | -/-/-/**98.0** | -/-/-/93.7 | 99.5/52.5/52.8/97.8 | **99.6/57.5/54.8/98.1** |
| usb | **99.5**/36.7/38.1/97.1 | -/-/-/96.5 | -/-/-/97.0 | -/-/-/87.7 | 99.2/45.4/45.5/98.0 | **99.5/47.1/49.6/98.5** |
| usb_adaptor | 98.1/14.3/23.7/87.4 | -/-/-/92.7 | -/-/-/90.3 | -/-/-/81.5 | 99.5/15.3/25.2/96.6 | **99.6/23.5/31.8/96.8** |
| u_block | 99.5/27.1/33.9/95.5 | -/-/-/97.8 | -/-/-/**98.3** | -/-/-/93.4 | **99.7/37.4/43.1/98.5** | 99.7/37.2/43.0/97.5 |
| vcpill | 98.8/53.2/58.5/89.6 | -/-/-/94.4 | -/-/-/96.0 | -/-/-/94.0 | 99.5/63.3/65.9/95.9 | **99.6/67.6/68.0/96.6** |
| wooden_beads | 97.3/33.1/37.9/84.4 | -/-/-/89.3 | -/-/-/91.6 | -/-/-/84.1 | 99.2/41.0/47.1/**94.9** | **99.3/47.6/49.2**/94.7 |
| woodstick | 91.8/14.9/23.3/68.0 | -/-/-/69.7 | -/-/-/92.5 | -/-/-/77.3 | 99.1/31.3/40.9/94.2 | **99.3/42.2/48.5/95.1** |
| zipper | 98.5/35.0/37.7/96.1 | -/-/-/91.4 | -/-/-/**98.1** | -/-/-/96.2 | 99.0/44.1/48.7/**97.8** | **99.2/45.3/50.1**/96.8 |
| **Mean** | 98.4/31.1/36.3/91.7 | -/-/-/93.0 | -/-/-/95.0 | -/-/-/87.2 | 99.1/38.7/44.0/**95.9** | **99.2/41.2/45.8**/95.6 |

*Table 28.* Per-class performance on Real-IAD dataset for single-class anomaly detection with AUROC/AP/F1-max metrics. Noise Ratio 40

| | SoftPatch | PatchCore | RD | UniAD | Dinomaly | Dinomaly + MeDS (ours) |
|---|---|---|---|---|---|---|
| audiojack | 85.0/77.5/71.4 | 82.8/-/- | 77.3/-/- | 80.4/-/- | 80.6/66.2/63.6 | **89.2/80.5/72.8** |
| bottle_cap | **95.6/92.6/85.5** | 95.5/-/- | 92.2/-/- | 92.6/-/- | 89.4/79.5/71.2 | 95.3/91.5/82.5 |
| button_battery | 79.8/**75.5**/67.7 | 79.9/-/- | **80.0**/-/- | 66.5/-/- | 75.9/67.4/65.0 | 81.2/73.2/**68.9** |
| end_cap | 81.3/77.7/72.2 | 79.8/-/- | 77.0/-/- | 76.2/-/- | 85.3/79.0/75.6 | **88.1/83.0/78.3** |
| eraser | 92.7/**86.8/78.7** | **92.8**/-/- | 89.7/-/- | 89.6/-/- | 90.8/76.2/72.4 | **93.5**/84.8/78.0 |
| fire_hood | 83.4/68.2/65.3 | 83.0/-/- | 78.0/-/- | 77.9/-/- | 87.7/77.4/69.9 | **90.0/80.8/71.5** |
| mint | 76.0/65.6/59.9 | 77.5/-/- | 75.2/-/- | 66.8/-/- | 78.7/62.7/62.3 | **82.5/70.7/65.9** |
| mounts | 91.7/**81.3/71.5** | **92.6**/-/- | 75.8/-/- | 90.5/-/- | 90.2/78.6/69.0 | 90.1/75.1/69.8 |
| pcb | 92.5/93.4/85.7 | 91.2/-/- | 89.1/-/- | 75.7/-/- | 92.0/92.4/83.5 | **94.8/95.3/87.2** |
| phone_battery | 91.4/85.5/75.2 | 88.4/-/- | 86.8/-/- | 82.0/-/- | 92.0/84.7/76.8 | **94.6/88.5/81.4** |
| plastic_nut | **93.5/85.5/78.9** | 92.6/-/- | 88.6/-/- | 74.5/-/- | 90.0/77.7/73.2 | 92.5/82.4/77.6 |
| plastic_plug | 92.0/**84.5/77.0** | **92.2**/-/- | 89.7/-/- | 75.2/-/- | 89.2/78.3/71.2 | 90.7/76.9/74.8 |
| porcelain_doll | 87.3/77.1/68.6 | 89.0/-/- | 87.8/-/- | 83.0/-/- | 89.9/77.3/70.2 | **91.8/81.9/71.9** |
| regulator | **86.1/76.1/68.6** | 84.3/-/- | 83.8/-/- | 57.1/-/- | 77.7/58.8/50.3 | 83.9/70.8/61.7 |
| rolled_strip_base | 95.6/95.5/89.0 | **95.8**/-/- | 91.0/-/- | 96.5/-/- | 93.2/92.1/86.4 | **98.3/98.2/93.7** |
| sim_card_set | 95.8/94.8/85.8 | **96.1**/-/- | 83.9/-/- | 84.7/-/- | 94.7/91.9/87.1 | 96.9/**95.5/88.9** |
| switch | 94.2/92.7/85.9 | 87.1/-/- | 71.8/-/- | 82.6/-/- | 96.4/95.4/88.5 | **98.0/97.6/91.5** |
| tape | 97.4/94.3/88.4 | 97.7/-/- | **97.8**/-/- | 97.0/-/- | 96.7/93.4/87.0 | **98.2/96.5/90.0** |
| terminalblock | 96.0/95.3/88.9 | 94.1/-/- | 92.6/-/- | 87.2/-/- | 93.1/91.3/79.7 | **97.6/97.0/91.7** |
| toothbrush | 86.2/80.7/75.2 | 85.6/-/- | 81.3/-/- | 76.9/-/- | 81.6/75.4/71.4 | **87.4/84.6/75.7** |
| toy | **87.7/83.7/79.1** | 83.5/-/- | 76.2/-/- | 58.8/-/- | 80.8/75.6/72.7 | 82.7/77.2/75.0 |
| toy_brick | 77.0/**61.1/59.9** | 78.1/-/- | **78.9**/-/- | 73.1/-/- | 69.1/49.4/52.0 | 72.4/55.3/55.0 |
| transistor1 | 92.2/89.6/84.1 | **94.5**/-/- | 92.9/-/- | 87.9/-/- | 92.5/89.9/84.3 | 96.2/**95.5/89.7** |
| usb | 92.4/86.3/80.1 | 89.9/-/- | 89.4/-/- | 76.9/-/- | 91.7/86.3/79.2 | **95.6/91.3/84.6** |
| usb_adaptor | 80.9/67.7/63.3 | 74.4/-/- | 73.1/-/- | 71.0/-/- | 80.4/60.0/61.0 | **84.0/68.0/64.8** |
| u_block | 93.5/**85.5/78.3** | **94.5**/-/- | 93.0/-/- | 88.1/-/- | 91.2/77.9/73.3 | 91.7/79.7/71.8 |
| vcpill | 89.7/86.0/77.8 | 89.4/-/- | 84.5/-/- | 83.5/-/- | 91.1/87.5/79.7 | **94.3/92.2/83.9** |
| wooden_beads | 86.6/79.7/72.1 | 85.9/-/- | 82.2/-/- | 75.4/-/- | 86.0/73.7/69.9 | **89.8/83.1/76.3** |
| woodstick | 77.1/54.9/54.5 | 78.1/-/- | 79.9/-/- | 79.4/-/- | 88.2/67.6/68.1 | **90.8/76.4/71.7** |
| zipper | 95.9/95.5/89.9 | 96.6/-/- | **96.7**/-/- | 94.9/-/- | 97.3/**96.4**/92.2 | **97.6**/96.2/**94.4** |
| **Mean** | 88.9/82.4/76.0 | 88.1/-/- | 84.5/-/- | 80.0/-/- | 87.8/78.7/73.6 | **91.0/84.0/78.0** |

*Table 29.* Per-class performance on Real-IAD dataset for single-class anomaly localization with AUROC/AP/F1-max/AUPRO metrics. Noise Ratio 40

| | SoftPatch | PatchCore | RD | UniAD | Dinomaly | Dinomaly + MeDS (ours) |
|---|---|---|---|---|---|---|
| audiojack | 98.2/32.0/39.2/89.1 | -/-/-/88.3 | -/-/-/90.4 | -/-/-/84.5 | 99.2/41.0/46.7/94.9 | **99.5/47.7/54.1/96.3** |
| bottle_cap | 99.6/19.7/24.4/98.0 | -/-/-/98.0 | -/-/-/98.3 | -/-/-/96.8 | **99.8/33.9/39.1/98.6** | 99.8/32.2/38.2/**98.6** |
| button_battery | 98.6/**37.6**/35.8/88.6 | -/-/-/88.3 | -/-/-/90.4 | -/-/-/73.9 | 98.6/16.1/34.2/91.8 | **99.0**/18.5/**38.0**/**92.6** |
| end_cap | 94.2/13.8/23.3/87.9 | -/-/-/91.0 | -/-/-/92.8 | -/-/-/83.3 | 98.9/25.7/30.1/96.4 | **99.1/27.5/30.9/96.7** |
| eraser | 99.5/28.8/32.2/96.4 | -/-/-/98.3 | -/-/-/97.8 | -/-/-/94.3 | 99.7/32.6/33.3/**98.6** | **99.8/44.3/46.9**/98.4 |
| fire_hood | 97.9/17.7/24.2/87.2 | -/-/-/89.6 | -/-/-/93.1 | -/-/-/85.8 | 99.6/39.2/42.8/**96.7** | **99.7/44.7/48.0**/96.5 |
| mint | 97.2/15.8/26.7/78.9 | -/-/-/82.3 | -/-/-/**86.9** | -/-/-/56.3 | 97.7/12.7/25.9/84.0 | **98.0/16.3/29.5/86.9** |
| mounts | 98.2/25.9/30.5/86.3 | -/-/-/85.8 | -/-/-/94.3 | -/-/-/94.2 | **99.7**/34.2/41.2/**97.7** | 99.6/**42.5/47.3**/96.7 |
| pcb | 99.3/42.9/42.9/93.5 | -/-/-/93.9 | -/-/-/94.1 | -/-/-/82.3 | 99.6/56.9/57.5/97.1 | **99.8/61.0/61.0/97.8** |
| phone_battery | 99.3/43.7/46.3/95.8 | -/-/-/95.1 | -/-/-/97.7 | -/-/-/91.6 | 99.8/50.7/53.6/98.6 | **99.9/54.7/55.3/98.8** |
| plastic_nut | 99.4/27.9/33.3/96.2 | -/-/-/96.0 | -/-/-/96.6 | -/-/-/88.2 | 99.7/33.7/42.4/**98.1** | **99.8/37.6/43.2**/97.8 |
| plastic_plug | 99.1/**29.1/33.9**/93.7 | -/-/-/96.6 | -/-/-/96.7 | -/-/-/87.3 | **99.6**/27.5/33.4/**97.3** | 99.2/26.2/32.6/93.8 |
| porcelain_doll | 99.0/24.4/31.4/94.1 | -/-/-/96.4 | -/-/-/97.8 | -/-/-/93.1 | 99.6/34.7/39.4/98.4 | **99.7/37.4/42.7**/98.2 |
| regulator | 99.2/26.0/35.0/96.0 | -/-/-/96.7 | -/-/-/97.8 | -/-/-/72.7 | 99.6/40.1/46.6/97.4 | **99.8/47.6/51.2/98.1** |
| rolled_strip_base | 99.4/19.3/26.5/98.1 | -/-/-/98.7 | -/-/-/98.6 | -/-/-/98.2 | **99.8/41.6/46.5/98.8** | 99.8/36.0/45.1/98.7 |
| sim_card_set | 99.6/49.6/52.2/96.4 | -/-/-/97.4 | -/-/-/96.7 | -/-/-/86.4 | **99.8**/50.6/53.8/**98.1** | 99.8/**58.8/57.5**/97.7 |
| switch | 91.2/19.0/27.6/90.7 | -/-/-/91.3 | -/-/-/90.6 | -/-/-/90.8 | 97.0/53.6/57.1/95.5 | **98.9/72.8/68.4/97.5** |
| tape | 99.6/35.6/40.9/98.2 | -/-/-/98.0 | -/-/-/**99.3** | -/-/-/98.4 | 99.8/43.1/48.8/**99.2** | **99.9/50.3/52.5**/99.0 |
| terminalblock | 99.7/32.2/35.4/97.6 | -/-/-/98.4 | -/-/-/**99.1** | -/-/-/95.2 | **99.8**/44.7/48.4/98.9 | 99.8/**48.5/49.9**/98.9 |
| toothbrush | **97.6/35.9/41.2**/90.7 | -/-/-/92.8 | -/-/-/91.0 | -/-/-/87.0 | 95.5/21.5/29.9/86.5 | 96.5/31.4/39.0/**89.2** |
| toy | **96.7**/13.8/18.2/89.3 | -/-/-/91.2 | -/-/-/**92.3** | -/-/-/74.8 | 95.4/12.1/19.6/91.0 | 96.1/**15.8/20.6**/90.5 |
| toy_brick | 96.1/16.5/23.4/81.2 | -/-/-/84.3 | -/-/-/**88.5** | -/-/-/80.9 | 96.9/14.1/21.2/79.5 | **97.0/19.8/26.3**/78.7 |
| transistor1 | 98.8/35.1/36.2/94.7 | -/-/-/97.4 | -/-/-/**97.8** | -/-/-/92.7 | 99.4/47.7/49.2/97.2 | **99.6/57.0/53.8/97.9** |
| usb | 99.4/33.5/35.3/96.3 | -/-/-/95.3 | -/-/-/96.4 | -/-/-/87.2 | 99.2/43.0/43.9/97.7 | **99.4/47.1/49.6/98.2** |
| usb_adaptor | 98.0/11.7/20.1/86.8 | -/-/-/92.0 | -/-/-/91.1 | -/-/-/79.6 | 99.4/12.2/22.3/96.1 | **99.5/21.9/30.2/96.4** |
| u_block | 99.5/25.9/33.4/95.7 | -/-/-/97.8 | -/-/-/**98.2** | -/-/-/93.0 | **99.7**/34.9/41.0/**98.4** | 99.7/**36.5/43.7**/97.5 |
| vcpill | 98.8/50.5/56.1/88.9 | -/-/-/93.9 | -/-/-/94.2 | -/-/-/93.2 | 99.4/61.8/64.3/95.6 | **99.5/67.7/67.9/95.9** |
| wooden_beads | 97.2/33.1/38.3/84.2 | -/-/-/88.8 | -/-/-/91.2 | -/-/-/84.0 | 99.2/37.4/45.5/**94.5** | **99.3/48.0/50.3**/94.2 |
| woodstick | 91.3/10.9/21.2/66.0 | -/-/-/68.3 | -/-/-/92.6 | -/-/-/77.6 | 99.0/26.1/37.5/94.3 | **99.2/41.8/47.7/94.9** |
| zipper | 98.3/31.2/35.6/95.6 | -/-/-/91.2 | -/-/-/**98.0** | -/-/-/95.5 | 98.7/38.7/43.0/97.4 | **99.2/45.1/51.9**/97.1 |
| **Mean** | 98.0/28.0/33.4/91.1 | -/-/-/92.5 | -/-/-/94.7 | -/-/-/86.6 | 99.0/35.4/41.3/95.5 | **99.2/41.2/45.8/95.7** |

*Table 30.* Active label correction on MVTecAD dataset with Dinomaly baseline

| metric | AUPRC (↑) | | | | | | Inspection depth (↓) | | | | | |
|---|---|---|---|---|---|---|---|---|---|---|---|---|
| noise ratio | 10 | | 20 | | 40 | | 10 | | 20 | | 40 | |
| class | Dinomaly | + MeDS | Dinomaly | + MeDS | Dinomaly | + MeDS | Dinomaly | + MeDS | Dinomaly | + MeDS | Dinomaly | + MeDS |
| carpet | 97.78 | 99.41 | 98.68 | 99.88 | 98.97 | 99.87 | 10.71 | 10.06 | 17.86 | 17.56 | 25.47 | 24.93 |
| grid | 99.05 | 100 | 99.2 | 99.41 | 99.24 | 99.53 | 9.31 | 8.97 | 20.19 | 24.29 | 22.74 | 24.3 |
| leather | 94.59 | 100 | 94.84 | 99.83 | 96.74 | 99.77 | 9.67 | 8.92 | 18.03 | 17.01 | 29.38 | 27.6 |
| tile | 95.97 | 98.49 | 96.66 | 99.24 | 98.31 | 99.74 | 11.07 | 10.67 | 21.38 | 18.12 | 33.12 | 28.03 |
| wood | 97.72 | 100 | 98.29 | 99.92 | 98.22 | 99.95 | 12.87 | 9.19 | 20.95 | 17.23 | 23.45 | 20.2 |
| bottle | 96.83 | 100 | 98.55 | 100 | 99.09 | 100 | 23.91 | 9.13 | 25.9 | 16.73 | 33.46 | 23.16 |
| cable | 75.56 | 92.83 | 75.12 | 92.72 | 75.54 | 94.55 | 71.95 | 9.76 | 98.14 | 53.53 | 99.68 | 66.88 |
| capsule | 85.36 | 90.8 | 88.35 | 96.03 | 92.01 | 97.93 | 32.78 | 24.9 | 49.81 | 29.28 | 61.56 | 44.63 |
| hazelnut | 83.68 | 99.87 | 85.08 | 99.72 | 83.61 | 99.81 | 74.03 | 11.16 | 80.94 | 12.56 | 82 | 15.4 |
| metal_nut | 97.19 | 100 | 94.78 | 100 | 89.32 | 100 | 11.16 | 9.09 | 24.24 | 16.67 | 87.01 | 28.57 |
| pill | 88.54 | 95.73 | 91.79 | 97.89 | 95.03 | 99.51 | 55.1 | 13.95 | 66.88 | 38.44 | 67.11 | 43.05 |
| screw | 81.67 | 91.72 | 86.78 | 92.92 | 87.94 | 95.18 | 37.22 | 33.81 | 71.35 | 47.92 | 79.73 | 76.77 |
| toothbrush | 94.44 | 95.83 | 99.36 | 100 | 99.51 | 100 | 13.64 | 12.12 | 18.06 | 16.67 | 30.95 | 28.57 |
| transistor | 84.7 | 99.01 | 81.42 | 98 | 80.26 | 98.53 | 35.47 | 10.68 | 71.54 | 18.97 | 66.8 | 19.76 |
| zipper | 100 | 100 | 100 | 100 | 99.91 | 99.97 | 9.09 | 9.09 | 16.67 | 16.67 | 29.17 | 29.46 |
| Mean | 91.54 | 97.58 | 92.59 | 98.37 | 92.91 | 98.96 | 27.86 | 12.77 | 41.46 | 24.11 | 51.44 | 33.42 |
| Std | 7.61 | 3.34 | 7.63 | 2.52 | 7.89 | 1.77 | 22.85 | 7.08 | 28.49 | 12.56 | 27.06 | 17.5 |

