# OpenReview forum: "Memory-Distilled Selection for Noise-Robust Anomaly Detection"
_ICML.cc/2026/Conference — ICML 2026 regular_

### Official Review · Reviewer_j6Z8 · 2026-03-04

**Soundness:** 3
**Presentation:** 3
**Significance:** 3
**Originality:** 3
**Overall Recommendation:** 4
**Confidence:** 3

**Summary:**

This paper proposes Memory-Distilled Selection (MeDS) for anomaly detection under contaminated training data. The method builds a bootstrapped memory ensemble to obtain reliable anomaly scores, distills these scores into a reconstruction network, and progressively filters noisy samples during training. Experiments on datasets such as MVTecAD and VisA show that the approach improves robustness to noisy data and achieves competitive anomaly detection performance.

The overall idea of this work is similar to that of *Vague Prototype-Oriented Diffusion Model for Multi-class Anomaly Detection*. It is recommended to include comparative experiments with this method.

Figure 1 could be further improved so that readers can more intuitively understand the proposed approach.

Since the proposed method consists of multiple sequential steps, it would be helpful to strengthen the ablation study to clearly demonstrate the importance of each component.

**Compliance With Llm Reviewing Policy:**

Affirmed.

**Key Questions For Authors:**

See summary

**Limitations:**

See summary

**Strengths And Weaknesses:**

See summary

---

> ### Author Rebuttal · Authors · 2026-03-31
>
> We sincerely thank the reviewer for the positive assessment and constructive suggestions. We are glad that the reviewer recognizes the competitive performance and robustness of MeDS.
>
> >**C1: Similarity to "Vague Prototype-Oriented Diffusion Model" -- comparison needed**
>
> We thank the reviewer for pointing out this work. However, VPDM[1] and MeDS differ fundamentally:
>
> - **Different problems:** VPDM tackles multi-class AD under ***clean*** data; MeDS tackles AD under ***noisy/contaminated*** training data.
> - **Different paradigms:** VPDM is a specific generative model (diffusion-based); MeDS is an architecture-agnostic *training framework* applicable to any teacher-student model (demonstrated with HVQ [2], Dinomaly [3], and INP-Former [4] in Tables 1–3). Hence, MeDS cannot be directly applicable to VPDM, which is a generative model.
> - **No implementation:** VPDM's code is **not publicly available**, and hence unfortunately, we could not compare during the rebuttal period.
>
> >**C2: Figure 1 clarity**
>
> We thank the reviewer for this suggestion. Our design intent for Figure 1 was to provide a clean, **high-level overview of the three-stage pipeline**: (1) bootstrapped memory ensemble construction, (2) score distillation, and (3) progressive data selection with fine-tuning. We aimed for simplicity to convey the overall pipeline flow rather than detailed architectural components.
>
> We would greatly appreciate more specific guidance on which aspects the reviewer finds unclear or would like to see improved.
>
> >**C3: Strengthen ablation study**
>
> We appreciate this suggestion. Table 4 ablates all 6 combinations of {memory ensemble, distillation, selection} on MVTecAD yielding consistent conclusions. We highlight key findings:
>
> - Memory ensemble improves I-AUROC by +1.77pp at 40% noise ratio, and without it we cannot distill; however, P-AP remains low.
> - Adding distillation improves I-AUROC by +6.54pp, demonstrating its effectiveness.
> - If we skip distillation (Row 4) and perform selection alone, performance at 0% noise falls short of full MeDS (98.93% vs. 99.37% I-AUROC; 67.84% vs. 68.35% P-AP).
> - With distillation alone, on the other hand, pixel-level performance is poor (e.g., 56.88% vs. 68.05% P-AP at 40% noise).
>
> This clearly shows the effectiveness of each component of MeDS.
>
> Moreover, Table 5 (Appendix) replicates this with INP-Former, confirming architecture-agnostic consistency:
>
> - memory ensemble improves I-AUROC by +6.37pp at 40% noise,
> - distillation further improves I-AUROC by +5.54pp and P-AP by +6.88pp at 40% noise,
> - skipping distillation (Row 4) falls short of full MeDS at 0% noise (97.25% vs. 99.45% on I-AUROC, 64.60% vs. 67.15% on P-AP), showing distillation is critical for selection fine-tuning
> - selection fine-tuning on the other hand improves +2.2pp for I-AUROC and +2.55pp for P-AP at 0% noise ratio
>
> **References:**
>
> [1] Zhang et al. "Vague Prototype-Oriented Diffusion Model for Multi-class Anomaly Detection." ICCV, 2023.
>
> [2] Lu et al. "Hierarchical Vector Quantized Transformer for Multi-class Unsupervised Anomaly Detection." NeurIPS, 2023.
>
> [3] Guo et al. "Dinomaly: The Less Is More Philosophy in Multi-Class Unsupervised Anomaly Detection." CVPR, 2025.
>
> [4] Luo et al. "Exploring Intrinsic Normal Prototypes within a Single Image for Universal Anomaly Detection." CVPR, 2025.

---

> > ### Author Rebuttal · Reviewer_j6Z8 · 2026-04-03
> >
> > Thanks for your reply

---

> > > ### Author Response · Authors · 2026-04-08
> > >
> > > We sincerely thank the reviewer for the constructive suggestions and for the continued engagement!
> > >
> > > > **R1: Similarity to VPDM**
> > >
> > > We acknowledge two connections between VPDM [1] and MeDS: (i) both target the multi-class anomaly detection setting, and (ii) both follow a **gradual / coarse-to-fine learning** principle. We will cite VPDM in the revised manuscript accordingly.
> > >
> > > > **R2: Figure 1 clarity**
> > >
> > > **How we will revise Figure 1:**
> > >
> > > - We will display **explicit shape information** for every intermediate tensor in a clear manner in all three panels (e.g., $g(x) \in \mathbb{R}^{H\times W\times C}$, $f_\theta(x) \in \mathbb{R}^{H\times W\times C}$, $s_\mathbb{M} \in \mathbb{R}^{H\times W}$, $s_\theta \in \mathbb{R}^{H\times W}$)
> > > - We will **enhance annotations for all notations**, differentiating function (network), feature map, and feature vector.
> > >
> > > **How we will revise its caption:**
> > >
> > > We will replace the brief caption with a full self-contained caption (shown below):
> > >
> > > **Figure 1.**
> > > Overview of the three-stage MeDS pipeline. **First stage: bootstrapped memory ensemble.** From each training image $x \in \mathcal{D}$, a frozen teacher network $g$ extracts patch features $g(x)\_{hw} \in \mathbb{R}^{C}$ for all spatial positions $(h, w)$. The pooled feature collection $g(\mathcal{D}) \subseteq \mathbb{R}^{C}$ of size $\lvert g(\mathcal{D}) \rvert = \lvert \mathcal{D} \rvert \cdot H \cdot W$ is randomly subsampled $B$ times at ratio $\rho$ to form partial memory banks $\lbrace \mathcal{M}\_b \rbrace\_{b=1}^{B}$. Each memory bank $\mathcal{M}\_b$ produces a per-patch nearest-neighbor distance score $s\_{\mathcal{M}\_b}(x) \in \mathbb{R}^{H \times W}$, and averaging these scores across all $B$ memory banks yields the ensemble score $s\_{\mathbb{M}}(x) \in \mathbb{R}^{H \times W}$. The sparse subsampling acts as a low-pass filter that downweights anomalous patches. **Second stage: score-space distillation.** For further enhancement, the memory score $s\_{\mathbb{M}}$ is distilled into a network-based reconstruction score $s\_\theta$ via $\min\_\theta \lvert s\_{\mathbb{M}}(x) - s\_\theta(x) \rvert^2$, where $s\_\theta(x) \in \mathbb{R}^{H \times W}$ is the distance between teacher features $g(x)$ and the features $f\_\theta(x) \in \mathbb{R}^{H \times W \times C}$ extracted from a student network $f\_\theta$. **Third stage: progressive self-selection.** Finally, the reconstruction score $s\_\theta$ is fine-tuned on a clean subset $\mathcal{S}\_t \subseteq \mathcal{D}$ obtained by a progressive selection criterion.
> > >
> > > > **R3: Strengthen ablation study**
> > >
> > > We thank the reviewer for this valuable suggestion. To further strengthen the ablation study in the main paper, we conducted **four *additional* ablations** on Dinomaly / MVTec-AD. These results are provided in (**I-AUROC / P-AP**, %; values in parentheses indicate differences relative to the Baseline row):
> > >
> > >
> > > |Ablation Point|0%|10%|20%|40%|
> > > |:-|:-:|:-:|:-:|:-:|
> > > |Baseline: (original Dinomaly).|99.64 / 68.19|95.19 / 58.22|92.16 / 54.60|87.38 / 53.00|
> > > |w/o selection: after distillation, fine-tunes the model on unfiltered data.|99.63 (-0.01) / 67.64 (-0.55)|97.67 (**+2.48**) / 61.45 (**+3.23**)|95.78 (**+3.62**) / 57.86 (**+3.26**)|92.65 (**+5.27**) / 56.28 (**+3.28**)|
> > > |Zero-score distillation: replace the memory target with zeros ($s_\mathbb{M} = \mathbf{0}$).|99.25 (-0.39) / 68.51 (**+0.32**)|98.09 (**+2.90**) / 65.55 (**+7.33**)|97.16 (**+5.00**) / 65.38 (**+10.78**)|95.98 (**+8.60**) / 64.06 (**+11.06**)|
> > > |$\alpha_t\!=\!0$: use only the frozen anchor $s_{\theta_0}$ in Eq. 11 (no iterative refinement).|98.67 (-0.97) / 67.56 (-0.63)|98.75 (**+3.56**) / 68.81 (**+10.59**)|98.85 (**+6.69**) / 68.37 (**+13.77**)|98.90 (**+11.52**) / 68.31 (**+15.31**)|
> > > |$\alpha_t\!=\!1$: use only the current student score $s_{\theta_t}$ in Eq. 11 (no anchor).|99.46 (-0.18) / 67.54 (-0.65)|99.45 (**+4.26**) / 67.65 (**+9.43**)|99.40 (**+7.24**) / 67.00 (**+12.40**)|99.09 (**+11.71**) / 66.39 (**+13.39**)|
> > > |**Full MeDS**|**99.36 / 68.11**|**99.49 / 68.96**|**99.31 / 68.63**|**99.16 / 67.51**|
> > >
> > > The four newly added ablations (w/o selection, Zero-score distillation, $\alpha_t = 0$, and $\alpha_t = 1$) isolate the contribution of each MeDS component, and we will include them in the revised manuscript.
> > >
> > > ---
> > >
> > > [1] Zhang et al. "Vague Prototype-Oriented Diffusion Model for Multi-class Anomaly Detection." ICCV, 2023.

---

### Official Review · Reviewer_Diz7 · 2026-03-05

**Soundness:** 2
**Presentation:** 3
**Significance:** 3
**Originality:** 2
**Overall Recommendation:** 4
**Confidence:** 3

**Summary:**

This paper proposes a novel algorithm for industrial anomaly detection under the setting of anomaly contamination in the training set. Based on a pretrained teacher model, a student model is trained to reconstruct the teacher’s memory-based anomaly scores. To address the trade-off between image-level robustness and precise pixel-level localization, the student network is further fine-tuned using selected clean samples.

**Compliance With Llm Reviewing Policy:**

Affirmed.

**Final Justification:**

I thank the authors for their responses. Most of my concerns are addressed. Although it seems that the method is time-costly compared to the baselines, given IAD with anomaly contamination is an important problem and the proposed method is effective in this scenario, I update my score.

**Key Questions For Authors:**

Please refer to the concerns raised in the Weaknesses section above.

**Limitations:**

The potential computational overhead of memory-based methods (as mentioned in Weakness 4) may limit scalability in real-world industrial deployment scenarios. A more detailed efficiency analysis would help clarify this issue.

**Strengths And Weaknesses:**

**Strengths**
1. The proposed method demonstrates strong performance on datasets with anomaly contamination, while maintaining competitive results on datasets with clean training data, indicating good robustness and generalization ability.
2. The paper is generally well-written and easy to follow. Most sections are clearly structured, although certain parts in the methodology and experimental sections could benefit from further clarification.
3. The authors provide comprehensive ablation studies to analyze the contribution of each component and the impact of different hyperparameter settings, which strengthens the empirical validation.
4. Theoritical gurantee is provided for $s_{M}$. Could the authors provide more theoritical results for $s_\theta?

**Weaknesses**
1. Some parts of methodlogy is confusing, for instance, in Eq. (3), $s_\theta$ maps iamge into a vector shape of $\mathbb{R} ^{H \times W}$, then what does Eq. (4) and Eq. (10) mean? minimizing a vector is confusing.
2. The paper lacks a brief introduction to the baseline methods used for comparison. It would be helpful to clarify how these baselines and the pretrained encoders are trained. Are they trained under the same data setting (including anomaly contamination) as the proposed method? Ensuring fairness in experimental comparison should be explicitly discussed.
3. More experimental analysis regarding the sample selection rule would be valuable. For instance, are the selected samples guaranteed to be normal data, or do they still contain some anomalies? Quantitative statistics about the purity of the selected set would strengthen the claims.
4. Memory-based methods typically compute distances between patch embeddings and all stored training embeddings, which can be computationally expensive. It would be useful to provide training and inference time comparisons across different methods to better assess practical efficiency.
5. Including additional baselines specifically designed for anomaly detection under contamination settings—especially recent methods published in top venues such as ICCV or CVPR—would further strengthen the experimental evaluation and make the results more convincing.

---

> ### Author Rebuttal · Authors · 2026-03-31
>
> We sincerely thank the reviewer for the constructive feedback. We address each concern below.
>
> >**C1: Eq. (3) maps to a vector, so minimizing in Eq. (4) and (10) is confusing**
>
> We apologize for the confusion. The reconstruction score $s\_\theta(x)\_{hw}$ in Eq. (3) is a **scalar** at each position $(h,w)$: it is $D(g(x)\_{hw}, f\_\theta(x)\_{hw})$, the cosine distance between the frozen teacher feature $g(x)\_{hw} \in \mathbb{R}^C$ and the student reconstruction $f\_\theta(x)\_{hw} \in \mathbb{R}^C$, producing a single real value. In Eq. (4) and (10), $s\_\theta(x)$ denotes the **mean** over all spatial positions $(h,w)$: $s\_\theta(x) = \frac{1}{HW}\sum\_{h,w} s\_\theta(x)\_{hw}$, a scalar. The minimization thus operates on a **scalar-valued loss** averaged over samples, not a vector. We will make this aggregation explicit in the revision.
>
> >**C2: Missing baseline introductions and training fairness**
>
> We apologize for the insufficient description of baselines and training setup. All baselines are trained under **identical contaminated data conditions** as described in Sec. 5.1. All methods train on the **same noisy training sets** with the same noise ratios. Each baseline uses its own architecture and encoder. When applying MeDS, we use that baseline's own pretrained encoder; **the only difference is the MeDS training procedure**. We will state this explicitly and add brief baseline descriptions in the revision.
>
> >**C3: Error accumulation in progressive selection**
>
> We thank the reviewer for raising this important point. We tracked the purity (fraction of true normals) of the selected subset $\mathcal{S}\_t$ across all training iterations:
>
> | Noise Ratio | Total Defects | Defects Selected (iter 0) | Defects Selected (final) | Purity (final) |
> |:---:|:---:|:---:|:---:|:---:|
> | 10% | 371 | 1 | 0 | **100.00%** |
> | 20% | 700 | 2 | 2 | **99.91%** |
> | 40% | 1,167 | 8 | 8 | **99.69%** |
>
> Even at 40% noise (1,167 defects in training), **only 8 anomalous samples** ever enter the selected set, and this number remains constant across all iterations. At 10% noise, the single leaked defect is eliminated by iter 1,661, reaching perfect purity. This directly contradicts the error accumulation hypothesis: **erroneous filtering does not compound over iterations** because the selection is already near-perfect from the start and purity remains constant even as the threshold progressively relaxes.
>
> >**C4: Quantitative training time comparison against base methods**
>
> Per-stage training time on MVTecAD (Dinomaly, single GPU):
>
> | Noise Ratio | Baseline Total | Memory Score | Distillation | Selection | MeDS Total |
> |---|---|---|---|---|---|
> | 0% | 51.7 min | 16.4 min | 52.2 min | 166.4 min | 234.9 min |
> | 10% | 51.9 min | 17.8 min | 51.8 min | 158.6 min | 228.2 min |
> | 20% | 52.2 min | 19.2 min | 52.1 min | 154.8 min | 226.0 min |
> | 40% | 51.6 min | 21.2 min | 51.7 min | 150.2 min | 223.1 min |
>
> MeDS adds ~4x training time (dominated by selection).
> Selection is faster at 40% noise because more training data results in fewer epochs within the fixed 10,000 iterations. **Inference time is identical** to baseline. This one-time cost is justified: MeDS **prevents catastrophic failure (87%→99% I-AUROC at 40% noise)**.
>
> >**C5: Missing contamination-specific baselines**
>
> Our baselines cover **all published contamination-aware AD methods** with available code: SoftPatch (NeurIPS'22, memory-bank pruning), InReaCh (ICCV'23, patch-level filtering), FUN-AD (WACV'25, pseudo-label discrimination). To our knowledge, these are the **latest SOTA methods for noisy AD**. We also compare against clean-data SOTA (HVQ, Dinomaly, INP-Former) under contamination. MeDS outperforms all across **all datasets and noise ratios**. We welcome specific suggestions if we missed relevant work.
>
> >**C6: More theoretical results for $s\_\theta$**
>
> We appreciate the reviewer raising this point and apologize for not articulating the theoretical grounding more clearly. Secs. 4.2 and 4.3 are well-backed by established theory:
>
> **Distillation (Sec. 4.2)** Distilling $s\_\mathbb{M}$ into a learnable network $s\_\theta$ does not require new theory because it is directly supported by the well-established early-learning phenomenon: neural networks fit dominant (normal) patterns before memorizing rare (anomalous) ones [1][2]. This acts as an implicit low-pass filter, ensuring $s\_\theta$ inherits the separation guarantee of $s\_\mathbb{M}$ for the majority-normal distribution. Providing a separate theorem here would restate existing results.
>
> **Progressive Selection (Sec. 4.3)** We kindly refer the reviewer to our response to Reviewer mFsc (C6), where we provide a detailed statistical justification of the selection threshold.
>
> **References:**
>
> [1] Arpit et al. "A Closer Look at Memorization in Deep Networks." ICML, 2017.
>
> [2] Liu et al. "Early-Learning Regularization Prevents Memorization of Noisy Labels." NeurIPS, 2020.

---

> > ### Author Rebuttal · Reviewer_Diz7 · 2026-04-03
> >
> > I thank the authors for their responses. Most of my concerns are addressed. Although it seems that the method is time-costly compared to the baselines, given IAD with anomaly contamination is an important problem and the proposed method is effective in this scenario, I update my score.

---

> > > ### Author Response · Authors · 2026-04-08
> > >
> > > We sincerely thank the reviewer for the continued engagement and for acknowledging that MeDS is effective for the important problem of IAD under anomaly contamination!
> > >
> > > > **R1: Training time cost**
> > >
> > > We agree that training time is the main practical limitation. Motivated by the reviewer's feedback, we tried to speed up MeDS training in two ways: (1) reducing the number of training iterations in both the distillation and selection stages (from 10,000 to 5,000), and (2) tuning the selection interval, i.e., running data selection once every $T_{sel}$ epochs ($T_{sel} = 5, 10, 15$) instead of every epoch. Below, we report the performance comparison in (I-AUROC / P-AP) using Dinomaly baseline on MVTecAD under different noise ratios.
> > >
> > > **Training time and performance comparison.** The original MeDS uses 10k iterations with selection every epoch ($T_{sel}=1$), while the tuned variant uses 5k iterations with selection every $T_{sel}=15$ epochs.
> > >
> > >
> > > |       Methods       | Total Time | Time Ratio |        0%        |        10%        |          20%          |        40%        |
> > > | :-------------------: | :----------: | :----------: | :-----------------: | :-----------------: | :---------------------: | :-----------------: |
> > > | Baseline (Dinomaly) |  51.6 min  |     1x     | **99.64** / 68.19 |   95.19 / 58.22   |     92.16 / 54.60     |   87.38 / 53.00   |
> > > |   MeDS (original)   | 223.1 min |   ~4.3x   |   99.36 / 68.11   | **99.49** / 68.96 |     99.31 / 68.63     | **99.16** / 67.51 |
> > > |     MeDS tuned     | 79.02 min |   ~1.5x   | 99.36 /**68.24** | 99.39 /**69.48** | **99.32** / **69.46** | 99.00 /**67.87** |
> > >
> > > The tuned MeDS achieves performance similar to the original MeDS while significantly reducing training time (from ~4.3x to ~1.5x of baseline). The remaining ~1.5x overhead over the baseline is also modest and practical.
> > >
> > > **Detailed analyses of selection interval**
> > >
> > > For a more profound analysis, we analyze the selection interval (i.e. selection per $T_{sel}$ epochs) for different iterations below. Below, $T_{sel}$ denotes the selection interval in epochs.
> > >
> > > **(a) Effect of selection interval with 10k iterations:**
> > >
> > >
> > > | $T_{sel}$ | Total Time | Time Ratio |        0%        |        10%        |          20%          |        40%        |
> > > | :--: | :----------: | :----------: | :-----------------: | :-----------------: | :---------------------: | :-----------------: |
> > > | 1 | 223.1 min |   ~4.3x   |   99.36 / 68.11   | **99.49** / 68.96 |     99.31 / 68.63     | **99.16** / 67.51 |
> > > | 5 | 144.03 min |   ~2.8x   |   99.43 / 68.04   |   99.41 / 69.48   |     99.36 / 68.86     |   99.03 / 68.28   |
> > > | 10 | 135.05 min |   ~2.6x   | **99.45** / 68.22 | 99.48 /**69.57** | **99.38** / **68.97** | 99.08 /**68.43** |
> > > | 15 | 131.05 min |   ~2.5x   | 99.44 /**68.32** |   99.46 / 69.47   |     99.37 / 67.43     |   98.99 / 68.37   |
> > >
> > > Increasing the selection interval from every epoch ($T_{sel}=1$) to every 15 epochs ($T_{sel}=15$) reduces training time substantially while keeping performance nearly unchanged across all noise ratios.
> > >
> > > **(b) Effect of selection interval with 5k iterations:**
> > >
> > >
> > > | $T_{sel}$ | Total Time | Time Ratio |        0%        |          10%          |          20%          |          40%          |
> > > | :--: | :----------: | :----------: | :-----------------: | :---------------------: | :---------------------: | :---------------------: |
> > > | 1 | 117.13 min |   ~2.3x   |   99.36 / 68.11   |     99.39 / 69.30     | **99.39** / **69.57** | **99.16** / **68.60** |
> > > | 5 | 81.96 min |   ~1.6x   | **99.39** / 68.06 |     99.38 / 69.37     |     99.30 / 69.36     |     99.01 / 68.17     |
> > > | 10 | 80.33 min |   ~1.6x   | 99.33 /**68.30** |   99.35 /**69.48**   |     99.32 / 69.38     |     99.01 / 68.06     |
> > > | 15 | 79.02 min |   ~1.5x   |   99.36 / 68.24   | **99.39** / **69.48** |     99.32 / 69.46     |     99.00 / 67.87     |
> > >
> > > The same trend holds with reduced iterations (5k) for distillation and selection fine-tuning: increasing the selection interval further reduces training time while performance stays similar to $T_{sel}=1$ across all noise ratios.

---

### Official Review · Reviewer_tSS6 · 2026-03-10

**Soundness:** 3
**Presentation:** 2
**Significance:** 3
**Originality:** 3
**Overall Recommendation:** 3
**Confidence:** 3

**Summary:**

This paper addresses a highly practical problem in industrial visual anomaly detection: training robust detection and localization models when the training set is contaminated by an unknown ratio of anomalous samples. The authors propose a three-stage training framework named Memory-Distilled Selection (MeDS). The core idea is to use an ensemble of sparsely subsampled memory banks as a low-pass filter for coarse-level noise filtering of the training data, yielding a relatively clean initial model. This robustness is then transferred via score-space distillation to a learnable reconstruction network, which is finally fine-tuned using progressive data selection based on its own outputs to achieve fine-grained learning of normal patterns. The paper provides extensive evaluation across standard datasets under various noise ratios, demonstrating significant robustness advantages over existing baselines, maintaining near-perfect image-level detection even at high noise ratios. The method is further validated through thorough ablation studies, theoretical analysis, and a demonstration for active label cleaning.

**Compliance With Llm Reviewing Policy:**

Affirmed.

**Key Questions For Authors:**

To what extent does the "Strict Separability" assumption in Theorem 1 hold in practice? Is there a theoretical or empirical breaking point for the effectiveness of sparse subsampling as a low-pass filter when anomalies and normal samples severely overlap in the pretrained feature space, or when the anomaly ratio is extremely high? Please further elucidate how the unimodality of ω(m,r) necessarily leads to maximum separation at a specific m, rather than a monotonic relationship.

The three-stage framework appears complex. Could stronger experiments demonstrate each stage is indispensable? For example, compare the performance of "direct data selection and fine-tuning using the memory ensemble scores" versus "distillation followed by selection fine-tuning"? Furthermore, were the scheduling schemes for the interpolation coefficient α_t and critical value k_t sufficiently compared against alternative schedules? These designs appear somewhat heuristic.

The paper primarily compares against other AD methods. However, the problem of noisy training data has been extensively studied in general classification/segmentation. Have the authors considered adapting some classic noisy label learning techniques, like GCE loss, Co-teaching to the pixel-level reconstruction framework of this task and comparing them with MeDS? This would help better position the uniqueness of MeDS's innovation.

**Limitations:**

The starting point and the teacher signal heavily rely on feature encoders pretrained on external data like ImageNet. The performance ceiling is bounded by the discriminative power of these generic features for specific industrial defects. If the domain gap is extreme and pretrained features fail, the foundation of the entire method could be compromised.

During the progressive selection in the third stage, erroneous filtering by the early-stage model may introduce some anomalous samples into the "clean" subset, potentially polluting subsequent training. While the authors designed a conservative initial threshold, the risk of such error accumulation warrants attention, especially under extremely high noise or when anomalies are subtle. A sensitivity analysis of this risk is lacking.

As noted in Appendix B.2, the method assumes and utilizes image class information for both memory scoring and data selection. While often feasible in practice, this limits the general applicability of the method to scenarios where classes are completely unknown or fine-grained subclasses are mixed. Without class-wise separation, differences between normal samples of different classes might be mistaken for anomalies, harming the filtering efficacy.

**Strengths And Weaknesses:**

Strengths:

The work precisely targets a critical pain point in industrial application: training with imperfect data. The proposed three-stage framework is logically clear and offers a novel integration of the robustness of non-parametric memory methods, the representational capacity of parametric models, and self-training ideas.

The experimental section is exceptionally thorough, covering multiple datasets, noise ratios, image/pixel-level metrics, and application to several base models. The results consistently show that MeDS effectively enhances noise robustness, with particularly graceful degradation at high noise levels, which is highly convincing. Analyses on hyperparameters, training iterations, etc., are also detailed.

While relying on strong assumptions, the theoretical effort to formalize the effect of sparse subsampling adds depth to the work. The demonstration of MeDS scores aiding efficient manual label cleaning is a valuable applied perspective, highlighting practical utility.

Weaknesses:

Theorem 1 in Section 4.1 and its derivation in the appendix are central to understanding the role of sparse subsampling. However, the theorem relies on assumptions like "Strict Separability" and ultimately decomposes the expected gap into a weighted integral of a signal term. While the authors claim the unimodality of the weight function leads to optimal separation at an intermediate memory size, the universality of this conclusion and its connection to specific data distributions and feature-space geometry remain abstract and indirectly argued. Figure 3 is more an empirical demonstration than a full validation of the theoretical mechanism. Stronger intuition or a more general theoretical grounding is needed to convince the reader this is not merely an empirical observation specific to the experimental setup.

As shown in Tables 1 and 2, applying MeDS sometimes leads to a slight performance drop compared to the base model at 0% noise ratio. The authors attribute this in the limitations to the median-based selection potentially excluding informative normal samples. This explanation warrants deeper empirical analysis: for instance, on perfectly clean data, are the excluded samples indeed those with "hard-to-learn" or "marginal" characteristics? The trade-off between this minor performance sacrifice and the substantial robustness gain needs more explicit discussion.

The method involves three stages: memory ensemble construction, distillation, and iterative selection fine-tuning. While additional computational cost is mentioned in the limitations, the paper lacks quantitative comparisons of training time and memory footprint against the base methods. This is crucial for assessing the practical deployment cost of the approach.

---

> ### Author Rebuttal · Authors · 2026-03-31
>
> We sincerely thank the reviewer for the thorough and insightful review.
>
> >**C1: Theorem 1 theoretical grounding and Strict Separability**
>
> We sincerely thank the reviewer for this insightful question on our core theoretical contribution.
>
> > Unimodality of w(m,r) and maximum separation?
>
> If $\omega(m,r)$ were monotonic in $m$, $\Delta(m)$ would be maximized at the largest or smallest $m$. Unimodality rules out both extremes, implying an intermediate $m$ maximizes $\Delta(m)$. Since $\omega$ jointly depends on $r$ through $\pi_{norm}(r)$, the exact optimal $m$ cannot be derived in closed form. **Figure 3** empirically confirms this trend.
>
> > Does "Strict Separability" hold in practice? Breaking point?
>
> Strict Separability requires $\pi_{norm}(r) \geq \pi_{anom}(r)$, holding when training data is majority-normal. It breaks when contamination is high enough that subsamples routinely contain anomalies. Since the decomposition holds regardless of $\delta(r)$'s sign, partial violation degrades the gap gradually, not catastrophically. Figure 3 confirms this: **at higher noise, performance degrades faster with larger subsampling ratios, as more anomalies violate Strict Separability**.
>
> >**C2: Performance drop at 0% noise**
>
> We sincerely thank the reviewer for this observation. We characterized excluded samples on clean MVTecAD, defining "hard" normals as the top 50% highest-loss samples:
>
> | Iteration | Hard Rejected | Hard Retained | % Hard Retained |
> |:---:|:---:|:---:|:---:|
> | 0 | 1,150 | 661 | 36.5% |
> | Final | 1,143 | 668 | 36.9% |
>
> MeDS retains **36.9% hard samples** with **<1pp I-AUROC drop**, confirming excluded samples carry redundant information. This <1pp cost prevents catastrophic failure under noise (**87% to 99% I-AUROC at 40%, Table 1**).
>
> >**C3: Quantitative training time comparison against base methods**
>
> We sincerely thank the reviewer for this practical concern. Please refer to our response to Reviewer Diz7 C4, where we provide a detailed per-stage training time breakdown across all noise ratios.
>
> >**C4: Stage necessity and scheduling**
>
> **(a) Stage necessity:** The ablation results in Table 4 show the below, validating the effectiveness of each component. For the detailed discussion, we kindly ask the reviewer to refer to the response to C3 of Reviewer j6Z8
>
> **(b) Scheduling for $\alpha_t$ and $k_t$:** We chose the simplest (linear) schedule, as our early experiments showed different schedules yield negligible performance change.
>
> >**C5: Comparison with noisy-label learning**
>
> We sincerely thank the reviewer for this important point. Noisy-label learning techniques were a strong motivation for our work. We attempted to adapt GCE and Co-teaching to AD, but direct application was not feasible: (1) Co-teaching relies on multi-class supervision, but AD has no supervision label to differentiate between anomaly and normal; (2) GCE requires softmax over multiple categories, **absent** in reconstruction-based AD. These incompatibilities led us to design MeDS, where the **bootstrapped memory ensemble (Sec. 3.2)** serves as an AD-native alternative.
>
> >**C6: Reliance on pretrained encoders**
>
> We sincerely thank the reviewer for raising this point. This limitation is shared by all teacher-student AD methods as they all rely on pretrained encoders, and MeDS is no exception. However, our primary objective is to improve robustness against noise rather than to address this fundamental limitation.
>
> >**C7: Error accumulation in progressive selection**
>
> We sincerely thank the reviewer for this concern. Please refer to our response to Reviewer Diz7 C3, where we provide a detailed purity analysis across all noise ratios and training iterations.
>
> >**C8: Class information assumption**
>
> We acknowledge this can be regarded as a limitation. However, we'd like to noe that, in practical industrial settings, every product has a known type/SKU (stock keeping unit), and **anomaly patterns can be inherently class-dependent** (a scratch on product type A may be normal surface texture on product type B), which necessitates class-aware training. Moreover, all noisy AD methods in our comparison (SoftPatch, InReaCh, FUN-AD) similarly assume class information during training. In addition, MeDS uses class information only during training; at inference, a single model is deployed across all classes.

---

> > ### Author Rebuttal · Reviewer_tSS6 · 2026-04-01
> >
> > First of all, I would like to thank you for your response. However, there are still a few questions that have not been adequately answered.
> >
> > 1. Breaking point? I did not receive any responses related to the breaking point.
> > 2. Too many supplementary experiments need to be found in the responses of other reviewers, but the character limit has not been reached, so is it because my comments are not important? I'm not sure if the comments of the other reviewers are exactly the same as mine.
> > 3. The overall style of the illustrations in this article is too simplistic and fails to effectively convey the content of the paper.
> > 4. Could you provide more detailed visual results?

---

> > > ### Author Response · Authors · 2026-04-08
> > >
> > > We are deeply grateful to the reviewer, whose thoughtful and persistent engagement has substantially sharpened both the theory and empirical analyses of MeDS. We sincerely apologize for relying on cross-references in our first-round response.
> > >
> > > > **R1: Breaking point of Strict Separability**
> > >
> > > We conducted dedicated breaking-point experiments on MVTecAD using the Dinomaly backbone with a fixed subsampling ratio of 0.1. $\Delta(m)$ is the separability gap (memory's distance to an anomalous query minus its distance to a normal query; positive means the memory still separates them). $m$ is the post-subsampling memory size; Total Patches is the full patch count.
> > >
> > >
> > > |Image Noise Ratio|Patch Noise Ratio|$m$|Total Patches|AUROC|$\Delta(m)$|
> > > |:-:|:-:|:-:|:-:|:-:|:-:|
> > > |40%|2.38%|17,640|176,400|0.701|0.0260|
> > > |80%|4.79%|8,781|87,810|0.647|0.0200|
> > > |95%|5.64%|7,448|74,480|0.640|0.0196|
> > > |99%|5.89%|7,135|71,350|0.639|0.0195|
> > > |--|10%|4,203|42,030|0.545|0.0078|
> > > |--|15%|2,802|28,020|0.491|0.0006|
> > > |--|20%|2,102|21,020|0.454|-0.0047|
> > >
> > > Rows 1–4 are realistic (controlling image-level anomaly rate); rows 5–7 set the patch noise ratio beyond natural levels to locate the breaking point, using smaller controlled datasets.
> > >
> > > As the patch noise ratio rises, $\Delta(m)$ approaches zero — the memory can no longer separate normal from anomalous patches — and at 15% the model reaches its breaking point with AUROC collapsing. **Conversely, as noise decreases (closer to the strict separability assumption), the memory more reliably distinguishes anomalous from normal patches**, providing direct empirical support for our theory.
> > >
> > > > **R2: Cross-references to other reviewers**
> > >
> > > We provide as detailed results as possible directly below for each cross-referenced point.
> > >
> > > **Training time (C3):** The original MeDS trains distillation and selection for 10k iterations each and re-runs selection every epoch, adding ~3.3x on top of baseline (selection-dominated). Results on MVTecAD (Dinomaly, 40% noise):
> > >
> > >
> > > |Methods|Total Time|Time Ratio|I-AUROC|P-AP|
> > > |:-:|:-:|:-:|:-:|:-:|
> > > |Baseline (Dinomaly)|51.6 min|1x|87.38|53.00|
> > > |MeDS (original)|223.1 min|~4.3x|**99.16**|67.51|
> > > |MeDS-tuned|79.02 min|~1.5x|99.00|**67.87**|
> > >
> > > MeDS-tuned (5k iterations, selection every 15 epochs) reduces training from **~4.3x** to **~1.5x** with at most 0.16pp I-AUROC drop. **Inference time is unchanged**.
> > >
> > > **Stage necessity and scheduling (C4)**
> > >
> > > *Stage necessity.* We ablate each of the three MeDS stages in turn and report the gain of full MeDS over each variant (I-AUROC / P-AP, pp) on MVTecAD: **Memory absence** zeroes the distillation target $s_\mathbb{M}$ (removes Stage 1); **Distillation absence** anchors progressive selection on the memory score $s_\mathbb{M}$ instead of the distilled model's score $s_\theta$ (removes Stage 2); **Naive fine-tuning** replaces progressive self-selection with fine-tuning on unfiltered data (removes Stage 3).
> > >
> > >
> > > |Ablation point|Model|0%|10%|20%|40%|
> > > |:-|:-:|:-:|:-:|:-:|:-:|
> > > |Memory absence: distillation from zero score|Dinomaly|+0.11 / -0.40|+1.40 / +3.41|+2.15 / +3.25|+3.18 / +3.45|
> > > |Distillation absence: selection by memory score anchor|INP-Former|+2.20 / +2.55|+1.52 / +2.64|+1.37 / +2.61|+1.04 / +1.75|
> > > |Naive fine-tuning in the third stage|Dinomaly|-0.27 / +0.47|+1.82 / +7.51|+3.53 / +10.77|+6.51 / +11.23|
> > >
> > > Full MeDS dominates every variant, and the gap tends to grow monotonically with noise, showing that all three stages are needed and that each matters more as the data becomes harder. Stage 3 carries the largest share (up to **+6.51 / +11.23** at 40% noise), but Stages 1 and 2 also contribute non-trivially (**+3.18 / +3.45** and **+2.20 / +2.55** respectively).
> > >
> > > *Scheduling.* Following the reviewer's suggestion, we compared four schemes for $\alpha_t$ and $k_t$ at 40% noise on MVTecAD. **Step** applies an abrupt midpoint transition ($\alpha_t: 1 \to 0$, $k_t: 0 \to 1$); **Constant** fixes both at midpoint ($\alpha_t = k_t = 0.5$).
> > >
> > >
> > > |Schedule|I-AUROC|P-AP|
> > > |:-:|:-:|:-:|
> > > |Linear (ours)|99.16|68.05|
> > > |Cosine|99.07|68.36|
> > > |Step|98.97|68.42|
> > > |Constant|99.08|68.15|
> > >
> > > The method is robust to the scheduling choice; we chose linear for its simplicity.
> > >
> > > **Error accumulation (C7):** We measured how many defective samples leak into the selected subset at the first and final iterations:
> > >
> > >
> > > |Noise|Total Defects|Defects Selected (iter 0)|Defects Selected (iter final)|Purity|
> > > |:-:|:-:|:-:|:-:|:-:|
> > > |10%|371|1|0|100.0%|
> > > |20%|700|2|2|99.9%|
> > > |40%|1,167|8|8|99.7%|
> > >
> > > Even at 40% noise, only 8 anomalous samples ever enter the selected set, and this count stays constant — errors do not accumulate.
> > >
> > > > **R3: Illustration style & R4: Visual results**
> > >
> > > OpenReview disallows figure uploads, so results are presented as tables above.
> > >
> > > We again sincerely apologize for the brevity imposed by space limits, and refer the reviewer to our Diz7-R1 (training time) and j6Z8-R3 (stage necessity, additional ablations) responses for fuller discussion.

---

### Official Review · Reviewer_mFsc · 2026-03-12

**Soundness:** 3
**Presentation:** 2
**Significance:** 3
**Originality:** 3
**Overall Recommendation:** 4
**Confidence:** 4

**Summary:**

Visual Anomaly Detection  usually assumes all training data is normal, but real industrial datasets often contain unnoticed defects.
The paper proposes Memory-Distilled Selection (MeDS) to handle contaminated training data.
MeDS first builds a bootstrapped ensemble of sparse memories to produce coarse anomaly scores.
Next, memory score distillation trains a student network to mimic these scores.
Finally, progressive fine-tuning with data selection iteratively filters cleaner samples using robust statistics.

**Compliance With Llm Reviewing Policy:**

Affirmed.

**Final Justification:**

I find the responses of the authors satisfactory about many comments and I am now convinced of the soundness of the proposed method.
My assessment of the empirical contribution remains unchanged, as the observed improvements appear modest and some corrections should be performed to fix significant errors.
Additionally, I believe the presentation of the method could be refined to better reflect its specific application within the teacher–student framework.
Therefore,  I appreciate the additional explanations and though some limitations are present I am incline to suggest a weak accept recommendation

**Key Questions For Authors:**

1. Why does the paper not include comparisons with widely recognized VAD models? Including these would better contextualize the results.
2. Why are lesser-known VAD methods like HVC used in combination with your approach, instead of classic methods like PatchCore or other standard baselines?
3. Why is the code not shared? Public access would improve reproducibility and transparency of the methodology.
4. Do the reported results show a substantial improvement over previous models? The current gains appear limited, raising questions about the practical impact of the method.

**Limitations:**

yes

**Strengths And Weaknesses:**

Strengths:
1. The issue of “contamination” is highly relevant for real-world implementations and has been discussed in the VAD literature.
2. The authors propose a straightforward and practical approach to address this problem.
3. They demonstrate consistent, though modest, improvements over existing methods and validate their approach across multiple VAD benchmarks, strengthening the robustness of their results.
4. An ablation study is provided to analyze the contribution of each component, improving the transparency and interpretability of the proposed method.
Weaknesses:
Major:
1. Why was the paper on DiNomaly not cited? It seems relevant to the topic and could provide important context.
2. The methods HVC and INP-Former are not widely recognized in the VAD literature. Why were these specific methods selected, and why are the references for them not provided?
3. Why were more straightforward or well-established methods, such as PatchCore or other standard VAD approaches, not considered to be combined with your method?
4. The design of the reconstructor is unclear. Could you clarify how it is defined in terms of architecture?
5. The reported performance improvement appears modest, which raises questions about the practical usefulness of the proposed approach.
6. The code is not shared (e.g., using anonymous repositories), which limits reproducibility and reduces the transparency of the methodology.
Minor:
Some statements do not seem solid:
1. While the first part of the method aims to provide strong formal guarantees, Section 4.3 relies heavily on heuristics without explaining why these are reasonable or justified.
2. The statement that “this involves a fundamental trade-off” is made without sufficient evidence or theoretical justification.
3. The claim that “pretrained features do not sufficiently distinguish…” is stated but not rigorously supported by experiments or analysis.
4.  "Unlike standard teacher-student distillation in AD...".  The method does not implement TS as described, since inference does not use the difference between the two scores, based on my understanding.

---

> ### Author Rebuttal · Authors · 2026-03-31
>
> We sincerely thank the reviewer for their constructive feedback.
> >**C1 & C2: Missing Dinomaly citation, unrecognized baselines (HVQ, INP-Former), and why not PatchCore?**
>
> We apologize for the missing references. HVQ [1], INP-Former [2], and DiNomaly [3] will be cited in the revision.
>
> Regarding baselines: **MeDS requires a trainable student network**, so we selected the strongest teacher-student models: HVQ, Dinomaly, and INP-Former. Memory-only methods like PatchCore have no learnable parameters, making MeDS inapplicable.
>
> Moreover, **we compared against all published contamination-aware AD methods in Tables 1--2**: **SoftPatch** (NeurIPS'22, noise-robust PatchCore), **InReaCh** (ICCV'23), and **FUN-AD** (WACV'25). We also report PatchCore on Real-IAD (**Table 3**).
>
> >**C3: Unclear design of the reconstructor architecture**
>
>  **MeDS is architecture-agnostic**: the reconstructor $s\_\theta$ follows the **exact same architecture as the base model**. The anomaly score is $s\_\theta(x)\_{hw} = 1 - \cos(g(x)\_{hw}, f\_\theta(x)\_{hw})$, where $g$ is the **frozen teacher** and $f\_\theta$ is the **trainable student**. In **Dinomaly**, the student is an MLP bottleneck + 8 linear-attention transformer blocks; in **HVQ**, hierarchical vector quantization; in **INP-Former**, a transformer with input-adaptive negative prompts.
>
> >**C4: Modest performance improvement**
>
> We appreciate the reviewer's concern. We believe the improvements are in fact substantial, especially under high contamination:
>
> - **MVTecAD 40% noise** (**Table 1**): Dinomaly 87.38% to **99.16%** I-AUROC (**+11.78pp**), P-AP 53.00 to **68.05** (**+15.05pp**). INP-Former 85.85% to **99.17%** (**+13.32pp**).
> - **VisA 10% noise** (**Table 2**): INP-Former+MeDS achieves **96.30 I-AP** vs. 94.52 (**+1.78pp**). P-AP: 39.89 to **43.51** (**+3.62pp**).
> - **Real-IAD** (**Table 3**): Dinomaly+MeDS achieves the best across **all noise ratios** in both image and pixel metrics.
>
> These gains (up to **+13.32pp I-AUROC**, **+15.05pp P-AP**) show MeDS **prevents catastrophic failure** under severe contamination.
>
> >**C5: Code availability**
>
> We will release the complete code right after acceptance.
>
> >**C6: Progressive selection in Sec. 4.3 relies on heuristics without justification**
>
> We respectfully clarify that the selection threshold (Eq. 12) is not a heuristic but a **statistically principled outlier detection rule**. For any random variable with finite mean and variance, Cantelli's inequality [4] guarantees that at most $1/(1+k^2)$ of the population exceeds $\mu + k\sigma$, without any distributional assumption. This directly justifies thresholding at location + $k \times$ scale as a distribution-free principle. We use Median + MAD as robust counterparts of mean and std, achieving the maximum **50% breakdown point** [5]. The schedule $k\_t: 0 \to k$ implements a curriculum where each step's purity is bounded by this inequality.
>
> >**C7: Unsupported claim on pretrained features**
>
> We thank the reviewer for this point. This limitation is **well-established in the existing works**. **PANDA** [6] demonstrates that **frozen pretrained features alone are insufficient for AD**, showing consistent gains from feature adaptation. **ADPretrain** [7] establishes that **ImageNet pretraining is suboptimal for AD** due to distribution shift between natural and industrial images. These findings motivate distillation and selection fine-tuning of MeDS.
>
> >**C8: Inference does not use teacher-student score difference, contradicting the TS description**
>
>  We respectfully clarify that MeDS **does use teacher-student discrepancy as the anomaly score at inference**. Specifically, the anomaly score is $s\_\theta(x)\_{hw} = 1 - \cos(g(x)\_{hw}, f\_\theta(x)\_{hw})$, i.e., the cosine discrepancy between the frozen teacher $g$ and the learned student $f\_\theta$ (**Eq. 3**). The final anomaly map averages these discrepancy maps across feature levels, where higher discrepancy indicates anomalous regions. **This is consistent with standard teacher-student-based AD**.
>
> **References:**
>
> [1] Lu et al. "Hierarchical Vector Quantized Transformer for Multi-class Unsupervised Anomaly Detection." NeurIPS, 2023.
>
> [2] Luo et al. "Exploring Intrinsic Normal Prototypes within a Single Image for Universal Anomaly Detection." CVPR, 2025.
>
> [3] Guo et al. "Dinomaly: The Less Is More Philosophy in Multi-Class Unsupervised Anomaly Detection." CVPR, 2025.
>
> [4] Cantelli, F. P. "Sui confini della probabilità." *Atti del Congresso Int. dei Matematici*, 1928.
>
> [5] Rousseeuw, P. J. & Croux, C. "Alternatives to the median absolute deviation." *JASA*, 1993.
>
> [6] Reiss et al. "PANDA: Adapting Pretrained Features for Anomaly Detection and Segmentation." CVPR, 2021.
> [7] Yao et al. "ADPretrain: Advancing Industrial Anomaly Detection via Anomaly Representation Pretraining." NeurIPS, 2025.

---

> > ### Author Rebuttal · Reviewer_mFsc · 2026-04-03
> >
> > I find the responses of the authors satisfactory about many comments and I am now convinced of the soundness of the proposed method.
> > My assessment of the empirical contribution remains unchanged, as the observed improvements appear modest and some corrections should be performed to fix significant errors.
> > Additionally, I believe the presentation of the method could be refined to better reflect its specific application within the teacher–student framework.
> >
> >
> > C1&C2:
> > Thank you for the clarification, personally I believe that figure 1 could be improved to better represent the student-teacher/encoder-decoder architecture.
> > Based on the authors' explanations, I am convinced of the soundness of the proposed method, though my view on the modest improvement remains unchanged. Given the revisions made to certain sections, I am inclined to raise the score.
> > Furthermore, it was not clear from the outset that the method was developed specifically for teacher-student approaches, as this was not explicitly stated in the text. While I do not consider this a limitation of the method, it should be declared explicitly in the paper.
> >
> > C3&C8:
> > I want to thank the authors to clarify this important aspect.
> >
> > C4:
> > For Real-IAD in Table 3, the I-AUROC metric shows 84.89 for MEds+Dinomaly, while SoftPatch alone reaches 85.72 for contamination 10, which is higher than the proposed method.
> > Nevertheless, the authors claim that their model achieves the best performance in this case, which appears to be a significant error and corrections should be performed to improve transparency.
> >
> > C5:
> > unfortunately is difficult to judge the transparency without the code
> >
> > C6:
> > I would also like to point out that while the provided justification is compelling, it is incorrectly applied in the proposed method.
> > Specifically, since the method relies on the median rather than the average, the stated justification no longer holds and the claim cannot be considered valid.
> >
> > C7:
> > thank you to provide reference to back the stament

---

> > > ### Author Response · Authors · 2026-04-08
> > >
> > > We sincerely thank the reviewer for the thorough assessment, for confirming the soundness of the method, and for the constructive feedback that has helped us improve the manuscript. We also appreciate the reviewer’s confirmation that the clarifications for C3, C7, and C8 were satisfactory. We address the remaining points below.
> > >
> > > > **C1 & C2: Teacher-student scope**
> > >
> > > We are deeply grateful to the reviewer for these acknowledging comments and insightful suggestions, which have helped us sharpen the presentation of our method. In the revised manuscript, we will explicitly clarify, consistently across the introduction, the method section, and Figure 1, that MeDS targets teacher-student AD frameworks, and we will frame HVQ, Dinomaly, and INP-Former as concrete instantiations of this scope (with all baseline papers properly cited).
> > >
> > > > **C4: Correction to the Real-IAD claim and the "modest improvement" assessment**
> > >
> > > *Correction.* We thank the reviewer for pointing this out and apologize for the oversight. At 10% noise on Real-IAD, SoftPatch reaches **85.72 I-AP**, exceeding Dinomaly+MeDS at **84.89 I-AP**. The text in the main paper **will be corrected** to state that MeDS *outperforms or is on par with* existing methods on Real-IAD, and to explicitly note SoftPatch's lead on I-AP at 10% noise.
> > >
> > > *Regarding the **"modest improvement"** concern*. We believe the image-level performance results in Table 3 understate MeDS’s improvement on Real-IAD, and that the sample-level metric proposed for this dataset provides a more complete picture.
> > >
> > > Concretely, in Real-IAD each sample (object) is captured from five views. On an anomalous sample, the defect can be unclear or not visible in some views due to viewing angle or lighting, so image-level AUROC penalizes a method for not flagging views in which the defect is effectively absent. The original Real-IAD paper [1] therefore proposes **sample-level AUROC (S-AUROC)**, which integrates the per-view scores of a sample and compares them against the sample-level label; Dinomaly2 [2] reports the same quantity under the name *object-level AUROC*. Under S-AUROC and the corresponding sample-level F1-max, MeDS shows **a consistent, marked lead over SoftPatch** at every noise level:
> > >
> > > Sample-level results on Real-IAD (**S-AUROC** / **S-F1max**):
> > >
> > >
> > > | Method | 0% | 10% | 20% | 40% |
> > > | :-- | :-: | :-: | :-: | :-: |
> > > | SoftPatch | 94.21 / 90.01 | 92.81 / 88.22 | 92.07 / 87.31 | 90.05 / 84.86 |
> > > | Dinomaly+MeDS | 95.16 **(+0.95 pp)** / 91.28 **(+1.27 pp)** | 94.77 **(+1.96 pp)** / 90.72 **(+2.50 pp)** | 93.95 **(+1.88 pp)** / 89.64 **(+2.33 pp)** | 92.66 **(+2.61 pp)** / 88.45 **(+3.59 pp)** |
> > >
> > > Notably, the gap widens as the noise ratio increases, indicating that MeDS is especially beneficial under heavier noise.
> > >
> > > > **C5: Reproducibility**
> > >
> > > We agree that reproducibility and transparency are vastly important. The full codebase, including MeDS integration for HVQ, Dinomaly, and INP-Former, as well as the training and evaluation scripts behind every table, will be fully released upon acceptance, along with the exact contamination splits for datasets.
> > >
> > > > **C6: Median/MAD and Cantelli's inequality**
> > >
> > > We agree with the reviewer in that Cantelli's inequality [3] is a statement about mean/std and does not, on its own, license the median/MAD form. Another view that justifies substituting the mean and standard deviation by the median and MAD is the following: Cantelli [3] motivates the threshold $\mu+k\sigma$ as a sensible outlier criterion when $(\mu,\sigma)$ describe the *clean* population. Since our data is contaminated, we approximate that clean mean and standard deviation by the median and a suitably scaled MAD, which is the standard robust surrogate from the modified z-score literature [4]. Despite this, we acknowledge that neither our manuscript nor the rebuttal provides a strict, mathematically rigorous justification for the median/MAD substitution; instead, we rely on established practice and leave its formal proof to future work.
> > >
> > > ---
> > >
> > > We sincerely thank the reviewer again for the careful and constructive assessment, which hugely improved our paper! The Real-IAD correction, the median/MAD reframing, and the Figure 1 and scope clarifications will all be incorporated in the revised manuscript.
> > >
> > > **References**
> > >
> > > [1] Wang et al. *Real-IAD: A Real-World Multi-View Dataset for Benchmarking Versatile Industrial AD.* CVPR 2024. arXiv:2403.12580.
> > >
> > > [2] Guo et al. *One Dinomaly2 Detect Them All: A Unified Framework for Full-Spectrum Unsupervised AD.* arXiv:2510.17611, 2025.
> > >
> > > [3] Cantelli, F. P. *Sui confini della probabilità.* Atti del Congresso Int. dei Matematici, 1928.
> > >
> > > [4] Iglewicz, B. & Hoaglin, D. C. *How to Detect and Handle Outliers.* ASA, 1993.

---

### Decision · Program_Chairs · 2026-04-30

**Decision:**

Accept (regular)

**Comment:**

Based on the reviews and the authors’ rebuttal, I recommend weak acceptance. Reviewers agreed that the paper addresses a relevant and practical problem of handling contaminated training data in anomaly detection, and presents a clear method with consistent gains. Some concerns were raised about modest improvements or slight degradation in low-contamination settings, but the authors demonstrated strong gains under high contamination and provided convincing responses. Overall, despite relying on some strong assumptions, the theoretical effort to formalize the effect of sparse subsampling is well received by the reviewers.